# A modular strategy for extracellular vesicle-mediated CRISPR-Cas9 delivery through aptamer-based loading and UV-activated cargo release

Omnia M. Elsharkasy[1,10], Charlotte V. Hegeman [2,10], Tom A. P. Driedonks [1], Xiuming Liang [3,4,5], Ivana Lansweers[2], Olaf L. Cotugno[1], Ingmar Y. de Groot[1], Zoë E. M. N. J. de Wit[2], Antonio Garcia-Guerra[6,7], Niels J. A. Moorman[2], Sjoerd H. Boonstra[2], Esmeralda D. C. Bosman[2], Juliet W. Lefferts[2], Willemijn S. de Voogt [1], Jerney J. François[1], Annet C. W. van Wesel[1], Samir El Andaloussi [3,4,5], Raymond M. Schiffelers [1], Sander A. A. Kooijmans [1,8,9], Enrico Mastrobattista [2], Pieter Vader [1] & Olivier G. de Jong [1,2]

CRISPR-Cas9 gene editing technology offers the potential to permanently repair genes containing pathological mutations. However, efficient intracellular delivery of the Cas9 ribonucleoprotein complex remains a major hurdle in its therapeutic application. Extracellular vesicles (EVs) are biological nano-sized membrane vesicles that play an important role in intercellular communication, and have an innate capability of intercellular transfer of biological cargos, including proteins and RNA. Here, we present a versatile, modular strategy for EV-mediated loading and delivery of Cas9. We leverage the high affinity binding of MS2 coat proteins fused to EV-enriched proteins to MS2 aptamers incorporated into guide RNAs, in combination with a UV-activated photocleavable linker domain, PhoCl. Moreover, we demonstrate that Cas9 can readily be exchanged for other variants, including transcriptional activator dCas9-VPR and adenine base editor ABE8e. Taken together, we describe a robust, modular strategy for successful Cas9 delivery, which can be applied for CRISPR-Cas9-based genetic engineering and transcriptional regulation.

The CRISPR (Clustered Regularly Interspaced Short Palindromic Repeats)-Cas9 gene editing system offers the potential to permanently modify or repair genes that contain pathological mutations[1]. As such, CRISPR-Cas9 has significant therapeutic potential in the treatment of a plethora of pathologies with underlying genetic causes. The CRISPR-Cas9 system is based on the use of the prokaryotic Cas9 endonuclease, which is guided to a specific genetic sequence by a single guide RNA (sgRNA)[2,3]. Together, Cas9 and the sgRNA form a ribonucleoprotein

(RNP) complex that can generate a double-stranded DNA break at a specific genomic locus that is targeted by the spacer sequence in the sgRNA[4].

However, the last decade has seen a vast expansion of additional CRISPR-Cas9 based technologies, by fusing various functional domains to a catalytically inactivated Cas9 protein (dCas9)[5]. Such functional domains may provide a variety of functionalities, such as transcriptional activators or inhibitors, which facilitate temporary

inhibition or induction of transcription of target genes[6]. Alternatively, Cas9 nickases (nCas9) may be fused to base editing domains to change specific nucleotides within the sgRNA target sequence. Here, an adenine base editor consists of an N-terminal fusion of a deoxyadenosine deaminase (TadA) to nCas9, resulting in A > G conversion, or a cytosine base editor contains a cytidine deaminase, resulting in a C > T conversion[7,8]. These base editors allow for the generation of specific mutations without the need for an HDR DNA template.

Whereas the CRISPR-Cas9 toolbox has been extensively characterized and expanded over the last decade, intracellular delivery remains one of the major hurdles for therapeutic applications. Efficient intracellular delivery of the Cas9 RNP complex is hampered due to its large size, negative charge, and immunogenicity[9]. To this end, various intracellular delivery strategies have been employed to facilitate Cas9 RNP delivery, including viral vectors, lipid nanoparticles (LNPs), polymers, and cell penetrating peptides (CPPs)[10–13]. Whereas these vectors have shown promising potential, they have substantial limitations. Viral vectors have limited cargo capacity, have increased risk of potential immunogenic responses, and long-term expression of the CRISPR-Cas9 system from viral expression vectors increases the chance of unintended off-target effects, and clearance of target cells by the immune system[14,15]. Both LNPs and polymer-based systems have shown potential for the delivery of Cas9 RNP and mRNA, but show cellular toxicity and limited delivery efficiency due to endosomal entrapment[16]. That being said, recent studies on the effects of formulation on biodistribution have lead to an increased understanding and control of LNP tissue targeting[17,18]. CPPs show similar limitations and may suffer from decreased particle stability in serum-rich environments[19]. Thus, there is a need for a for a CRISPR-Cas9 delivery strategy that shows low toxicity and immunogenicity, and high stability in serum-rich conditions.

Extracellular vesicles (EVs) are a heterogeneous population of lipid bilayer nanoparticles released by all cell types, and play an important role in intercellular communication[20]. EVs facilitate intercellular communication through the transfer of various biological cargos, including (transmembrane) proteins, various types of RNA molecules, lipids, and metabolites. Intercellular communication is shown to be facilitated through direct receptor-ligand interactions at the cell surface, or through intracellular delivery of the intraluminal EV cargo in the recipient cell[21]. This form of intercellular communication has been shown to play an important role in a large number of physiological and pathological processes, including immune regulation, tumor progression, angiogenesis, and wound healing[22–25]. Given their innate capacity of intercellular transfer of biological cargos, the capability of crossing difficult-to-cross biological barriers, and their potential for low immunogenicity, EVs have raised increasing interest as potential vectors for therapeutic delivery of a variety of biological cargos, including proteins and RNA. Various studies have shown that these natural nanoparticles can be enriched with therapeutic proteins during biogenesis by expressing fusion proteins of EV-enriched proteins and the protein of interest in the EV-producing cells[26,27]. Such EV-enriched proteins commonly include tetraspanins CD9, CD63, and CD81. One potential limitation of this approach for Cas9 loading is that fusion of an EV-enriched protein directly to Cas9 may interfere with the functionality of the aforementioned additional enzymatic domains present on the C-terminus or the N-terminus of Cas9. Moreover, as Cas9 needs to translocate to the nucleus upon delivery, tethering of Cas9 to the EV membrane may prevent its transfer to the appropriate cellular compartments in recipient cells.

Here, we present a modular and versatile strategy to facilitate EV-mediated delivery of various Cas9-based applications. This strategy is based on the fusion of RNA-binding domains to EV-enriched moieties, separated by a UV-cleavable linker[28,29]. The RNA-binding domains within these fusion proteins facilitate the loading of Cas9 RNPs into EVs by high-affinity binding of aptamers that are incorporated into the tetraloop and second stem loop of the Cas9 sgRNA[30]. As such, no direct fusion to either the C- or N-terminus of Cas9 is required, providing a flexible platform for delivery of a multitude of Cas9 variants. As cleavage of the UV-cleavable linker can be efficiently activated after EV isolation, this approach shows efficient delivery of various Cas9 variants, including transcriptional activation and adenine base editing, using multiple reporter systems as well as Cas9 and ABE8e gene editing on the endogenous *CCR5* gene. Altogether, this work demonstrates the development of a versatile and efficient EV-based Cas9 RNP delivery platform.

## Results

### Generation of a modular aptamer-based strategy for Cas9-sgRNA RNP loading into EVs

As the interest in Cas9-mediated gene therapy has increased over the last decade, the abundance of Cas9-based tools, such as transcriptional regulators and base editors, has grown along with it. In order to generate a universal loading strategy that can be applied interchangeably to all these variants, we opted to avoid the use of direct Cas9-tetraspanin fusion proteins. Instead, we intraluminally fused tandem MS2 coat proteins (MCPs), lacking the Fg loop to prevent capsid formation, to the N-terminus of the EV-enriched tetraspanin CD63 (Fig. 1A). These MCP proteins robustly bind to MS2 aptamers, specific RNA hairpin sequences. To bind the Cas9-sgRNA ribonucleoprotein (RNP) complex, MS2 aptamers that can be bound by the tandem MCPs were placed in the tetraloop and second stemloop of the sgRNA. Previous studies have shown that replacing the hairpins in these sections of the sgRNA does not interfere with the Cas9 RNP functionality, and that these hairpins protrude from the Cas9 RNP structure[30]. Thus, we envisioned that cellular co-expression of MS2-sgRNAs and Cas9 results in the formation of intracellular Cas9 RNPs with protruding MS2 aptamers, that can then be bound by the intraluminal tandem MCPs fused to CD63. As CD63 is highly enriched in EVs, we hypothesized that this would result in Cas9 RNP loading during EV biogenesis (Supplementary Fig. S2A). In this modular approach, sgRNAs and Cas9 are readily interchangeable, as no direct Cas9 fusion proteins are employed.

HEK293T cells were transfected with plasmids expressing the MCP-CD63 loading construct, Cas9, and MS2-sgRNAs, and 48 h later EVs were isolated by tangential flow filtration (TFF), followed by size exclusion chromatography (SEC). Nanoparticle tracking analysis (NTA, Fig. 1B) and transmission electron microscopy (TEM, Fig. 1C) confirmed the isolation of particles with a morphology and size distribution (mode = 75.1 +/− 5.1 nm) in line with the common observed characteristics of EVs. Additionally, western blot analysis showed enrichment for common EV-markers CD63, ALIX and TSG101, and a negative enrichment for the organelle marker Calnexin, as compared to cell lysates (Fig. 1D). To assess loading of Cas9 into EVs, HEK293T cells were transfected with plasmids for expressing Cas9 and MS2-sgRNA, with- or without the co-expression of MCP-CD63. Western blot analysis revealed that Cas9 was highly enriched in EVs when MCP-CD63 was co-expressed, confirming the efficacy of this loading strategy (Fig. 1E). To ensure that the observed Cas9 protein was indeed associated with EVs, isolated EVs were further analyzed using an OptiPrep density gradient. Indeed, the presence of Cas9 was only observed in EV-associated fractions positive for the EV-marker CD63 (Fig. 1F). This was further confirmed by immune precipitation of MCP-CD63 RNP-loaded EVs with CD63, CD9, or IgG isotype control antibodies, using magnetic protein G-coated beads (Supplementary Fig. S2A). Indeed, using beads with a CD63 antibody resulted in an enrichment of Cas9 in the pulldown fraction as compared to its respective supernatant sample. CD9 capture did result in the pulldown of some CD63 and Cas9 protein, but the majority of both proteins remained in the supernatant (Supplementary Fig. S2B). No Cas9 protein or EV markers were detected after a

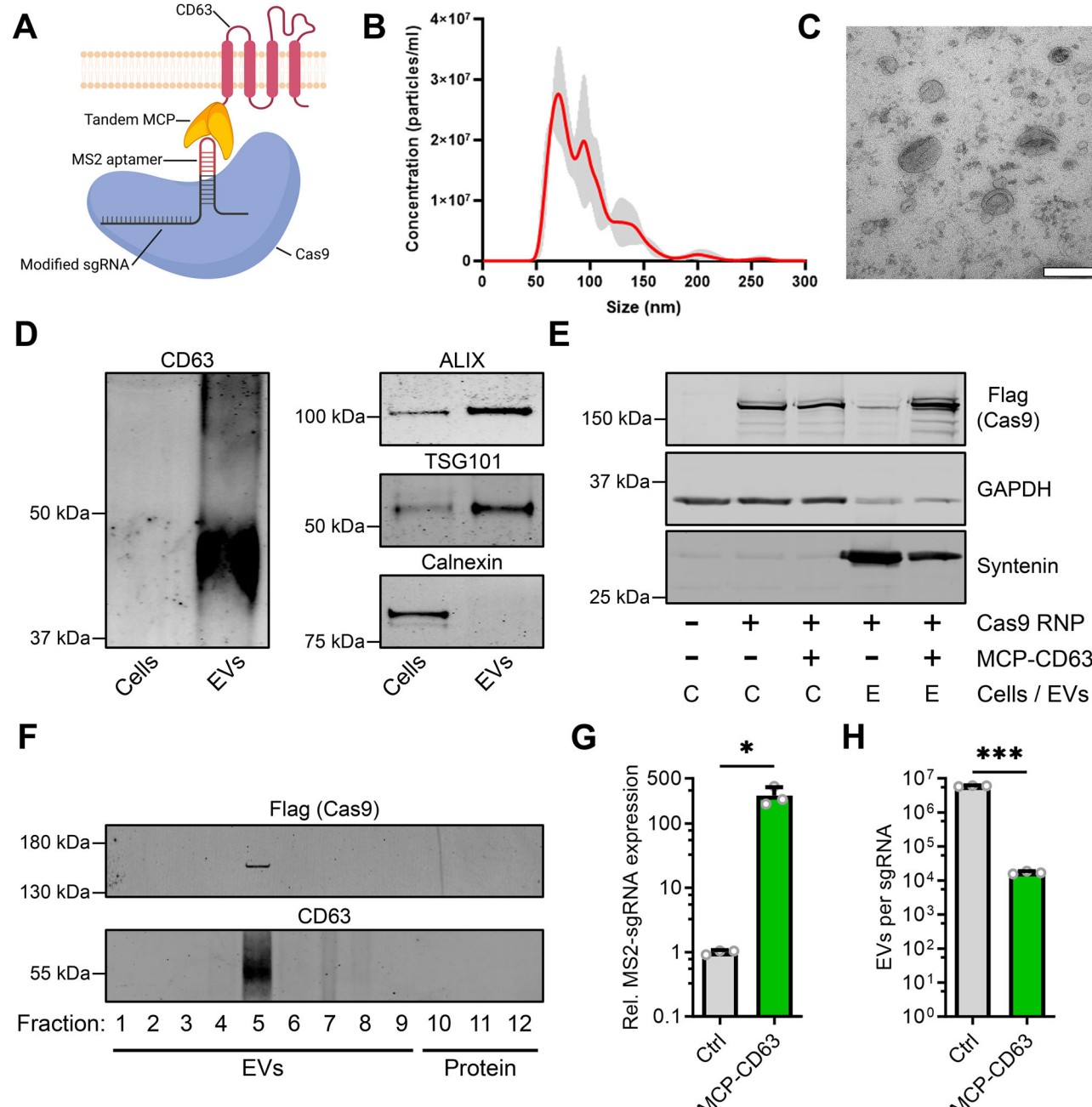

**Fig. 1 | Engineering EVs for the targeted loading of Cas9 ribonucleoprotein complexes. A** Schematic of the EV engineering strategy for active loading and delivery of Cas9 RNPs in HEK293T-derived EVs. Tandem MCPs, lacking the Fg loop involved in capsid formation, are intraluminally fused to the N-terminus of EV-enriched CD63 (MCP-CD63). These MCPs bind MS2 aptamers protruding from the RNP, which are present on MS2-modified sgRNAs. Created in BioRender. Utrecht University, P. (2025) https://BioRender.com/0up6d3r. **B** Nanosight particle tracking analysis displaying the size distribution of isolated EVs. Means ± SD, $n = 5$ technical replicates. Representative NTA analysis, observed > 10 times in independent experiments. **C** Transmission electron microscopy image of isolated EVs, scalebar represents 200 nm. Representative TEM analysis, observed >3 times in independent experiments. **D** Western blot analysis of cell lysates and EVs shows an enrichment for EV markers CD63, ALIX and TSG101, and a negative enrichment for the ER organelle marker Calnexin, in isolated EVs. Due to the highly glycosylated nature of CD63, its western blot analysis presents a commonly observed "smear" pattern. Representative Western Blots, observed > 3 times in independent experiments. **E** Western blot analysis shows similar levels of Cas9 in cell lysates regardless of co-expression of MCP-CD63, whereas high levels of Cas9 in isolated EVs are only observed upon co-expression of MCP-CD63. Representative Western Blots, observed 2 times in independent experiments. **F** Western blot analysis of an OptiPrep density gradient of isolated EVs shows presence of Cas9 in the same EV-associated fractions as EV-marker CD63, $n = 1$. **G, H** qPCR analysis (**G**) of RNA isolated from EVs derived from cells MS2-sgRNA with- or without the co-expression of MCP-CD63 shows enriched loading of MS2-sgRNAs into isolated EVs, which is further quantified by ddPCR analysis (**H**), corrected for NTA particle count and Spike-in RNA, to calculate RNA isolation efficiency, into absolute EV per sgRNA counts. Means + SD, $n = 3$ biologically independent samples, Student's $t$ test. * $p < 0.05$, *** $p < 0.001$.

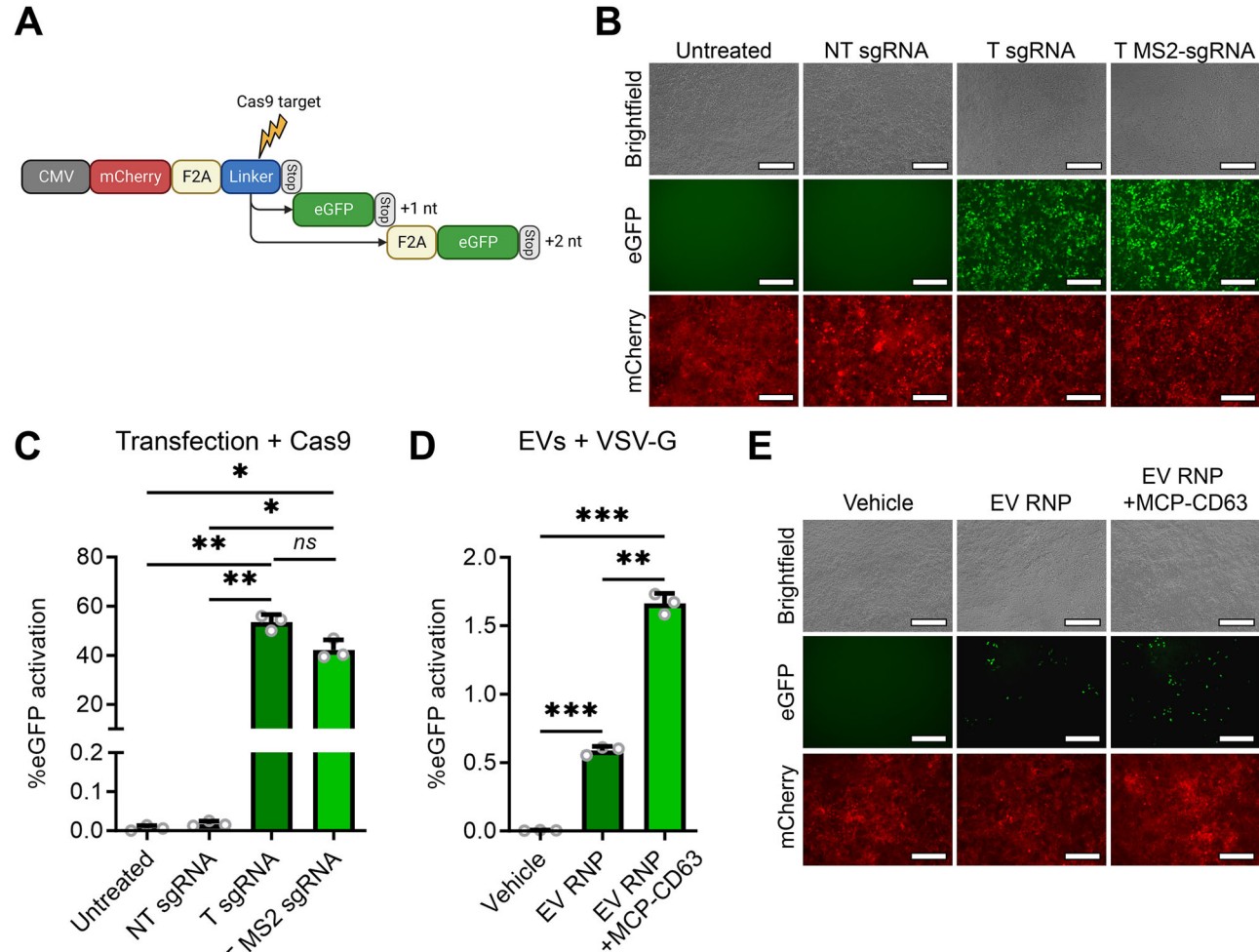

**Fig. 2 | MCP-CD63 facilitates a limited increase of EV-mediated Cas9 RNP delivery. A** Schematic of the fluorescence "stoplight" reporter construct for Cas9 activity. mCherry (red) is stably expressed under a CMV promoter followed by a small "linker" region (blue), containing a Cas9 target site, and a stop codon. Frameshifts in the linker region, resulting from non-homologous end joining (NHEJ)-mediated repair mechanisms due to Cas9-mediated double stranded breaks, result in expression of downstream eGFP (green) open reading frames. Created in BioRender. Utrecht University, P. (2025) https://BioRender.com/olgq16g. **B, C** Fluorescence microscopy images (**B**) and flow cytometry analysis (**C**) of HEK293T cells expressing the stoplight reporter construct, transfected with Cas9 and a non-targeting (NT) sgRNA, or a targeting (T) sgRNA with or without MS2 aptamers. eGFP expression is observed after transfection of Cas9 with both wild-type targeting (T) sgRNA or a T MS2-sgRNA. Scalebar represents 200 μm. Means + SD, $n = 3$ biologically independent samples, One-way ANOVA with post-hoc Tukey's multiple comparisons test. **D, E** Flow cytometry analysis (**D**) and fluorescence microscopy images (**E**) of HEK293T cells expressing the stoplight reporter construct, 72 h after addition of EVs isolated from HEK293T cells expressing Cas9 + MS2-sgRNA + VSV-G (EV RNP) or expressing Cas9 + MS2-sgRNA + MCP-CD63 + VSV-G (EV RNP + MCP-CD63) shows that MCP-CD63 facilitates a significant but limited increase of EV-mediated RNP delivery. $1.0 \times 10^{12}$ EVs per well. Scalebar represents 200 μm. Means + SD, $n = 3$ biologically independent samples, One-way ANOVA with post-hoc Tukey's multiple comparisons test. * $p < 0.05$, ** $p < 0.01$, *** $p < 0.001$.

bead capture pulldown using isotype control IgG. These data confirm that Cas9 loaded through MCP-CD63 is, at least partially, enriched in CD63-positive EVs.

To assess whether MS2-sgRNA was also actively enriched in EVs by MCP-CD63, MS2-sgRNA loading was analyzed by qPCR (Fig. 1G). As observed for Cas9 protein loading, co-expression of MCP-CD63 resulted in a significant increase in MS2-sgRNA abundance in isolated EVs, showing an approx. 270 fold increase (SD ± 94). To more accurately and quantitatively assess MS2-sgRNA loading, digital droplet PCR (ddPCR) was performed (Fig. 1H). Using a Spike-in RNA to correct for RNA isolation efficiency, absolute RNA loading was extrapolated from NTA particle counts. In line with previous results, a substantial increase in MS2-sgRNA loading was observed of 350-fold (SD ± 12.4), increasing the abundance from 1 sgRNA per ~ $6.0 \times 10^6$ EVs to 1 sgRNA per ~ $1.7 \times 10^4$ EVs after active loading using MCP-CD63. Whereas these numbers are low as compared to synthetic nano-carriers, such orders of magnitude are not unexpected, as similar

numbers have previously been reported for sgRNA loading into EVs in other cell lines[31,32].

To confirm and analyze EV-mediated Cas9 RNP delivery to target cells in a sensitive and robust manner, we employed a fluorescent reporter for Cas9 activity that we previously published[31]. In this fluorescent "stoplight reporter system", an mCherry open reading frame (ORF), followed by a short linker region that contains a Cas9 target site, is constitutively expressed. When Cas9 and a targeting sgRNA are functionally delivered, a resulting frameshift can activate one of the out-of-frame downstream eGFP ORFs, resulting in permanent high eGFP expression (Fig. 2A). First, to test whether inclusion of MS2 aptamers in the sgRNA tetraloop and second stemloop did not interfere with RNP functionality, HEK293T cells expressing the stoplight reporter construct were transfected with Cas9 in combination with a non-targeting (NT) sgRNA, a targeting (T) sgRNA, or a targeting MS2 (T MS2)-sgRNA, and analyzed by fluorescence microscopy (Fig. 2B) and flow cytometry (Fig. 2C and Supplementary Fig. S1A). Both analyses

showed no activation of eGFP expression when Cas9 was co-transfected with an NT sgRNA, and showed high levels of activation when Cas9 was co-transfected with both the wild-type T sgRNA and the T MS2-sgRNA. Moreover, no significant decrease in Cas9-mediated activation of eGFP expression was observed in T MS2-sgRNA as compared to WT T sgRNA, confirming that this modification does not interfere with Cas9 activity. Next, the capacity of the engineered EVs to deliver Cas9 RNPs was assessed using the stoplight reporter construct. To further enhance EV-mediated Cas9 delivery, EV-producing cells were also transfected to express the Vesicular stomatitis virus (VSV) envelope glycoprotein VSV-G[33]. This glycoprotein facilitates endosomal escape of delivered cargo in recipient cells by inducing membrane fusion in the late endosome, triggered by the local decrease in pH. To test the effect of MCP-CD63 on EV-mediated Cas9 delivery, $1.0 \times 10^{12}$ EVs from cells expressing VSV-G, Cas9 and MS2-sgRNA, with- or without MCP-CD63 co-expression, were isolated and added to HEK293T stoplight reporter cells. Whereas both flow cytometry (Fig. 2D) and fluorescence microscopy (Fig. 2E) confirmed an increase in EV-mediated Cas9 delivery due to MCP-CD63 expression, levels of gene editing in the reporter cells were surprisingly low, at less than 2%. Addition of $5.0 \times 10^{10}$ and $1.0 \times 10^{11}$ EVs showed even lower levels of gene editing, at 0.2% and 0.3% respectively (Supplementary Fig. S3A, B).

## Incorporation of a photocleavable domain facilitates efficient cargo release and delivery

Since the significant increase in MS2-sgRNA and Cas9 loading by MCP-CD63 did not result in a similar level of increase in Cas9 delivery, we hypothesized that the loaded Cas9 RNP was not sufficiently released from the EV membrane due to the strong binding affinity of the MS2 aptamers in the sgRNA to the MCPs tethered to CD63. As a result, Cas9 would be unable to translocate into the nucleus of the target cell. To address this issue, we introduced the PhoCl photocleavable protein between the tandem MCP domains and the CD63 tetraspanin (Fig. 3A)[29]. This PhoCl protein is a derivate of the photoconvertible mMaple protein, and is cleaved when exposed to 395 nm (ultra)violet light. To confirm that this construct was indeed cleaved upon UV exposure, HEK293T cells were transfected with the MCP-PhoCl-CD63 construct, and after 48 h were exposed to UV light from a 50W 395 nm LED panel for various lengths of time. As the PhoCl domain was given an HA-tag, protein cleavage from the ~82 kDa fusion protein into a ~55 kDa subunit could be observed by western blot analysis (Fig. 3B). Here, protein cleavage was confirmed, and optimal cleavage in cells was achieved in a 10 – 20 min timeframe. UV exposure to EVs isolated from cells expressing the MCP-PhoCl-CD63 construct showed somewhat faster kinetics (Fig. 3C), likely due to cells being treated in 1 ml full culture medium, whereas EVs were treated with UV in a small volume of PBS.

Next, EVs from cells expressing VSV-G, Cas9, MS2-sgRNA, and MCP-PhoCl-CD63 were isolated and subsequently used for direct addition to stoplight reporter cells, or were first exposed to UV light for 20 min on ice prior to addition to stoplight cells. 48 hours after EV addition, flow cytometry analysis (Fig. 3D) and fluorescence microscopy (Fig. 3E) both confirmed a significant and substantial increase in EV-mediated Cas9 delivery in MCP-PhoCl-CD63 EVs treated with UV, increasing recombination in reporter cells from ~2% to ~28%. To confirm that these observations are not specific to HEK293T reporter cells, a cell line that is easy to transfect[34], 3 additional reporter cell lines were tested for EV-mediated Cas9 delivery (Supplementary Fig. S4). Alongside HEK293T cells (Supplementary Fig. S4A, B), human primary tumor cell lines MDA-MB-231 (Supplementary Fig. S4C, D), MCF-7 (Supplementary Fig. S4E, F), and T47D (Supplementary Fig. S4G, H) were treated with various dosages of MCP-PhoCl-CD9 and MCP-PhoCl-CD63 VSV-G⁺ EVs. All cell types showed a dose-dependent EV-

mediated delivery of Cas9, albeit at a 5 – 10 fold lower efficiency as compared to HEK293T cells.

To assess whether the 395 nm UV treatment of the EVs prior to addition to recipient cells did not have adverse effects on the functionality of the loaded Cas9 RNPs, VSV-G⁺ EVs loaded with MCP-CD63 or MCP-PhoCl-CD63 were used for an addition assay, with- or without UV treatment. As observed before, flow cytometry analysis (Supplementary Fig. S7C) and fluorescence microscopy (Supplementary Fig. S7D) showed that UV treatment of MCP-PhoCl-CD63 EVs highly increased Cas9 delivery. Importantly, we also observed that UV treatment of MCP-CD63 EVs without PhoCl did not result in a decrease in Cas9 delivery, confirming that the UV treatment did not negatively affect the EV-associated Cas9 RNP functionality. To further study whether UV treatment had any deleterious effects on EV morphology or cargo, MCP-CD63 EVs were treated for 20 min with UV and characterized by NTA analysis (Supplementary Fig. S5). No significant changes in size distribution (Supplementary Fig. S5A, B), mean and mode particle size (Supplementary Fig. S5C, D), or in particle count (Supplementary Fig. S5E) were observed. Whereas UV wavelengths generally associated with nucleotide damage are below 340 nm[35], we measured the effect of UV treatment on MS2-sgRNA abundance in MCP-CD63 RNP-loaded EVs using qPCR (Supplementary Fig. S5F) and ddPCR (Fig. S5G) analysis to rule out UV-mediated sgRNA degradation. Indeed, both qPCR and ddPCR showed no difference in EV MS2-sgRNA abundance after UV treatment. Lastly, we measured the effect of UV treatment on protein cargo content and studied the effect on EV integrity and morphology using western blot analysis and transmission electron microscopy, respectively. Western blot analysis showed no difference in levels of Cas9 protein after UV treatment, or in EV markers ALIX, CD63, and syntenin. Total protein stain also showed no notable differences (Supplementary Fig. S5H). Transmission electron microscopy also did not reveal any changes in EV morphology, integrity, or particle count (Supplementary Fig. S5I, J). To determine whether Cas9 delivery through UV-treated EVs resulted in increased off-target gene editing events due to potential UV-mediated sgRNA damage, genomic DNA of HEK293T stoplight cells treated with MCP-PhoCl-CD63 VSV-G⁺ EVs, or plasmid DNA transfection, for Cas9 delivery was isolated and analyzed by amplicon next-generation sequencing (NGS). Using Cas-OFFinder[36], 5 off-target sites with 2 – 4 nucleotide mismatches were uncovered and analyzed (Supplementary Fig. S6A–G). EV-mediated delivery did not result in any observable increase in off-target gene editing events. Lastly, we performed an MTS assay on HEK293T cells after MCP-PhoCl-CD63 EV or plasmid DNA transfection-mediated delivery of Cas9 to test for potential toxicity (Supplementary Fig. S6H). The MTS assay showed no signs of toxicity over the course of 72 hours after MCP-PhoCl-CD63 RNP-loaded VSV-G⁺ EVs were added. Plasmid DNA transfection of the same constructs showed a trend towards a low level of toxicity ($p = 0.086$), and treatment with Triton X-100 resulted in complete loss of viability. Altogether, we observe no evidence of notable deleterious changes in EV content or morphology after 20 min of 50W 395 nm UV exposure.

Having established an efficient platform for EV-mediated Cas9 loading and delivery, we opted to test the effect of replacing CD63 with alternative EV-enriched moieties on EV-mediated Cas9 delivery efficiency. To this end, CD63 was replaced with tetraspanins CD9 or CD81, Arrestin Domain Containing 1 (ARRDC1), or an N-terminal myristoylation tag (Fig. 3F). EVs were isolated from cells expressing these loading constructs alongside VSV-G, MS2-sgRNA and Cas9, normalized by nanoparticle tracking analysis, treated with 395 nm UV, and added to stoplight reporter cells. Over the course of multiple experiments, flow cytometry analysis (Fig. 3G) and fluorescence microscopy (Fig. 3H) both confirmed that the choice of EV-enriched fusion protein has a significant effect on EV-mediated Cas9 delivery levels. Whereas CD81 performed similarly to CD63, both CD9 and the myristoylation tag showed significantly higher levels of Cas9 delivery, with CD9

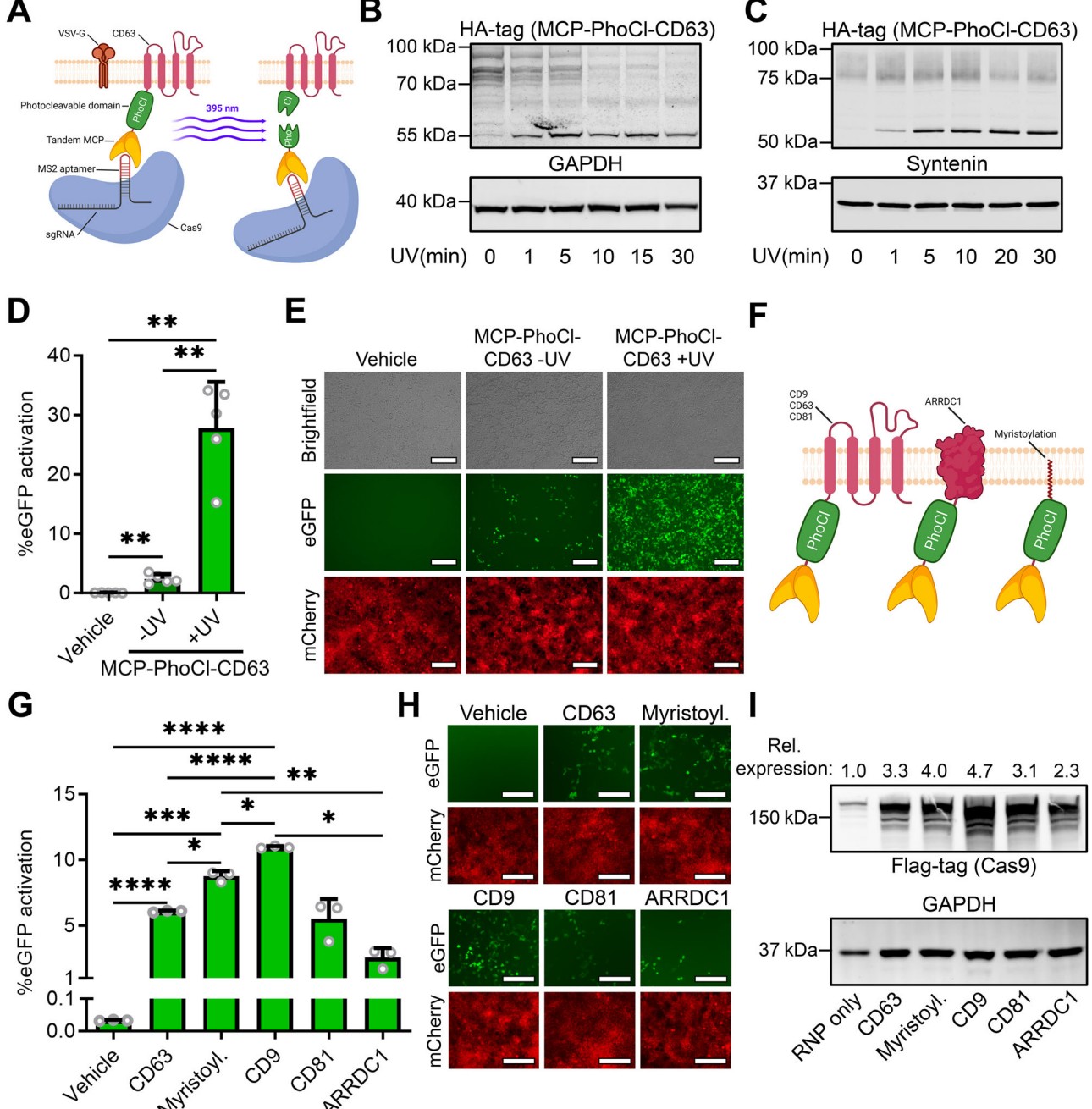

**Fig. 3 | Incorporation of a photocleavable domain to facilitate cargo release strongly increases EV-mediated Cas9 RNP delivery. A** Schematic of the EV engineering strategy for photo-activatable release of membrane-bound MCP-loaded Cas9 RNPs. A photocleavable domain (PhoCl), with an N-terminal HA-tag for western blot analysis, is placed between the tandem MCPs and CD63 (MCP-PhoCl-CD63). Upon exposure to 395 nm UV light, PhoCl is cleaved, releasing the MCP-RNP complex from the EV membrane. Created in BioRender. Utrecht University, P. (2025) https://BioRender.com/r3y3yik. **B, C** Western blot analysis of MCP-PhoCl-CD63 cleavage in cells (**B**) and isolated EVs (**C**). Upon UV exposure, the ~82 kDa fusion protein is cleaved, revealing a ~55 kDa cleavage product. **D, E** Flow cytometry analysis (**D**) and fluorescence microscopy images (**E**) of HEK293T cells expressing the stoplight reporter construct, 72 h after addition of EVs isolated from HEK293T cells expressing Cas9 + MS2-sgRNA + MCP-PhoCl-CD63 + VSV-G shows that UV treatment of EVs prior to addition to cells strongly increases EV-mediated

RNP delivery. $1.0 \times 10^{12}$ EVs per well. Mean + SD, $n = 5$ independent experiments, One-way ANOVA with post-hoc Tukey's multiple comparisons test. Scalebar represents 200 µm. **F** Schematic of additional EV-targeted loading constructs. Tandem MCPs are fused to EV-enriched moieties CD9, CD63, CD81, ARRDC1, and a myristoylation tag via a photocleavable (PhoCl) domain. Created in BioRender. Utrecht University, P. (2025) https://BioRender.com/f95r962. **G, H** Flow cytometry analysis (**G**) and fluorescence microscopy images (**H**) comparing EV-mediated RNP delivery of various MCP-PhoCl fusion proteins. Addition normalized by particle count; $5.0 \times 10^{10}$ particles added per well. Scalebar represents 200 µm. Means + SD, $n = 3$ independent experiments, One-way ANOVA with post-hoc Tukey's multiple comparisons test. **I** Western blot analysis of Cas9 loading in EVs by various MCP-PhoCl fusion proteins. Relative Cas9 loading is corrected for loading control and normalized to the "RNP only" condition. * $p < 0.05$, ** $p < 0.01$, *** $p < 0.001$, **** $p < 0.0001$.

outperforming CD63 almost 2-fold. Interestingly, the ARRDC1-mediated microvesicle (ARMM) marker ARRDC1 showed substantially lower levels of Cas9 delivery. To assess whether these changes in efficiency of Cas9 delivery were the result of differences in cargo loading, EVs were analyzed for Cas9 loading abundance by western blot analysis (Fig. 3I). Indeed, western blot analysis revealed a pattern in Cas9 loading levels showing correlation with the activation of eGFP expression levels observed in Cas9 delivery, indicating that the observed differences in Cas9 delivery efficiency of these constructs are likely due to differences in levels of Cas9 loading. To further characterize the effects of the choice of EV-marker for targeted cargo loading, cells were once more transfected with MCP-PhoCl-CD9, MCP-PhoCl-CD63, and MCP-PhoCl-CD81, alongside VSV-G, MS2-sgRNA and Cas9, but were normalized by the number of EV-producing cells, instead of the number of EVs (Supplementary Fig. S8A). Similar to previous observations, CD63 and CD81 showed equal levels of Cas9 delivery, and CD9 again showed substantially higher levels of Cas9 delivery. In a further comparison, a dose response range from $1.0 \times 10^8$ – $2.5 \times 10^{12}$ EVs loaded with Cas9 using either MCP-PhoCl-CD9 or MCP-PhoCl-CD63 was added to $1.0 \times 10^5$ HEK293T reporter cells. Flow cytometry analysis (Supplementary Fig. S8B) and fluorescence microscopy (Supplementary Fig. S8C) both showed dose-dependent activation of the stoplight reporter cells, wherein MCP-PhoCl-CD9 once more substantially outperformed MCP-PhoCl-CD63 at all tested concentrations, reaching up to 57% activation at the highest dose. Taken together, these data show that the choice of the EV-enriched marker strongly affects the efficiency of EV-mediated Cas9 delivery across various conditions and dosages.

To assess the importance of including the VSV-G envelope glycoprotein, EVs from cells expressing Cas9 and MS2-sgRNA, with- or without co-transfection of MCP-CD63 (Supplementary Fig. S7A) or MCP-PhoCl-CD63 +/- UV treatment (Supplementary Fig. S7B) without co-expression of VSV-G were added to stoplight reporter cells. In all conditions, flow cytometry analysis showed no increase in eGFP expression. These data indicate that, despite the observed high increase in Cas9 RNP loading in the presence of MCP-CD63, co-expression of VSV-G is pivotal for efficient Cas9 delivery in this approach. To further elucidate the role of VSV-G in EV-mediated Cas9 delivery, EV-mediated delivery of fluorescently labeled Cas9 was analyzed by confocal microscopy. To this end, Cas9 was labeled with the highly stable mStayGold green fluorescent protein[37], and loaded into EVs using MCP-PhoCl-CD9 (Supplementary Fig. S9A), either with- or without co-expression of VSV-G. After UV-mediated cargo release, the EVs were added to HEK293T cells stably expressing mTag-BFP2 labeled Histone 2B (H2B), which gives high levels of fluorescence in the nucleus[38]. 1 hour after addition no substantial levels of Cas9-mStayGold were observed in the nuclei for both VSV-G⁻ and VSV-G⁺ MCP-PhoCl-CD9 EV-treated cells (Supplementary Fig. S9B, S9D). However, starting from 2 hours after EV addition, a significant increase in green fluorescent signal was observed in the nuclei of VSV-G⁺ EV-treated cells, as compared to both untreated and VSV-G⁻ EV treated cells (Supplementary Fig. S9C, S9D). This pattern remained unchanged up until the final timepoint of the experiment at 12 h after EV addition (Supplementary Fig. S9D), showing that VSV-G plays a pivotal role in facilitating EV-mediated Cas9 delivery to the nucleus of recipient cells.

### Delivery of Cas9 transcriptional activators and base editors

To test the versatility of this loading strategy, we tested the loading and delivery of the Cas9-based transcriptional activator dCas9-VPR[6]. Here, the nuclease domains of Cas9 have been deactivated, forming a "dead" Cas9 (dCas9). Instead, various transcriptional activators have been fused to this protein, allowing the temporary activation of a gene when targeting its promoter region. To measure dCas9-VPR activity, we generated reporter cell lines that contain a doxycycline inducible eGFP expression construct (Fig. 4A). This construct can be activated by the addition of doxycycline, which activates rtTA3, resulting in transcription of the eGFP gene downstream of a Tet Responsive Element (TRE) sequence. Alternatively, an sgRNA targeting the TRE sequence facilitates dCas9-VPR-mediated expression of eGFP. Functionality of this generated reporter construct in HEK293T cells was confirmed by fluorescence microscopy (Fig. 4B) and flow cytometry analysis (Fig. 4C). As activation of eGFP expression is induced in a dose-dependent manner, eGFP expression levels are shown in MFI (Supplementary Fig. S1B). Both analyses confirmed that the addition of doxycycline induced high levels of eGFP expression. Moreover, transfection of dCas9-VPR alongside a targeting (T) sgRNA or a T MS2-sgRNA also resulted in a substantial, albeit lower, level of eGFP expression. Unlike previously observed in Cas9 gene editing (Fig. 2C), transfection of MS2-sgRNA does show a slightly lower, but still significant, transcriptional activation of eGFP expression as compared to WT sgRNA (Fig. 4B, C). No increase in eGFP expression was observed when dCas9-VPR was transfected alongside a non-targeting (NT) sgRNA (Fig. 4C). Next, EVs were isolated from cells transfected with VSV-G, dCas9-VPR, MCP-PhoCl-CD9 or MCP-PhoCl-CD63, and NT MS2-sgRNAs or T MS2-sgRNAs, treated with 395 nm UV for 20 minutes on ice, and added to the HEK293T inducible eGFP reporter lines. 48 h later, eGFP expression levels were measured by flow cytometry analysis (Fig. 4D), showing a significant increase in MFI (ΔMFI) for both MCP-PhoCl-CD9 and MCP-PhoCl-CD63 EVs when isolated from cells expressing a T MS2-sgRNA, but not when expressing a NT MS2-sgRNA. A dose-range of MCP-PhoCl-CD63 VSV-G⁺ EVs show a dose-dependent effect on transcriptional activation (Supplementary Fig. S10A, B). However, as compared to delivery of WT Cas9, a substantially higher dose of EVs is required, as dosages of $1.0 \times 10^{11}$ EVs per $1.0 \times 10^5$ cells or lower showed no increase in eGFP fluorescence intensity. Once more, EVs loaded with MCP-PhoCl-CD9 showed a substantially higher level of reporter activation than the EVs loaded with MCP-PhoCl-CD63 (Fig. 4D). However, both showed significant levels of increase in eGFP expression as compared to untreated controls. These data confirm that the modular MCP-PhoCl-based loading platform is also suitable for the loading and delivery of the dCas9-VPR transcriptional activator, allowing the temporary transcriptional activation of targeted genes without the introduction of permanent genetic mutations.

Next, we tested the loading and delivery of adenine base editors (ABEs)[8]. Adenine base editors consist of a Cas9 nickase (nCas9), with a DNA-modifying deoxyadenosine deaminase (TadA) fused to its N-terminus. This deoxyadenosine deaminase has the capacity to change adenine to inosine, resulting in the conversion of A to G nucleotides within a limited window of the sgRNA targeting sequence. In order to study ABE delivery, we generated a fluorescent Cas9 adenine base editor (ABE) stoplight reporter construct (Fig. 5A)[13]. In this construct, mCherry is constitutively expressed under a CMV promoter, followed by a stop codon and directly thereafter an in-frame eGFP open reading frame. A PAM site was introduced 16 bp upstream of the TAA stopcodon, allowing the conversion of the TAA stop codon to CAA, encoding for a glutamine, by ABEs. This will result in the expression of an mCherry-eGFP fusion protein, of which the eGFP signal can be measured and quantified by flow cytometry (Supplementary Fig. S1C). To confirm the functionality of this ABE stoplight reporter, HEK293T reporter cells were generated and transfected with ABE7.10 with a non-targeting (NT) or a targeting (T) sgRNA. As a positive control, wildtype Cas9 with a targeting sgRNA with an HDR ssODN template was transfected. As shown by fluorescence microscopy (Supplementary Fig. S11A) and flow cytometry analysis (Supplementary Fig. S11B), both ABE7.10 with a targeting sgRNA and WT Cas9 in combination with a T sgRNA and an HDR template were able to activate eGFP expression, albeit at insufficient efficiency to serve as a suitable read-out for EV-mediated ABE delivery (12.9% and 8.5%, respectively). Next, to improve editing efficiency, the more recent

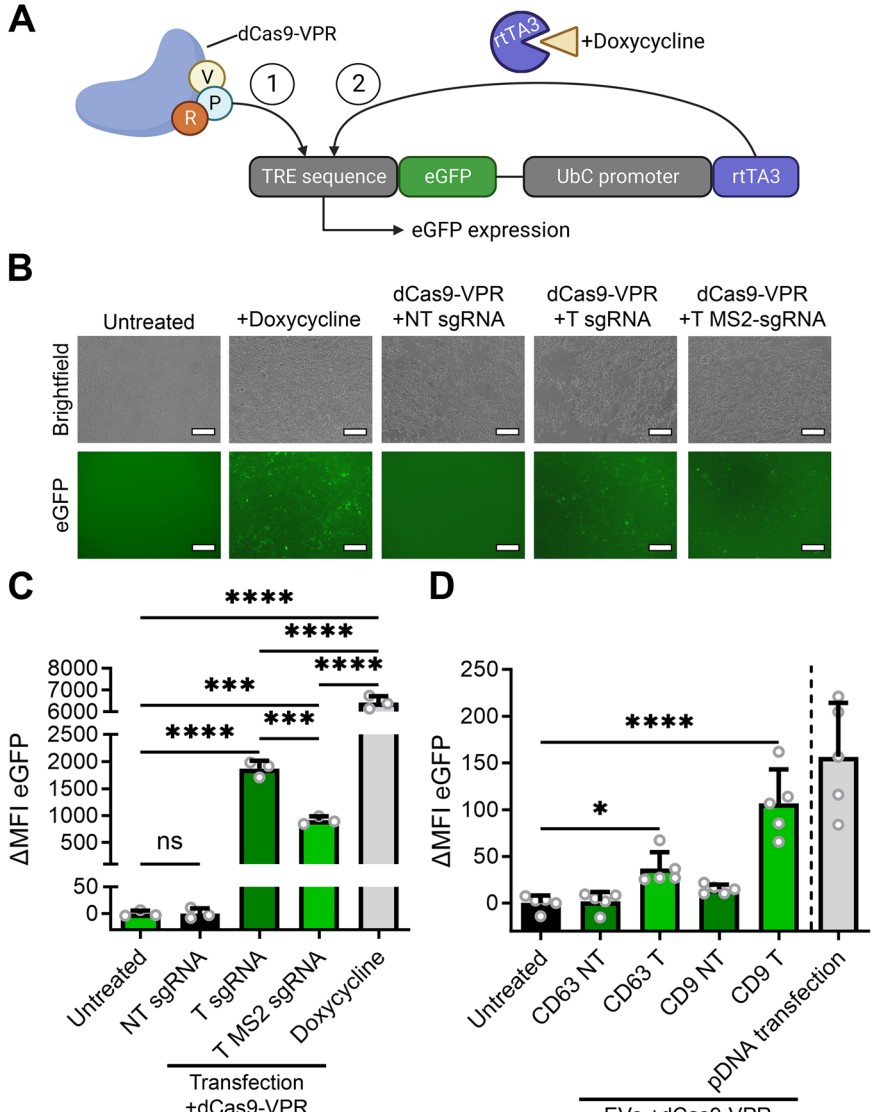

**Fig. 4 | EV-mediated functional delivery of the dCas9-VPR transcriptional activator. A** Schematic of the fluorescent reporter construct for transcriptional activation; pInducer20-eGFP. An eGFP open reading frame is placed after a Tet Responsive Element (TRE) sequence. Transcription of eGFP can either be facilitated by activation of the co-expressed reverse tet-transactivator rTA3 by addition of doxycycline (1) or by introduction of transcriptional activator dCas9-VPR with a sgRNA targeting the TRE sequence (2). Created in BioRender. Utrecht University, P. (2025) https://BioRender.com/v9au9y6. **B**, **C** Fluorescence microscopy images (**B**) and flow cytometry analysis (**C**) of HEK293T cells expressing the pInducer20-eGFP reporter construct, 48 h after addition of doxycycline (0.5 µg/ml), or transfection with plasmids encoding for dCas9-VPR with non-targeting (NT) sgRNAs, targeting (T)

sgRNAs, or targeting MS2-sgRNAs. Doxycycline and dCas9-VPR with targeting sgRNAs increase eGFP expression. MFI: mean fluorescence intensity. Scalebar represents 200 µm. Means + SD, $n = 3$ biologically independent samples, One-way ANOVA with post-hoc Tukey's multiple comparisons test. **D** Flow cytometry analysis of HEK293T cells expressing the pInducer20-eGFP reporter construct, 48 h after addition of EVs from HEK293T expressing dCas9-VPR alongside either MCP-PhoCl-CD63 or MCP-PhoCl-CD9, in combination with non-targeting (NT), or targeting (T) sgRNAs. Both loading constructs facilitate EV-mediated functional dCas9-VPR delivery, resulting in a significant increase in eGFP mean fluorescence intensity (MFI). $4.0 \times 10^{11}$ EVs per well. Means + SD, $n = 5$ independent experiments, One-way ANOVA with post-hoc Dunnett's multiple comparisons test. * $p < 0.05$, *** $p < 0.001$, **** $p < 0.0001$.

ABE8e and ABE8e-dimer proteins were compared to ABE7.10 and analyzed by fluorescence microscopy (Supplementary Fig. S11C) and flow cytometry analysis (Supplementary Fig. S11D). Transfection with both ABE8e and ABE8e-dimer resulted in a substantially higher level of eGFP expression, where both induced eGFP expression in over 60% of cells. As the ABE8e and ABE8e-dimer performed similarly, we opted to proceed with ABE8e, due to its smaller size.

Next, EVs were isolated from HEK293T cells expressing VSV-G, MS2-sgRNA, ABE8e, and MCP-PhoCl-CD63 or MCP-PhoCl-CD9, were treated with 395 nm UV, and added to the ABE stoplight reporter cells. Disappointingly, both constructs showed fairly low percentages of eGFP activation, around 1.4% and 3.5% respectively. Since the MCPs are still fused to the MS2 aptamers on the MS2-sgRNAs after UV-mediated

cleavage from the tetraspanins, we hypothesized that the MCP domains might physically block proper access for the deoxyadenosine deaminase to the DNA due to steric hindrance. Thus, we generated targeting MS2-sgRNAs with MS2 aptamers only in the Tetraloop (MS2-sgRNA 1.1), only in the second stemloop (MS2-sgRNA 1.2), in both (MS2-sgRNA 2.0), or in neither (WT sgRNA / sgRNA 1.0) (terminology in line with the manuscript from Konermann et al.[30]). To assess the effect of the MS2 aptamers on ABE8e functionality, plasmids for each of these sgRNAs were co-transfected with a plasmid expressing ABE8e into ABE stoplight reporter cells. Flow cytometry analysis showed no significant difference in eGFP expression between these sgRNAs, indicating that the MS2 aptamers themselves do not interfere with ABE8e functionality (Fig. 5D). However, when an additional plasmid

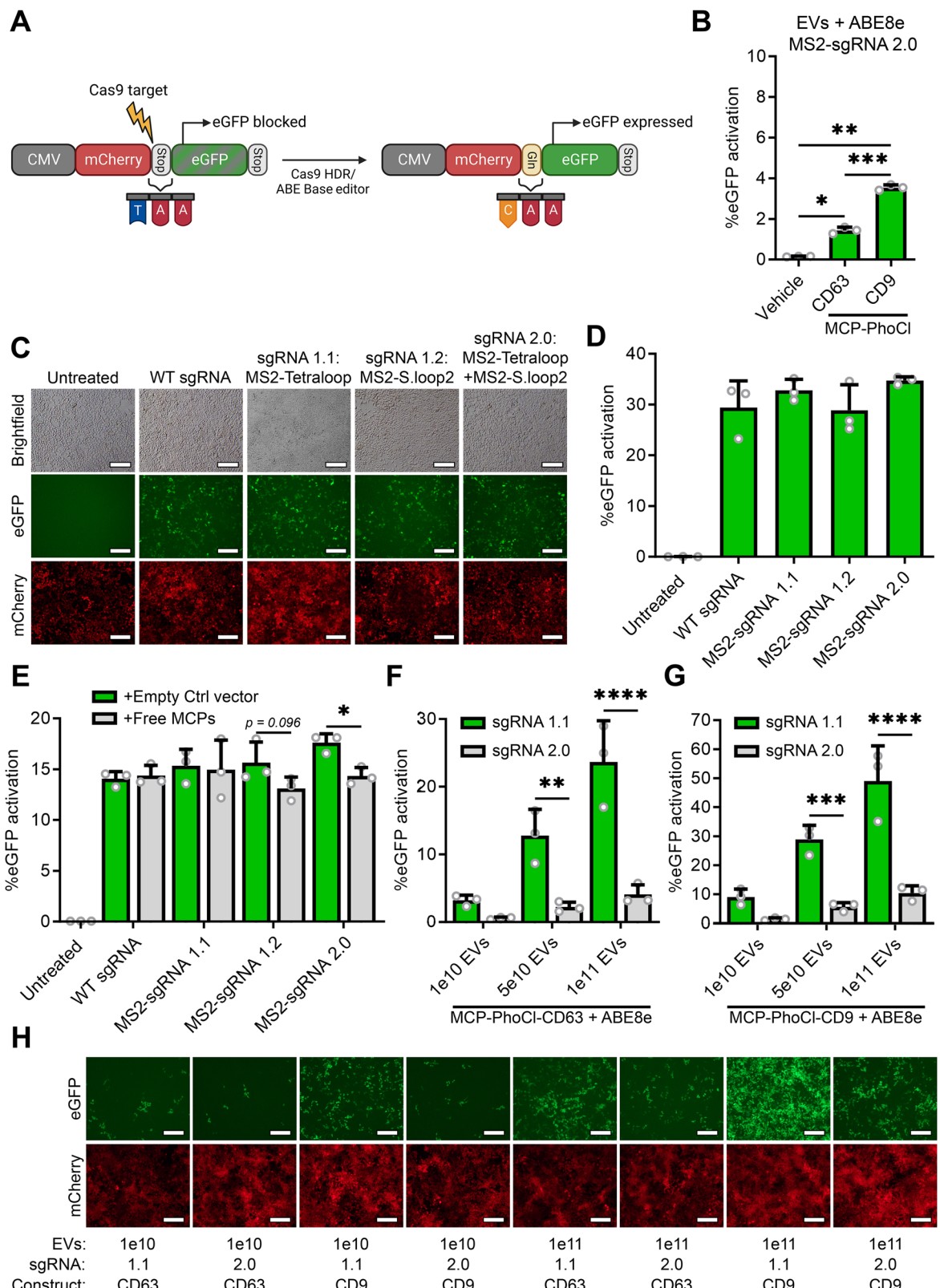

expressing free cytosolic tandem MCPs was co-transfected, a decrease in functionality was observed for MS2-sgRNA 1.2 and MS2-sgRNA 2.0, but not for WT sgRNA and sgRNA 1.1, indicating that binding of tandem MCPs to the second stemloop of the sgRNA interferes with ABE8e functionality (Fig. 5E). To test whether this indeed negatively affected our EV-mediated delivery of ABE8e using MCP-PhoCl constructs, EVs were isolated from HEK293T cells expressing VSV-G, ABE8e, MS2-

sgRNA 1.1 or MS2-sgRNA 2.0, in combination with MCP-PhoCl-CD63 or MCP-PhoCl-CD9. EVs were treated with 395 nm UV, and added to $1.0 \times 10^5$ ABE Stoplight reporter cells at multiple dosages ($1.0 \times 10^{10}$, $5.0 \times 10^{10}$, $1.0 \times 10^{11}$). As previously observed (Fig. 5B), addition of EVs with ABE8e loaded with MS2-sgRNAs with aptamers in both the tetraloop and second stemloop (MS2-sgRNA 2.0) resulted in low levels of eGFP expression (Fig. 5G, H). However, EV-mediated delivery of ABE8e

**Fig. 5 | MCP-PhoCl-based EV delivery of adenine base editor ABE8e facilitates high levels of base editing. A** Schematic of the fluorescent reporter construct for ABE activity. mCherry is followed by a stop codon and an in-frame eGFP ORF. A Cas9 target site is present at the mCherry stop codon, allowing ABE-mediated conversion of the stop codon into glutamine (Gln), resulting in the expression of a mCherry-eGFP fusion protein. Created in BioRender. Utrecht University, P. (2025) https://BioRender.com/90akg9p. **B** EV-mediated delivery of ABE8e using MCP-PhoCl-CD63 or MCP-PhoCl-CD9 with a targeting MS2-sgRNA 2.0 results in significant but limited ABE8e activity. $5.0 \times 10^{10}$ EVs per well. Means + SD, $n = 3$ independent experiments, One-way ANOVA with post-hoc Tukey's multiple comparisons test. **C, D** Transfection of plasmid DNA encoding for ABE8e with a targeting WT sgRNA (no MS2 stemloops), MS2-sgRNA 1.1 (MS2 aptamer in tetraloop), MS2-sgRNA 1.2 (MS2 aptamers in stem loop 2), or MS2-sgRNA 2.0 (MS2 aptamer in tetraloop and stemloop 2) results in similar levels of ABE8e activity, as shown by fluorescence microscopy images (**C**) and flow cytometry analysis (**D**),

48 h after transfection. Scalebar represents 200 µm. Means + SD, $n = 3$ biologically independent samples. **E** Flow cytometry analysis of HEK293T fluorescent reporter cells transfected with ABE8e and various targeting sgRNAs, alongside a plasmid for expression of cytosolic tandem MCPs, or an empty control vector. Only sgRNAs with an MS2 aptamer in the second stemloop show decreased ABE8e activity in the presence of free MCPs. Means + SD, $n = 3$ biologically independent samples, One-way ANOVA with post-hoc Sidak's multiple comparisons test. **F, G, H** Flow cytometry (**F, G**) and fluorescence microscopy analysis (**H**) for a dose response of EV-mediated ABE8e delivery using MCP-PhoCl-CD63 (**F**) and MCP-PhoCl-CD9 (**G**) comparing MS2-sgRNA 1.1 and 2.0 after addition of $1.0 \times 10^{10}$, $5.0 \times 10^{10}$, and $1.0 \times 10^{11}$ EVs. Both loading constructs show high ABE8e activity with MS2-sgRNA 1.1, in a dose-dependent manner. Scalebar represents 200 µm. Means + SD, $n = 3$ independent experiments, One-way ANOVA with post-hoc Sidak's multiple comparisons test. * $p < 0.05$, ** $p < 0.01$, *** $p < 0.001$, **** $p < 0.0001$.

using MS2-sgRNA 1.1 (tetraloop MS2 aptamer only), resulted in high levels of dose-dependent EV-mediated ABE8e delivery (Fig. 5F, G and Supplementary Fig. S10C, D). Once more, MCP-PhoCl-CD9 outperformed MCP-PhoCl-CD63 > 2-fold, in line with observations for WT Cas9 (Fig. 3G and Supplementary Fig. S8A, B) and dCas9-VPR (Fig. 4D).

These data demonstrate that MCPs bound to the second stemloop of the sgRNA inhibit EV-mediated ABE activity. To asses whether the deoxyadenosine deaminase domain also negatively affects EV loading by blocking the binding of MCPs to the MS2 hairpins, fluorescently labeled ABE8e with mStayGold was co-expressed with MCP-CD63 and sgRNAs with MS2 hairpins integrated in the tetraloop (sgRNA 1.1), second stemloop (sgRNA 1.2), both (sgRNA 2.0), or neither (sgRNA 1.0). EVs were then isolated, treated with CD63 antibodies, and captured using Protein G-coupled beads, stained with the lipid membrane dye MemGlow 640, washed, and analyzed by flow cytometry to determine the effects of the varying MS2-sgRNAs on ABE8e loading (Supplementary Fig. S12A, B). Indeed, only sgRNA 1.1 showed a significant increase in green fluorescent signal compared to EVs loaded with unlabeled ABE8e, as compared to WT sgRNA with no MS2 hairpins (Supplementary Fig. S12C). These data show that MCPs bound to the second stemloop do not only inhibit ABE activity, but the presence of the deoxyadenosine deaminase domain on Cas9 also has an inhibitory effect on MCP binding to MS2 hairpins on the second stemloop, resulting in decreased ABE RNP EV loading. Altogether, these data show that this platform is also suitable for the delivery of adenine base editors, but MS2 aptamer-MCP binding on the second stemloop of the sgRNA has a strong negative effect on ABE8e functionality, and should thus be avoided.

Finally, we investigated the suitability of MCP-PhoCl-mediated EV CRISPR-Cas9 delivery for endogenous gene editing, using T7E1 endonuclease assays. To verify the suitability of this assay, we first analyzed the stoplight reporter locus after MCP-PhoCl-CD63 VSV-G+ EV-mediated delivery of Cas9 at various dosages (Supplementary Fig. S13A). Indeed, this assay demonstrated consistent and dose-dependent gene editing (Supplementary Fig. S13B). In line with previous observations, measuring gene editing efficiencies with a T7E1 assay shows lower editing than fluorescence flow cytometry. This results from multiple genomic integrations of the reporter construct, resulting in lower relative indel numbers as compared to the percentage of cells that have undergone Cas9-mediated reporter recombination[13]. To assess gene editing in an endogenous locus, we targeted the *CCR5* gene on chromosome 3[39] using a previously verified targeting sequence[13]. Using a MS2-sgRNA 1.1 *CCR5* sgRNA (Supplementary Table S2), Cas9 (Supplementary Fig. S13C, D) and ABE8e (Supplementary Fig. S13E, F) were delivered to HEK293T cells through VSV-G+ MCP-PhoCl-CD63 and MCP-PhoCl-CD9 EVs, alongside pDNA transfection. Once more, successful gene editing was confirmed by T7E1 endonuclease analysis in the *CCR5* locus. As previously observed, MCP-PhoCl-CD9-mediated Cas9 delivery showed higher levels of gene

editing than was observed for MCP-PhoCl-CD63 for both Cas9 and ABE8e-mediated gene editing. As T7E1 assays have decreased efficiency on single-nucleotide substitutions, which are often generated with adenine base editors, we performed a BsaHI restriction enzyme-mediated analysis of the *CCR5* locus. Alongside increased sensitivity, this read-out is suitable to confirm specific adenine base editor activity of this specific target *CCR5* locus, as is shown in Supplementary Fig. S13G. In concordance with previous results, both EV-mediated ABE8e delivery and pDNA transfection resulted in high levels of BsaHI-mediated cleavage of PCR products of the targeted *CCR5* locus, confirming specific ABE8e-mediated base editing.

Altogether, these data show that MCP-PhoCl-mediated EV loading and release constructs provide a modular, versatile EV-mediated Cas9 delivery platform that allows for the efficient delivery of various Cas9-based proteins with high therapeutic potential, including wildtype Cas9, transcriptional activators, and adenine base editors.

## Discussion

Already in the first report where the CRISPR-Cas9 system was harnassed for targeted genomic engineering of specific sequences in prokaryotic cells by Jinek et al.[2], the authors envisioned the potential of a "methodology based on RNA-programmed Cas9 that could offer considerable potential for gene-targeting and genome-editing applications". Promising current clinical trials for direct in vivo Cas9 gene editing strategies are based on lipid nanoparticle (LNP)-mediated Cas9 delivery for the treatment of hereditary angioedema and for transthyretin amyloidosis, by disrupting liver expression of KLKB1 or TTR, respectively[40,41]. Whereas these LNP-based strategies have shown clinical promise for in vivo delivery of Cas9, their restricted biodistribution profile initially limited their application to gene editing strategies specifically targeting the liver. Recent studies have made great strides in improving tunability and control over LNP biodistribution, showing that LNP tropism can be shifted towards bone-marrow, lungs or spleen[17,18]. However, LNPs have been reported to show cellular toxicity and limited delivery efficiency due to endosomal entrapment[16]. Furthermore, recent studies have shown that repeated injections with LNPs can lead to antibody-mediated immune reactions against the polyethylene glycol (PEG) groups present on the exterior of LNPs, which may result in accelerated blood clearance and a decrease in effectivity[42]. Thus, there is a pressing need for the development of novel delivery strategies with tunable biodistribution profiles, low toxicity, and low immunogenicity for the in vivo delivery of CRISPR-Cas9.

In this study, we describe a modular strategy for extracellular vesicle (EV)-mediated delivery of CRISPR-Cas9 by employing an aptamer-based strategy for loading of Cas9 RNPs into EVs. This approach is combined with the incorporation of a photocleavable domain (PhoCl) to facilitate intraluminal cargo release from the EV membrane after stimulation with 395 nm UV light. We chose to study

the suitability of EVs for intracellular Cas9 RNP delivery due to their natural capacity to transfer proteins, RNA, and various other biological cargos as part of their role in intercellular communication[43]. Alongside their intrinsic capacity to transfer biological cargos, EVs are promising delivery vectors due to their reported low immunogenicity and their capacity to pass difficult-to-cross biological barriers such as the blood-brain-barrier[44,45]. To facilitate active Cas9 RNP loading into EVs, we made use of the capacity of MS2 coat proteins (MCPs) to strongly bind to MS2 RNA aptamers[28]. This strategy has previously been reported to facilitate mRNA loading into EVs in an approach called Targeted and Modular EV Loading (TAMEL) by Hung et al.[46]. Here, a 40-fold increase in mRNA loading into EVs was observed when directly fusing single MCP domains directly to VSV-G. Whereas substantial loading was observed in this study, no translation of mRNA cargo was observed in recipient cells. It should be noted that the MCP used in their study contained the V29I mutation, which has been reported to substantially increase MCP-MS2 aptamer affinity[47]. As such, a limiting factor in their delivery strategy may have been the retention of their mRNA cargo to the EV membrane, preventing release into the cytosol of the recipient cell. Here, we applied this strategy to load Cas9 RNPs, based on the incorporation of MS2 aptamers into the tetraloop and second stem-loop of the sgRNA[30]. Despite the exclusion of the V29I mutation in the MCP, we still saw limited functional RNP delivery initially. Only after including an activatable release strategy by incorporating the PhoCl photocleavable domain[29] did we observe a substantial increase in RNP delivery, confirming that cargo retention to the EV membrane was indeed hampering functional delivery.

In recent years, various additional strategies have been described to release cargos from the EV membrane to increase cargo delivery. Another light-based cargo loading and release strategy is the "exosomes for protein loading via optically reversible protein–protein interactions" (EXPLORs) platform[48]. This system is based on the incorporation of photoreceptor cryptochrome 2 (CRY2) on the N-terminus of CD9, and a truncated CRY-interacting basic-helix-loop-helix 1 protein module (CIBN) on the cargo protein. Whereas this approach has shown promising efficiency in cargo delivery, one limitation of this system is the requirement of constant blue light exposure throughout cell culture conditions, prior to EV isolation. Another reported strategy is the FK506 binding protein (FKBP) and the FKBP-rapamycin-binding domain (FRB) dimerization system[49,50]. Both these moieties bind to rapamycin analogs, and by fusing FKBP to an EV-enriched protein, and FRB to the cargo protein, cargo loading can be facilitated by the addition of rapamycin analogs to cell culture conditions. A limitation of this system is the requirement of the addition of rapamycin-analogs to cell culture conditions to facilitate cargo loading, resulting in increased costs of EV production. As compared to these strategies, an advantage of the PhoCl-based release strategy presented in this manuscript is the lack of any additional conditional requirements during cell culture and EV production, as the photocleavable domain is intact until stimulated with UV light. One recently published strategy that has a similar advantage is the incorporation of self-cleaving inteins[51]. These protein domains can either self-excise themselves from protein sequences or, after specific amino acid substitutions, facilitate protein cleavage, and may be activated by a variety of conditions, including changes in pH, temperature, or naturally over time[52]. Employing pH-dependent inteins as described by Liang et al. is an elegant strategy, as it facilitates cargo release in late endosomal compartments in recipient cells[51]. For further examples, we recommend a recent study from Osteikoetxea et al., which shows a thorough comparison of these systems, alongside various additional loading and release strategies for EV-mediated Cas9 delivery[53].

Our PhoCl-mediated approach also faces certain limitations, as UV-mediated cleavage occurs at a fairly long half-time (~ 500 s) and dissociation efficiency is limited to ~ 71%[29]. However, recent work from Lu et al. has demonstrated novel PhoCl derivatives that either show increased cleavage rates (PhoCl2f, 76 s half-time) and increased efficiency (PhoCl2c, 92% dissociation efficiency)[54], which may further improve the efficiency and ease-of-use of the delivery platform described in this study. Another point of initial concern was the potential damage to EV components or its Cas9 RNP cargo resulting from UV treatment. However, it should be noted that UV wavelengths associated with nucleotide damage are below 340 nm[35]. Concordantly, we also did not observe any decrease in Cas9 RNP delivery after applying 395 nm UV treatment to EVs that did not contain the PhoCl domain, indicating that the Cas9 RNP cargo indeed remained intact. Further analyses also revealed no noticeable effects on EV integrity, protein cargo, sgRNA levels, or downstream of target gene editing events. However, we cannot completely exclude the potential effects of our UV treatment on EV integrity or functionality. Moreover, despite the promising results achieved with these UV-activatable photocleavable domains, they do present certain limitations for in vivo applicability. In the methods employed in this manuscript, EVs are treated with UV for intraluminal release of the cargo, prior to their addition to cells. Such protocols could, theoretically, also be applied for in vivo cargo delivery: UV-treatment could either be applied prior to administration, or directly after isolation of EVs. However, UV-cleavable domains might not be suited for in vivo delivery, where these photocleavable domains are activated at a specific site of interest for local protein cleavage, as UV wavelengths show limited tissue penetration and potential skin toxicity[55]. For such approaches, photo-releasable strategies that are activated using longer wavelengths would be preferable. Whereas direct photocleavable proteins in long wavelength ranges have not been described, light-switchable dimerization phytochromes that are associated under treatment of 600 nm light, and actively disassociated by 780 nm light, such as MagRed, have been described[56]. However, as of yet, these dimerizing phytochromes also show spontaneous disassociation over time in dark conditions. Thus, further engineering to shift the activation wavelength of photocleavable domains or to inhibit self-disassociation of long-wavelength activatable phytochromes would be required for the future development of such in situ activatable release strategies.

Despite thorough documentation of EVs inately mediating intracellular transfer of biological cargos, including protein and RNA, we did not observe efficient EV-mediated Cas9 delivery without the incorporation of the Vesicular stomatitis virus G (VSV-G) fusogenic glycoprotein under the conditions described in this manuscript. VSV-G is commonly incorporated into biological particles such as lentiviral particles to aid in cargo delivery by facilitating endosomal escape[57]. Indeed, confocal analysis of recipient cells treated with Cas9-loaded EVs showed a significant increase in fluorescently labeled Cas9 localization to the nucleus in recipient cells, as opposed to EVs without VSV-G that showed no increase of Cas9 localization to the nucleus over time (Supplementary Fig. S9). Whereas incorporation of VSV-G facilitated high levels of Cas9 delivery, there are concerns regarding the immunogenicity of VSV-G. For example, a study by S. Kuate et al. demonstrated a 100-fold higher antibody titer response against structural proteins of viral like particles (VLPs) when VSV-G was co-expressed[58]. Thus, future incorporation of less immunogenic or endogenous (human) fusogenic proteins, such as members of the syncytin family, might be preferable in the design of potential therapeutic EV-based delivery strategies[59]. Moreover, despite multiple reports in preclinical and early-stage clinical trials showing a lack of immunotoxicity, various parameters should be taken into consideration during EV production such as cell source, isolation and storage methods, as these may have a significant impact on potential EV immunogenicity[60,61]. However, it should be noted that toxicity and immunogenicity studies revealed minimal effects in mice after sustained dosing of HEK293T-derived EVs[44].

For active cargo loading, our current approach relied on the MCP-MS2 aptamer interaction. Whereas this system is highly efficient and

well-designed it did present some challenges, most notably in terms of cargo release. Whereas this was, in part, alleviated by the incorporation of a photocleavable domain, the tandem MCP complex is still bound to the sgRNA MS2 aptamers on the RNP complex. In line with previous reports, this did not compromise the functionality of Cas9[30]. We did observe a decrease in dCas9-VPR-mediated transcriptional activation in MS2-sgRNAs as compared to unmodified sgRNAs. Furthermore, we did observe interference with the functionality of adenine base editor ABE8e using sgRNAs with MS2 hairpins in the second stemloop (Fig. 5). Whereas this issue can be addressed by only incorporating MS2 aptamers into the tetraloop of the sgRNA, it does reveal a limitation in the robustness of the modular versatility of this approach. A potential solution may be found in the utilization of different aptamer-based loading systems that rely on smaller RNA-binding proteins, such as the COM aptamer system, or systems that show differences in release kinetics, such as the recently reported optimized version of the designer Pumilio and FBF homology domain, termed PUFe[62,63]. Furthermore, it should be taken into consideration that the increase in size of Cas9 moieties containing additional enzymatic groups such as base editors, transcriptional activators and prime editors may also effect EV loading efficiency, as it has been reported that increased size of Cas9 due to protein conjugation may result in decreased EV loading[64].

Lastly, we also observed substantial differences in Cas9 RNP delivery based on the recipient cell type, and the choice of EV-enriched protein used for targeted loading. Whereas primary tumor cell lines showed lower gene editing rates than the easy-to-transfect HEK293T cells, differences were still within an order of magnitude (Supplementary Fig. S4). Moreover, as all tested cell types showed significant gene editing activity, these results suggest potential applicability on a broad range of cells. However, these data also indicate that recipient cell-specific optimization for gene editing may be necessary. Interestingly, whereas CD9 significantly outperformed CD63 in terms of both cargo loading and delivery in our study, a recent study from Zheng et al. showed substantially higher loading when using CD63 as compared to CD9[26]. One potential explanation for this observed difference might be that whereas our cargo loading strategy was based on N-terminal fusion to the EV-enriched proteins, Zheng et al. opted for C-terminal fusion, given that N-termini are often a site for signal peptides that affect intracellular trafficking. Furthermore, efficiency of our delivery strategy may be further improved by incorporating recently uncovered moieties that are highly enriched in EVs, including TSPAN2, PTGFRN, a MysPalm tag, or direct fusion to the VSV-G glycoprotein[26,27,46,53].

In closing, we describe a modular, versatile and efficient engineering strategy for EV-based delivery of Cas9, based on the combined utilization of aptamer-based RNP loading and UV-activated cargo release. This strategy is suitable for the dose-dependent delivery of various Cas9 variants, facilitating NHEJ-based gene editing, transcriptional activation, and adenine base editing. Given the modular design of this approach, there is strong potential for additional future applications, such as the delivery of mRNA, CRISPR-mediated transcriptional inhibitors (CRISPRi), and additional gene editing approaches such as prime- and cytidine base editing. Altogether, this work further broadens the utility of EVs for (bio)therapeutic delivery strategies and underlines the potential for EV-mediated treatment of genetic diseases.

## Methods

### Cell culture
HEK293T cells (CRL-3216), MCF-7 cells (HTB-22), and MDA-MB-231 cells (HTB-26) were obtained from the American Type Culture Collection (ATCC) and were cultured in Dulbecco's Modified Eagle Medium (DMEM) with L-Glutamine (Gibco) supplemented with 10% fetal bovine serum (FBS) (Sigma-Aldrich). T47D cells (85102201) were obtained from Sigma-Aldrich, and were cultured in DMEM/F12 with L-Glutamine supplemented with 10% FBS. HEK293T stoplight cells, MDA-MB231 stoplight cells, MCF-7 stoplight cells, T47D stoplight cells, and HEK293T adenine base editor stoplight cells were generated as previously described[13,31]. HEK293T cells with stable doxycycline-inducible eGFP expression and HEK2939T cells with stable H2B-mTag-BFP2 expression were generated for this study as described below. All cell lines were cultured at 37 °C and 5% $CO_2$.

### DNA constructs
For unmodified sgRNA expression, targeting sequences were cloned into a lentiGuide-Puro plasmid (Addgene #52963)[65] as previously described[31]. In short, complementary synthesized oligonucleotides (Integrated DNA technologies) were annealed and ligated into a lentiGuide-Puro plasmid after BsmBI digestion (New England Biolabs). For the expression of MS2-sgRNAs, complementary synthesized oligonucleotides were annealed and cloned into the lenti sgRNA(MS2)-Zeo backbone (Addgene #61427)[30] after BsmBI digestion. For expression of sgRNAs with a single MS2 hairpin integration, sgRNA sequences were synthesized as gBlocks (Integrated DNA technologies), and cloned into a Lenti_gRNA-Puro backbone (Addgene #84752)[66] after BsmBI digestion. Oligonucleotide and gBlock sequences used for cloning sgRNA expression constructs are listed in Supplementary Table 1, expressed sgRNA sequences are listed in Supplementary Table 2. For the generation of a lentiviral doxycycline-inducible eGFP expression construct, an eGFP open reading frame was transferred from pDONR221-eGFP (Addgene #25899)[67] into pInducer20 (Addgene #44012)[68] using the Gateway LR Clonase II Enzyme Mix (Thermo Fisher Scientific) according to the manufacturer's protocol. For expression of MCP loading constructs, including those with photocleavable PhoCl domains, and for H2B-mTag-BFP2 expression, human codon-optimized sequences were synthesized as gBlocks, and cloned into pHAGE2-EF1a-Multiple Cloning Site-IRES-PuroR plasmids using NotI and BamHI restriction enzymes (New England Biolabs), as described previously[69]. In short, both vector plasmids and gBlocks were restricted with NotI and BamHI according to the manufacturer's instructions, followed by 1% TAE agarose electrophoresis and gel extraction. Samples were ligated using T4 Ligase (Thermo Fisher Scientific), followed by transformation into Stbl3 *E. coli* (New England Biolabs). For fluorescent labeling of Cas9 and ABE8e with mStayGold[37], gBlocks with GS-linkers and compatible restriction sites were synthesized as gBlocks, and cloned into pLentiCas9-eGFP (Addgene #78546) using BamHI or pCMV-ABE8e (Addgene #138489) using EcoRI and AgeI, respectively. Amino acid sequences for all RNA-binding constructs and fluorescently labeled proteins are shown in Supplementary Table 3. An overview of the used plasmid DNA constructs is listed in Supplementary Table 4.

### Lentiviral production and generation of stable cell lines
For lentiviral production HEK293T cells were plated at 50% confluency in DMEM supplemented with 10% FBS and supplemented with 1x Antibiotic Antimycotic Solution (Sigma-Aldrich), and transfected with lentiviral transfer plasmids containing genes of interest (GOI), PSPAX2 (Addgene #12260) and pMD2.G (Addgene #12259) at a 2:1:1 ratio using 3 μg 25 kDa linear polyethylenimine (PEI) (Polysciences Inc.) per μg DNA. After 18 h culture medium was replaced with fresh DMEM supplemented with 10% FBS and 1 x Antibiotic Antimycotic Solution. After 48 hours, lentiviral supernatants were harvested. Cells were removed by a 10 min centrifugation at 500 x $g$, followed by 0.45 μm syringe-filtration (Sartorius). Cells were transduced overnight using lentiviral stocks supplemented with 8 μg/ml polybrene (Sigma-Aldrich), after which lentiviral supernatant was replaced by fresh culture medium. 24 h after lentiviral transduction, cells were cultured with their respective selection antibiotics. All reporter cell lines were cultured in the presence of 1000 μg/ml G418 (Invivogen). Stable donor cell lines

expressing MS2-sgRNA were cultured in the presence of 200 µg/ml Zeocin (Invivogen), and stable donor cell lines expressing MCP-CD63 were cultured in the presence of 2 µg/ml puromycin (Invivogen). HEK293T reporter cell lines transduced to express H2B-mTag-BFP2 were cultured in the presence of 2 µg/ml puromycin and subjected to 2 rounds of fluorescence-activated cell sorting for positive BFP signal using a BD FACSAria Fusion Cell Sorter (BD Biosciences).

## Plasmid transfection

For transfection of plasmid DNA for expression of EV cargos, HEK293T cells were plated at $1.0 \times 10^7$ cells per T175 flasks in culture medium supplemented with 1x Antibiotic Antimycotic Solution. After 24 h, HEK293T cells were transfected with 5 µg for each plasmid per flask. All plasmids were mixed in a 50 ml conical tube in a volume of 1 ml OptiMEM per transfected flask, and simultaneously, 2 µg 25 kDa linear PEI per µg plasmid DNA was mixed in a separate conical tube in a volume of 1 ml OptiMEM per transfected flask. After a 5 min incubation at room temperature, the contents of both tubes were transferred to a single conical tube, and gently mixed by inverting the tube several times. After a 15 min incubation at room temperature, 2 ml transfection mixture was added to each T175 flask, followed by an overnight transfection at 37 °C and 5% $CO_2$. Then, the transfection medium was removed, and cells were gently washed using 5 ml OptiMEM per flask. Hereafter, 20 ml OptiMEM supplemented with 1x Antibiotic Antimycotic Solution was added per flask. After 24 h, conditioned medium was isolated for subsequent EV isolation as described below.

For direct transfection of HEK293T cells to study reporter cell activation, cells were plated in 24-well plate wells at a density of $5.0 \times 10^4$ cells per well in 1 ml culture medium supplemented with 1x Antibiotic Antimycotic Solution, 24 h priors to transfection. HEK293T reporter cells were transfected with 250 ng per plasmid per well, unless stated otherwise. All plasmids were mixed in a 1.5 ml tube in a volume of 50 µl OptiMEM per transfected well, and simultaneously, 3 µg 25 kDa linear PEI per µg plasmid DNA was mixed in a separate 1.5 ml tube in a volume of 50 µl OptiMEM per transfected well. After a 5 min incubation at room temperature, both solutions were transferred to a single 1.5 ml tube and gently mixed. After a 15 min incubation at room temperature, 100 µl transfection mixture was added to each well. Reporter cells were then analyzed by flow cytometry and fluorescence microscopy as described below. Reporter cells were analyzed at least 48 hours after transfection, to ensure sufficient levels of eGFP expression.

## Extracellular vesicle isolation

HEK293T cells were plated in T175 culture flasks for EV isolation in DMEM supplemented with 10% FBS. HEK293T were plated to be at 80% confluency 24 hours prior to EV isolation, as described above. Cell culture medium was removed, and the cells were gently washed with 5 ml OptiMEM per flask. Hereafter, 20 ml OptiMEM supplemented with 1x Antibiotic Antimycotic Solution was added per flask. After 24 h, conditioned medium was isolated for extracellular vesicle isolation. First, cells and cellular debris were removed by a 5 minute centrifugation step at $300 \times g$, followed by a 15 min centrifugation step at $2000 \times g$. Supernatant was then collected and filtered through a 0.45 µm vacuum filter (Corning). The filtered supernatant was then concentrated by tangential flow filtration (TFF), using a Vivaflow 50 R 100 kDa TFF cassette (Sartorius) to a volume of 15 ml, and subsequently concentrated to a 0.5 – 1.0 ml volume using 100 kDa Amicon Ultra-15 Centrifugal filters (Merck). EVs were then isolated by size exclusion chromatography (SEC) on an Akta Pure chromatography system using a Tricorn 10/300 column with Sepharose 4 Fast Flow resin (all GE Healthcare Life Sciences). EVs were then sterilized by 0.45 µm syringe-filtration, and concentrated to a 300 – 400 µl volume using 100 kDa Amicon Ultra-15 Centrifugal filters (Merck).

## Nanoparticle tracking analysis

EV particle concentration and size distribution was determined using a Nanosight S500 nanoparticle analyzer, equipped with a 405 nm laser (Malvern Instruments). Samples were diluted in PBS (Sigma-Aldrich) to a suitable concentration for nanoparticle tracking analysis (between $2.0 \times 10^8$ and $2.0 \times 10^9$ particles per ml) and were analyzed using 5 x 30 s recordings per sample with a camera sensitivity setting at level 16. All acquisition settings were set to "Auto" for post-acquisition analysis, with the exception of a fixed detection threshold, which was set to 7. Recordings were analyzed using NTA software v3.4. Particle counts were corrected for measurements of PBS, followed by correction for dilution.

## Western blot

Cells or isolated EVs were lysed in RIPA buffer, supplemented with a protease inhibitor cocktail (Sigma-Aldrich). Cell lysates were incubated on ice for 30 minutes, followed by a 15 min 4 °C centrifugation step at $12,000 \times g$ to remove non-soluble materials. Protein concentrations were measured using a BCA Protein Assay Kit (Thermo Fisher Scientific), alongside a BSA protein standard (Thermo Fisher Scientific), according to the manufacturer's protocol. Samples were normalized for protein concentration and mixed with premixed sample loading buffer (Bio-Rad). With the exception of CD63 western blot analysis, 100 µM DTT reducing agent was included in the sample loading buffer. Prior to western blot analysis, samples were denatured by a 10 min incubation at 95 °C. Then, samples were loaded on 4–12% gradient Bis–Tris polyacrylamide gels in 1x MOPS buffer (Thermo Fisher Scientific), and subjected to electrophoresis. Hereafter, proteins were blotted onto Immobilon-FL PVDF membranes (Millipore), which was subsequently blocked for 1 hour at room temperature using a blocking buffer consisting of 1 part Intercept Blocking Buffer (LI-COR Biosciences) and 1 part Tris-Buffered Saline (TBS). Membranes were then probed overnight at 4 °C in staining buffer consisting of 1 part Intercept Blocking Buffer (LI-COR Biosciences) and 1 part Tris-Buffered Saline with 0.1% Tween-20 (TBS-T), using the following primary antibodies: ALIX 1:1000 (Thermo Fisher Scientific, MA1-83977), Calnexin 1:1000 (GeneTex, GTX101676), CD63 1:1000 (AB8219), Syntenin 1:500 (Origene, OTI2H6), TSG101 1:1000 (Abcam, ab30871), and H2B 1:1000 (Abcam, ab52599), GAPDH 1:500 (Abcam, AB9485) or α-Flag 1:1000 (Sigma-Aldrich, F1804). Blots were washed 5 times for 5 min at room temperature in TBS-T, followed by a 2 h probe with secondary antibodies in staining solution at room temperature, using the following secondary antibodies at a 1:10,000 dilution: anti-rabbit IgG conjugated to AlexaFluor 680 (Thermo Fisher Scientific, A-21076), anti-mouse IgG conjugated to AlexaFluor 680 (Thermo Fisher Scientic, A-21057), anti-rabbit IgG conjugated to IRDye 800CW (926-32211, LI-COR Biosciences), or anti-mouse IgG conjugated to IRDye 800CW (926-32212, LI-COR Biosciences). Next, blots were washed 3 x 5 min at room temperature in TBS-T, and 2 x 5 min washing steps at room temperature in TBS. Total protein stain was performed using No-Stain Protein Labeling Reagent (ThermoFisher Scientific), according to the manufacturer's protocol. Fluorescent imaging was done using an Odyssey Infrared Imager (LI-COR Biosciences) at 700 and 800 nm.

## Transmission electron microscopy

EVs were isolated as described above, and were subsequently adsorbed to carbon-coated coated formvar grids (TAAB Laboratories Equipment Ltd.) at room temperature for 15 min. Formvar grids were washed with PBS to remove any remaining unbound EVs, and grids were subsequently fixed using a fixing buffer (2% paraformaldehyde, 0.2% glutaraldehyde in PBS) for 30 min at room temperature. After counterstaining with uranyl-oxalate, grids were embedded an 1.8% methyl cellulose and 0.4% uranyl acetate mixture at 4 °C. Grids were imaged on a Jeol JEM-1011 microscope (Jeol), or a Tecnai T12 transmission electron microscope (FEI).

## Optiprep density gradient

After EVs were isolated as described above, EV SEC fractions were transferred to SW40 centrifuge tubes (Beckman Colter), PBS was added to a volume of 14 ml, and EVs were pelleted by ultra-centrifugation for 60 minutes at 100,000 x $g$ at 4 ° C in an SW40.Ti rotor (Beckman Colter). After centrifugation, the EV pellet was resuspended in 3 ml 40% Optiprep (Sigma-Aldrich) in PBS. Hereafter, 3 ml fractions of 30%, 20%, and 10% Optiprep were gently layered on top, and centrifuged for 16 hours at 4 °C at 200,000 x $g$ in an SW40.Ti rotor. 1 ml fractions were then gently isolated from the top of the Optiprep gradient and transferred to 2 ml tubes. Next, proteins were precipitated for each of the isolated fractions by trichloroacetic acid (TCA) precipitation. 250 μl TCA was added to each 1 ml fraction and mixed thoroughly. After an overnight precipitation at − 20 °C, proteins were pelleted by centrifugation at 10,000 x $g$ for 10 min at 4 °C. After supernatant removal, pellets were washed 3 times with 500 μl ice-cold acetone, followed by centrifugation for 10 minutes at 10,000 x $g$ at 4 °C. After the final washing step supernatant was removed, and pellets were dried at 95 °C for 10 min. Samples were then resuspended in sample buffer and analyzed by western blot analysis as described above.

## qPCR and ddPCR analysis

For qPCR and ddPCR analysis, RNA was isolated from $1.0 \times 10^{11}$ EVs per sample using TRIzol LS reagent (Invivogen) according to the manufacturer's protocol. 0.2 fmol RNA Spike-in was added to TRIzol LS samples prior to RNA isolation to correct for RNA isolation efficiency. To prevent any potential false positive signal that may arrive from co-isolation of DNA fragments from transient transfection, samples were subjected to DNase treatment using the TURBO DNA-free Kit (ThermoFisher Scientific). Reverse transcription was done using the SuperScript IV cDNA synthesis kit (ThermoFisher Scientific). For qPCR analysis, DNA was diluted 5 times in Milli-Q water, whereafter 5 μl was used for a 25 μl qPCR reaction using 2x SYBR Green Master Mix (Bio-Rad) in a CFX96 Real-Time PCR Detection System (Bio-Rad) according to the manufacturer's protocol. Cycle threshold (Ct) values were corrected for the Spike-in RNA. For ddPCR analysis, cDNA was diluted 10 times in Milli-Q water and denatured for 8 min at 90 °C. 1 μl of diluted cDNA was added to 21 μl of ddPCR Supermix for probes (no dUTP) (Bio-Rad) with 0.2 μM primers and 0.05 μM fluorescent, self-quenching target probes (IDT) directed against sgRNA (FAM) and Spike (HEX). Droplets were made in an Automated Droplet Generator (Bio-Rad) according to the manufacturer's instructions, after which the plate was sealed with a foil heat seal (Bio-Rad). PCR amplification was done as follows: 10 min at 95 °C, followed by 40 amplification cycles of 30 sec at 94 °C and 1 min at 60 °C, deactivation for 10 min at 98 °C and hold at 12 °C. Next, the plate was transferred to a QX200 Droplet Reader (Bio-Rad) to determine the number of fluorescent droplets for FAM (MS2-sgRNA) and HEX (Spike-in). Data were analyzed in Quanta-Soft (v. 1.7.4.0917), negative reverse transcriptase controls were used to define a detection threshold for sgRNA and Spike detection. PCR primers and ddPCR fluorescent probes were synthesized by Integrated DNA Technologies. Spike-in sequence and ddPCR probe sequences are listed in Supplementary Table 1, PCR primers are listed in Supplementary Table 5.

## Extracellular vesicle addition experiments

To study EV-mediated Cas9 RNP delivery, EVs were added to cells in 24-well plate wells, cultured in 1 ml culture medium. Unless stated otherwise, cells subjected to EV addition experiments were cultured in 24-well plate wells at a density of $1.0 \times 10^5$ cells per well in 1 ml culture medium at the time of EV addition. EV dosages were normalized based on nanoparticle tracking analysis, as described above. Prior to EV addition, EVs containing loading constructs with PhoCl domains were treated with UV light on ice for 20 min, using a custom 50W 395 nm

LED panel (SJLA). Reporter cells were analyzed 72 h after EV addition to ensure sufficient levels of eGFP expression, unless stated otherwise. Reporter cells were analyzed by flow cytometry and fluorescence microscopy.

## Fluorescence microscopy

Prior to flow cytometry analysis, reporter cells were analyzed by fluorescence microscopy using either an EVOS FL Cell Imaging System (Thermo Fisher Scientific), or a Nikon Eclipse TS2 Inverted Fluorescence Microscope (Nikon Instruments). Images were processed using ImageJ v1.54d software.

## Flow cytometry

After fluorescence microscopy, cells were gently washed with 0.5 ml PBS and trypsinized for 5 min using 250 μl TrypLE Express (Thermo Fisher Scientific). Cells were then transferred to 1.5 ml tubes using a similar volume of cell culture medium containing 10% FBS. Cells were centrifuged for 5 minutes at 300 x $g$, washed in 1 ml 1% FBS in PBS, and centrifuged for 5 minutes at 300 x $g$. After supernatant removal, cells were resuspended in 200 μl 1% FBS in PBS, transferred to 96-well U-bottom Falcon plates (Fisher Scientific), and analyzed on a BD FACSCanto II flow cytometry system (BD Biosciences), a CytoFLEX LX Flow Cytometer (Beckmann Colter), or a BD LSRFortessa Cell Analyzer (BD Biosciences). Flow cytometry results were analyzed using FlowJo v10 software. Gating strategies for all reporter cell lines are shown in Supplementary Fig. S1.

## Confocal microscopy analysis of cellular Cas9 localization

To study cellular localization of Cas9 after EV-mediated delivery, Cas9 fluorescently labeled with the highly stable green fluorescent protein mStayGold[37] was co-expressed with MS2-sgRNAs and loaded into EVs using the MCP-PhoCl-CD9 construct, either with- or without VSV-G co-expression (VSV-G⁺). EVs were isolated and treated with UV as described above, and added to HEK293T cells stably expressing Histone 2B fluorescently labeled with mTag-BFP2 (H2B-mTag-BFP2), presenting a high blue fluorescent signal in their nuclei, as follows: 24 hours prior to EV addition, $1.5 \times 10^4$ H2B-mTag-BFP2 HEK293T cells were seeded per well in CellStar 96-well cell culture black μClear bottom TC-treated microplates (Greiner-Bio) in 150 μl culture medium. EVs were isolated and quantified using NTA analysis as described above. To assure equal levels of fluorescent cargo delivery for both conditions, green fluorescent signal was determined by measuring fluorescence of EV samples diluted 30 times in PBS using a SpectraMax iD3 multi-mode microplate reader at 488 nm excitation and 512 nm emission wavelengths (Molecular Devices). Dosages were normalized to fluorescence levels of $1.5 \times 10^{11}$ Cas9-mStayGold RNP loaded MCP-PhoCl-CD9 VSV-G⁺ EVs after UV treatment. EVs were added directly to the H2B-mTag-BFP2 HEK293T cells in cell culture medium, and confocal pictures were taken once per hour over the course of 12 h using a Yokogawa C7000 confocal microscope with a live cell stage incubator at 37 °C and 5% $CO_2$ using a 60x magnification water lens. Per well, 4 images distributed throughout the well were imaged using the following settings: mTag-BFP2: emission = 405 nm, power = 30%, acquisition = BP445/45, exposure time = 200 ms. mStayGold: emission = 488 nm, power = 50%, acquisition = BP525/50, exposure time = 1000 ms. Images were taken at a shifting distance of − 3.0 μm, with a slicing interval of 1.0 μm over a range of + 4.0 μm ascending distance and − 4.0 μm descending distance relative to the shifting distance, at 1 x 1 binning settings. Images were then analyzed using the Columbus Image Data Storage and Analysis System v2.7.1 (Perkin-Elmer), using the following settings: [Find Image Region]: Channel BP445/45, Common threshold = 0.35, output population name: "Nuclei". [Calculate Intensity Properties]: Channel BP525/50, population: "Nuclei", Output properties: "Intensity Region

BP525/50". Cas9-mStayGold signal in the nuclei was then calculated by averaging the "Intensity Region BP525/50" signal from all 4 images per well, and corrected for the average intensity signal of untreated cells by subtracting the average "Intensity Region BP525/50" signal of untreated H2B-mTag-BFP2 HEK293T cells.

## EV bead pulldown for flow cytometry analysis

To determine ABE8e loading in EVs, ABE8e fluorescently labeled with mStayGold was loaded into EVs by co-expression of MCP-CD63 and MS2-sgRNA as described above. EVs were quantified using NTA analysis, and diluted to a concentration of $2.0 \times 10^9$ EVs per µl in PBS. $1.0 \times 10^{10}$ EVs were incubated with 0.3 µg CD63 antibody (ab8219, Abcam) in PBS in a 20 µl volume for 30 minutes at room temperature. 0.3 µg Protein G Dynabeads (ThermoFisher Scientific) per sample were washed twice with 0.001% Tween-20 in PBS, washed once in PBS, and resuspended in 50 µl PBS. Antibody-stained EVs were then added to the washed Protein G Dynabeads, and incubated overnight on a rotator at 4 °C. Beads were washed 5 times with 0.001% Tween-20 in PBS, resuspended and aliquoted in $3 \times 10$ µl PBS per sample. Beads were then stained by the addition of 50 µl 1:100 MemGlow 640 in PBS, followed by a 15 min incubation at room temperature. Beads were washed 2 times in 1% BSA in PBS, and once more in 0.001% Tween-20 in PBS. Finally, beads were resuspended in 200 µl 0.001% Tween-20 in PBS and analyzed by flow cytometry on a BD FACSCanto II flow cytometry system. The gating and analysis strategy is shown in Supplementary Fig. S12B.

## EV bead pulldown for western blot analysis

Flag-tag labeled Cas9 was loaded into EVs by co-expression of MCP-CD63 and MS2-sgRNA as described above. EVs were quantified using NTA analysis, and diluted to a concentration of $2.3 \times 10^9$ EVs per µl in PBS. $1.0 \times 10^{11}$ EVs were incubated with 0.3 µg CD63 antibody (ab8219, Abcam), 0.03 µg CD9 antibody (CBL162, Merck Chemicals), or 0.03 µg IgG control (PA5-33236, ThermoFisher Scientific) in PBS in a 43 µl volume for 30 minutes at room temperature, as previously described[70]. 50 µl untreated EV stock was lysed by adding 5.6 µl 10x RIPA buffer, and stored at −20 °C for future analysis. Protein G Dynabeads were washed as described above, resuspended in 17 µl PBS, and incubated with the antibody-stained EVs overnight on a rotator at 4 °C. The next day, beads were captured using a DynaMag-5 magnet (ThermoFisher Scientific), and 50 µl supernatant was isolated and incubated on ice for 30 minutes after the addition of 5.6 µl 10x RIPA buffer. Beads were then washed 5 times with 0.001% Tween-20 in PBS, resuspended in 56 µl 1x RIPA buffer, and incubated on ice for 30 min. All lysates were stored at −20 °C until further analysis, where lysates were mixed with 19 µl 4x Laemmli buffer and analyzed by western blot analysis as described above.

## T7E1 endonuclease and BsaHI enzymatic assays

To measure Cas9-mediated gene editing, DNA was isolated 72 h after EV-mediated Cas9 delivery or Cas9 plasmid DNA transfection, using the Qiagen DNeasy Blood & Tissue Kit (Qiagen) following the manufacturer's protocols. PCR was performed to amplify the sequences surrounding the targeted loci using Phusion High-Fidelity Polymerase (New England Biolabs), on 200 ng genomic DNA. PCR products were purified using the PureLink PCR Purification Kit (ThermoFisher Scientific) according to the manufacturer's instructions. Next, 200 ng PCR product was denatured per sample at 95 °C for 10 min in NEB Buffer 2.1 (New England Biolabs), and re-annealed by slowly lowering the temperature to 25 °C (95–85 °C at 2 °C per second and 85–25 °C at 0.1 °C per second) using a C1000 Touch Thermal Cycler (Bio-Rad). Re-annealed PCR products were incubated with 10 units of T7E1 (New England Biolabs) at 37 °C for 15 min. T7E1-treated PCR products were run on a 2% TAE agarose gel with 1:15.000 Midori Green Nucleic Acid Stain (NIPPON Genetics Europe), and imaged using a ChemiDoc XRS +

(Bio-Rad). Band intensities were measured using ImageJ v1.54d, and gene editing efficiency was calculated using the following formula: *%T7 efficiency = (1 - (1 - fraction cleaved)^0.5) * 100*, where *Fraction cleaved = (digested band intensity) / (digested band intensity + undigested band activity)*. All measured product intensities were corrected for lane background signal intensity prior to calculations. To measure specific ABE-mediated gene editing, DNA was isolated and PCR products were generated as described above, and were subsequently restricted with BsaHI (New England Biolabs) for 60 minutes at 37 °C according to the manufacturer's instructions. BsaHI-restricted PCR products were run on a 2% TAE agarose gel with Midori Green and image as described above. Gene editing efficiency was calculated using the following formula: *%BsaHI activity = (Fraction Cleaved) * 100*. All DNA and RNA concentrations were quantified using a NanoDrop One Spectrophotometer (ThermoFisher Scientific). Primer sequences are listed in Supplementary Table 5.

## Next generation sequencing

Genomic DNA was extracted from HEK293T cells following either transient transfection or extracellular vesicle (EV) treatment using the Maxwell® RSC Cell DNA Purification Kit (Promega, USA). PCR amplification was then performed using the extracted genomic DNA as templates, using the primers listed in Supplementary Table S5. The resulting PCR products were run on a 1% agarose gel, followed by gel extraction using the Monarch® Spin DNA Gel Extraction Kit (New England Biolabs, USA) to purify the target bands. The purified PCR amplicons were submitted to Eurofins Genomics for next-generation sequencing (NGS) using the INVIEW CRISPR Check service. Eurofins carried out adapter ligation to attach Illumina-compatible sequencing adapters, generating approximately 5 million paired-end reads per sample or sample pool. To assess genome editing efficiency and mutation profiles, we utilized a multi-step analysis pipeline combining local processing with online tools. Raw paired-end reads in FASTQ format were first merged using FLASH (Fast Length Adjustment of Short Reads) to reconstruct full-length amplicons, improving base accuracy near the Cas nuclease cleavage site. Following this, reads were demultiplexed based on barcode sequences to enable sample-specific analysis. The processed reads were analyzed using Cas-Analyzer[71].

## MTS cytotoxicity assay

To determine the toxicity of engineered VSV-G⁺ EVs, HEK293T cells were subjected to an MTS cytotoxicity assay. HEK293T cells were seeded in full culture medium, followed by an overnight incubation at 37 °C and 5% $CO_2$. MCP-CD63 RNP-loaded VSV-G⁺ EVs were added at a concentration of $1.0 \times 10^{10}$, $1.0 \times 10^{11}$ and $1.0 \times 10^{12}$ per $1.0 \times 10^5$ cells. 72 h post-transfection, cytotoxicity was determined with the CellTiter 96® AQueous One Solution Cell Proliferation Assay (MTS) (Promega, Madison, WI, USA) according to the manufacturer's protocol. Absorbance was measured at 490 nm using a SpectraMax iD3 multi-mode microplate reader (Molecular Devices). To calculate cell viability, 1% Triton X-100 treated samples were measured as a reference sample for 0% viability. Relative viability as compared to untreated samples was calculated using the following formula: ([*Sample absorbance*]-[*Triton X-100 sample absorbance*])/[*Untreated sample absorbance*].

## Statistics & Reproducibility

All statistical analyses were performed using GraphPad Prims v10.2 software. Unless stated otherwise, values are presented as mean ± standard deviation (SD). Two-sided statistical tests were performed in all statistical analyses within this manuscript. Statistical significance was considered at $p < 0.05$. No statistical method was used to predetermine sample size. No data were excluded from the analyses. The experiments were not randomized, and the Investigators

were not blinded to allocation during experiments and outcome assessment.

## Reporting summary

Further information on research design is available in the Nature Portfolio Reporting Summary linked to this article.

## Data availability

NGS datasets generated in this study are deposited to the NIH Sequence Read Archive with BioProject accession number PRJNA1346116 [https://www.ncbi.nlm.nih.gov/bioproject/1346116]. The data supporting the findings of this study are available within the Article, Supplementary Information, or Source Data file. The source data underlying Figs. 1b, d–h, 2c, d, 3b–d, g, i, 4c, d, 5b, d–g, and Supplementary Figs. 2b, 3b, 4b, d, f, h, J, 5a–h, 6a–g, 7a–c, 8a–b, 9 d, 10a, c, 11b, d, 12c, and 13a–f, h–i are provided as a Source Data file. The raw Source Data files of microscopy and flow cytometry analyses are available upon request from the corresponding author, o.g.dejong@uu.nl. Source data are provided in this paper.

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

## Acknowledgements

O.G.d.J. was supported by a VENI Fellowship (VI.Veni.192.174) from the Dutch Research Council (NWO). The work of O.E., R.M.S. and P.V. is supported by the European Union's Horizon 2020 Research and Innovation Program under grant agreement No. 825828 (EXPERT). W.S.d.V. and P.V. are supported by the European Research Council (ERC) Starting grant OBSERVE (No. 851936). C.V.H. was supported by the Utrecht Institute for Pharmaceutical Sciences (UIPS). S.A.A.K. was supported by an Open Mind grant (grant no. 21580) from the Dutch Research Council (NWO). Figures 1A, 2A, 3A, 3F, 4A, 5A, Supplementary Fig. S2A, S9A, and S12A were generated using Biorender.com.

## Author contributions

O.G.d.J and P.V. initiated this project. S.E.A., R.M.S., S.A.A.K. and E.M. helped design and plan the project. O.G.d.J., P.V., C.V.H. and O.E. planned and designed the experiments. T.A.P.D., X.L., I.L., O.L.C, I.Y.d.G., Z.E.N.M.J.d.W., A.G.G., N.J.A.M., S.H.B., E.D.C.B., J.W.L., W.S.d.V., J.J.F. and A.C.W.v.W. helped design and perform the experiments. O.G.d.J., O.E. and C.V.H. designed and developed most of the methodology. O.G.d.J. was responsible for the overall project strategy and management and wrote the manuscript, which was reviewed by all authors.

## Competing interests

O.G.d.J. is on the advisory board of The Organoid Company, Rotterdam, The Netherlands. S.E.A. is a consultant and stakeholder in Evox Therapeutics Limited, Oxford, United Kingdom. The remaining authors declare no competing interests.

## Additional information

[1]CDL Research, University Medical Center Utrecht, Utrecht University, Utrecht, The Netherlands. [2]Department of Pharmaceutics, Utrecht Institute of Pharmaceutical Sciences, Utrecht University, Utrecht, The Netherlands. [3]Division for Biomolecular and Cellular Medicine, Department of Laboratory Medicine, Karolinska Institutet, Stockholm, Sweden. [4]Karolinska ATMP Center, ANA Futura, Karolinska Institutet, Stockholm, Sweden. [5]Department of Cellular Therapy and Allogeneic Stem Cell Transplantation (CAST), Karolinska University Hospital, Stockholm, Sweden. [6]Department of Paediatrics, University of Oxford, Oxford, United Kingdom. [7]Institute of Developmental and Regenerative Medicine (IDRM), Oxford, United Kingdom. [8]Department of Metabolic Diseases, Wilhelmina Children's Hospital, University Medical Center, Utrecht University, Utrecht, The Netherlands. [9]Regenerative Medicine Center Utrecht, University Medical Center, Utrecht University, Utrecht, The Netherlands. [10]These authors contributed equally: Omnia M. Elsharkasy, Charlotte V. Hegeman.
✉e-mail: o.g.dejong@uu.nl

