## [Transparent Peer Review file · Nature Communications]

A modular strategy for extracellular vesicle-mediated CRISPR-Cas9 delivery through aptamer-based loading and UV-activated cargo release

Corresponding Author: Dr Olivier de Jong

Version 1:

Reviewer comments:

Reviewer #1

(Remarks to the Author)

I highly enjoyed reviewing this very elegant and interesting work by Elsharkasy and co-authors. The authors provide a sophisticated approach to engineer EVs for various gene editing approaches (Cas9 KO, CRISPRa, Base Editing) and I think this work will be of great interest to EV researchers. However, along with other comments shown below I believe it needs additional experiments to address off-targeting gene editing concerns, RNA integrity assessment after UV treatment, and in vivo or at a minimum in vitro disease-relevant gene editing.

Major comments

1. Please display individual data points in all bar graphs.
2. While the systems developed are very elegant and could help new generation of EV therapeutics, gene editing by EVs has already been demonstrated by multiple groups including with disease relevant gene corrections and/or in vivo systems which are lacking in this work. Can the authors please include in vivo experiments or at a minimum targeting of disease relevant genes instead of fluorescent reporters?
3. What was the rationale for EV dosage of "5.0 x 10¹⁰ EVs were added per 24-well plate well containing 1.0 x 10⁵ reporter cells"? Was this the highest possible dose to observe an effect (in terms of recipient cell viability, EV-production capacity, potential clinical application)? It appears that in initial experiments EVs were added at this fixed dose before implementing the PhoCI domain (Fig 2) or other recruiting tags (Fig 3A-E) and only later when trying different tags a dose response was assessed (Figure S3B). An issue with this is that it is a bit hard to follow the effect of each step in improving delivery due to the changes in dosage. In Fig 2D we see that the system has 1.5% editing activity before addition of PhoCI which shows in Fig 3D-E editing improving ~28% but closer inspection of figure legend (but not main text) reveals this can't be compared because dosage increases from 5x10¹⁰ to 1x10¹². Then within the same Fig3, comparison of recruiting tags is shown in 3G-H and editing is closer to 5% with CD63 and 5x10¹⁰ dosage. It may be hard for readers to follow that, if I understand correctly PhoCI may represent a ~3 fold increase (~5%/1.5%) but not a ~19 fold increase (~28%/1.5%). To improve the ability to assess the effect of each engineering step please show the results at the same 5x10¹⁰ or 1x10¹² dosage.
4. Figure 1 G and H show loading of sgRNA into EVs in the presence and absence of MCP-CD63, with an impressive 40~50 fold increase. Nevertheless, instead of showing increase compared to the control lacking MCP-CD63, please represent sgRNA loading in terms of copies per EV. This can easily be obtained using the qPCR results and the NTA particle counts as other studies have shown. For example, it has been reported that far less than one molecule of a given miRNA is present per exosome, actually closer to 1 miRNA per 100 exosomes, even for the most abundant miRNAs (PMID: 25267620). Using such an example, with a 40~50 fold increase in sgRNA does it mean we can assume 1 in 2 exosomes are loaded with sgRNA? Instead of having to assume, the readers would highly benefit from the actual number provided instead of the fold increase parameter. Additionally, this parameter could also help estimate copy number of Cas9 per EV it is actually recruited into EVs by the sgRNA. Lastly, comparison of Cell and EV fractions in 1G appears to indicate more sgRNA content in cells than EVs. Can Figure H also include a comparison relative sgRNA levels between cells and EVs?
5. Figure 2C and the accompanying text describe that "no significant decrease in Cas9-mediated activation of eGFP expression was observed in T MS2-sgRNA as compared to WT T sgRNA". However there appears to be a 10-20% reduction in GFP activation in the case of MS2-sgRNA. Were these two groups compared statistically to each other or just to

the untreated condition? To avoid confusion, the authors should label the statistical significance to NT sgRNA as a better control than untreated where no editing is reasonably expected. Furthermore, if there is no statistically significant difference despite the 10-20% decrease in editing between MS2 and unmodified sgRNA then please also include a label between both bars indicating "n.s.". However, if data show statistically significant difference in editing, please simply amend the accompanying text to reflect some x% decrease in editing activity is observed with MS2 modification of the sgRNA. Furthermore, Fig 4C also appears to show MS2 modification of the sgRNA results in lower dCas9-VPR transcriptional activation but also here no statistical label is included. Lastly, this is also noted in the discussion section "In line with previous reports, this did not compromise the functionality of Cas933" but the statistical comparison is missing indicating "n.s." differences between the conditions noted above.

6. Cleavage of PhoCI domain needs UV-treatment but it has been well established that UV treatment can degrade DNA and RNA. The authors note this in the discussion and point to wavelength for damage being below 340nm while here 395nm is used. If proteins were the main cargo assessed to be functional there would be less or no concerns. However, due to the closeness of these wavelengths the potential degradation of the cargo sgRNA by the UV treatment needs to be further assessed by RNA sequencing. What percentage of cargo sgRNA remains undamaged, what percentage is damaged?

7. One of the main concerns for CRISPR gene editing approaches is the potential of unintended off-targeted editing. For every sgRNA sequence potential off target sites can be predicted bioinformatically based on number of mismatches and other parameters. However, in this study's approach, given the potential of mutating the targeting sgRNA with UV treatment, off-target editing must be evaluated with more care as there may be more than the predicted off-target sites if the sgRNA is mutated. Overall, this aspect is a severe weakness of the study as no off-target analysis was performed. Thus, the authors should include experiments to assess off-targets. It would be very important to have a control plasmid sgRNA condition, not UV-treated, to compare with EV-delivered and UV-treated sgRNA to assess potential differences in off-target profiles.

Minor comments

8. The authors made use of multiple (12?) addgene constructs to generate the constructs needed to develop their experimental system. However, there is no indication in the manuscript of an intent to deposit any of the generated construct for other researchers to potentially replicate or build upon their findings. I would suggest to please make these constructs available on addgene as this would be hugely beneficial for the EV community and also would lead to more citations to this work.

Reviewer #2

(Remarks to the Author)

The authors in this work developed a versatile strategy based on HEK293T cell-derived exosomes with MCP-PhoCI-CD63 or CD9 expressed on the exosomal membrane to deliver Cas9 ribonucleoprotein (RNP) complex. The bio-engineered exosome could encapsulate RNP through aptamer interactions during the formation of exosome in HEK293T cells and achieve intracellular RNP delivery with high efficiency upon light irradiation. Besides, the fusion CD63 protein was replaced with a series of EV-enriched markers such as CD9 and CD81 and various RNP cargos including dCas9 and ABE8e adenine base editor were explored using this system. It can be considered for publication provided that the following issues are addressed.

1. In the Cell culture part (Line 136), the authors mentioned about two type of human cells, HEK293T and MDA-MB-231, to generate exosomes. What are the differences in the exosomes they derived? Why were MDA-MB-231 cell-derived exosomes used only in RNA isolation part (Line 284)?

2. In Line 350, HEK293 cells were transfected with the plasmid to fabricate RNP-loaded EV. For a better understanding, the detailed mechanism of RNP interaction with MCP-fusion protein and its entrapment into exosomes in HEK293 cells should be thoroughly explained.

3. In Fig 1E, the amount of GAPDH was not consistent in the 4th lane, and the Cas9 protein in the 5th lane was blurred. Please replace it with a clean background image.

4. In Fig 1E-H, to better demonstrate the interaction between Cas9 protein and CD63 fusion protein, the co-IP experiment is suggested using Flag beads to collect Cas9 and to detect whether CD63 is present.

5. In Fig 3B, the GAPDH in the last lane should be consistent with previous lanes to demonstrate that the photocleaved proteins are increased upon light irradiation. Besides, in Fig 3C the GAPDH background is quite dirty. New results with clean background are suggested to present.

6. In Fig 3I, the quantitative results of Cas9/GAPDH are needed to demonstrate the different Cas9 loading capacity in different fusion EV markers' groups, especially for RNP only group.

7. The flow cytometric results with MCP-PhoCI-CD63 in Fig 3D (around 30%) and 3G (only 5%) were not consistent. Please explain this phenomenon.

8. In Fig 4B, it seems that the eGFP intensity of T MS2-sgRNA group is stronger than T sgRNA group. However, the MFI results in Fig 4C showed that T sgRNA is better. The confocal imaging and flow cytometric data should be consistent to draw the conclusion.

9. In Fig 5C, it seems that some eGFP positive cells do not exhibit red fluorescence. Therefore, the colocalization figure of

eGFP and mCherry is suggested to present and show the co-expression of two fluorophores using the reporter system.

10. In Line 677, the authors mentioned that LNPs might be limited due to endosomal entrapment. To demonstrate the advantages of EV-based delivery system, eGFP or mCherry fragment is suggested to be inserted into the plasmid and acquire fluorescent EV with RNP loaded to show the endosomal escape (LysoTracker) behaviors and the nucleus accumulation (DAPI / Hoechst) ability using confocal imaging.

11. In Line 755-760, the authors provided a brief discussion regarding the issue of light wavelength. However, UV with potential skin toxicity and limited tissue penetration depth may not be applicable for human therapeutics. More discussion about wavelength and the possibility of using longer wavelength light-responsive protein linkers is suggested to be added in this part.

12. In Fig 3A, the authors mentioned about the photocleavage to release RNP complex to enhance the gene editing function. However, after cleavage, the RNP is still entrapped inside the exosome. The RNP drug release profiles with or without light irradiation under physiological conditions should be provided.

Reviewer #3

(Remarks to the Author)

The manuscript "A modular strategy for extracellular vesicle-mediated CRISPR-Cas9 delivery through aptamer-based loading and UV-activated cargo release" presents a versatile, modular approach for loading and delivery of Cas9 via extracellular vesicles (EVs). The work includes detailed investigations into the impacts and improvements of loading and delivery of various gene editing tools. The presentation of the work is well-organized. However, in my opinion the manuscript presents important flaws, for which I cannot recommend its acceptance.

Firstly, several key methods appear inaccurate or are missing. For instance, the use of GAPDH as an internal reference protein for quantifying cargos in EVs is not recommended; Proteins on the EV membrane like CD81 (or CD63) would be more reliable. Additionally, no direct evidence for gene editing outcomes was shown in this work. Although engineered cells, including HEK293T stoplight reporter cells and adenine base editor stoplight cells, were utilized to evaluate the cargo loading and delivery efficiency, precise gene editing and off-target effects should be assessed through deep sequencing, especially for endogenous genes. The manuscript mentions that MS2 aptamer-MCP binding on the second stemloop of the sgRNA has a strong negative effect on ABE8e functionality, potentially impacting editing efficiency; However, the loading capacity of the EVs in this work was not investigated. Notably, suboptimal activation efficiency was observed when EVs were loaded with dCas9-VPR transcriptional activator. This decrease in efficiency may be attributed to the larger size of the ABE8e editor compared to Cas9. Furthermore, the manuscript lacks investigation into the impacts of UV exposure on the integrity of EVs, including morphology changes, size alterations and protein profiles changes. The safety evaluation of the EVs treatment was also not investigated.

Secondly, the introduction and discussion sections require careful revision. Some information is outdated. For example, recent studies have reported that a series of LNPs exhibit tropism not only to the liver but also to various organs including the lung, spleen, and bone marrow. The lower immunogenicity of EVs in comparison with LNPs is controversial. It has been reported that factors that are likely to contribute to EV immunogenicity include origin, size, surface composition, internal content, production/storage methods, dosage, infusion rate, and the biomolecular corona (REF: Adv. Mater. 2024, 2403199).

Version 2:

Reviewer comments:

Reviewer #1

(Remarks to the Author)

Reviewer #2

(Remarks to the Author)

The authors have provided additional data and adequately responded the comments. The acceptance of this manuscript for publication is suggested.

Reviewer #3

(Remarks to the Author)

I co-reviewed this manuscript with one of the reviewers who provided the listed reports. This is part of the Nature Communications initiative to facilitate training in peer review and to provide appropriate recognition for Early Career

Researchers who co-review manuscripts.

Elsharkasy et al. reported an EV strategy for CRISPR-Cas9 editor delivery, they encapsulate the Cas9 or other editors by aptamer interaction and UV exposure. I find the experimental design of this work to be reasonable and logically coherent. However, the current version of the manuscript is too long to read. The authors should focus more on the main study and reduce unrelated content and discussion. Please bring the word count down to below 5,000 words, which is in line with the length of most Nature Communications articles. Besides, I have further concerns need to be issued.

Major concerns:

1. Most of the gene editing experiments described in the manuscript were performed in HEK293T cells, which are relatively easy to transfect. High editing efficiency in this cell line does not indicate therapeutic potential. To strengthen the evidence for an EV-based strategy, it would be more convincing to evaluate its effectiveness in human or mouse primary cells, or in animal models.
2. The T7E1 assay and TIDE cannot detect mutations below 1%, making them unsuitable for effectively identifying off-target editing. The authors should consider using deep sequencing for thorough off-target analysis.

Minor concerns:

1. VSVG has been used in these extracellular vesicles to enhance their cellular entry. However, in the authors' current schematic, the presence of VSVG is not depicted. It is recommended that the authors revise the schematic to include VSVG.
2. Western blot image of Cas9 in Fig3i is over exposure, which is not allowed in Nature Communications.

Version 3:

Reviewer comments:

Reviewer #1

(Remarks to the Author)

Thank you for an incredible work to address all comments. While there are some areas with respectful difference of opinion, all key scientific questions have been addressed resulting in much stronger evidence for the conclusions.

Reviewer #3

(Remarks to the Author)

After reviewing the newly included data in the manuscript, I believe that the main concerns I raised have been adequately addressed.

But the different colors of the response is quite confusing to read, just use two color, black for reviewer, blue for yourself! And you should attached revised data and figure in the point-by-point response to the reviewers, don't let us to find them in the maintext or SI.

REVIEWER COMMENTS

Reviewer #1 (Remarks to the Author):

I highly enjoyed reviewing this very elegant and interesting work by Elsharkasy and co-authors. The authors provide a sophisticated approach to engineer EVs for various gene editing approaches (Cas9 KO, CRISPRa, Base Editing) and I think this work will be of great interest to EV researchers. However, along with other comments shown below I believe it needs additional experiments to address off-targeting gene editing concerns, RNA integrity assessment after UV treatment, and in vivo or at a minimum in vitro disease-relevant gene editing.

We would like to thank the reviewer for their time and positive comments, and for their suggestions and comments. We will address their comments per point below. Comments of the reviewers are shown in black, our replies are shown in blue text. Quoted text that has been added to the manuscript is highlighted in yellow.

Major comments

1. Please display individual data points in all bar graphs.

Individual data points have now been included to all bar graphs in the main manuscript and supplementary figures.

2. While the systems developed are very elegant and could help new generation of EV therapeutics, gene editing by EVs has already been demonstrated by multiple groups including with disease relevant gene corrections and/or in vivo systems which are lacking in this work. Can the authors please include in vivo experiments or at a minimum targeting of disease relevant genes instead of fluorescent reporters?

Whereas we feel that in vivo gene editing experiments are outside of the scope of the current manuscript, we agree that targeting of endogenous genetic targets would be of significant value for the relevance and impact of this manuscript. To address this point, we have included data on targeting of the endogenous CCR5 gene by EV-mediated delivery of both Cas9 and the adenine base editor ABE8e. First we verified gene editing of WT and MS2-sgRNAs on the targeted CCR5 locus via plasmid transfection, and then performed a comparison of MCP-PhoCl-CD9 and MCP-PhoCl-CD63 delivery of Cas9. Lastly, we performed a dose-response experiment for MCP-PhoCl-CD63 delivery of ABE8e (shown in the newly included Supplementary Figure S10, panels A – C). The following text was added to the manuscript:

Finally, we investigated the suitability of MCP-PhoCl-mediated EV CRISPR-Cas9 delivery for endogenous gene-editing. To this end, we targeted the CCR5 gene, which is present on chromosome 3 and consists of four exons and two introns⁵¹. First, we cloned a targeting sequence that we verified in a previous study¹⁷ in plasmids for the expression of WT sgRNA, MS2-sgRNA 1.1, and MS2-sgRNA 2.0. Transfection of these 3 plasmids alongside a plasmid expressing Cas9 into HEK293T cells resulted in similar levels of gene editing of the CCR5 locus, as was confirmed by T7E1 endonuclease analysis (Fig. S10A). Next, CCR5 was targeted by EV-mediated Cas9 delivery using MCP-PhoCl-CD63 and MCP-PhoCl-CD9, with co-expression of VSV-G. Once more, successful CCR5 gene editing was confirmed by T7E1 endonuclease analysis (Fig. S10B). As previously observed, MCP-PhoCl-CD9-mediated Cas9 delivery showed higher levels of gene-editing than was observed for MCP-PhoCl-CD63. Lastly, adenine base editor ABE8e was delivered using MCP-PhoCl-CD63 in combination with a CCR5 targeting MS2-sgRNA 1.1, with co-expression of VSV-G. Using a T7E1 endonuclease assay, a dose-dependent effect on gene editing was observed, confirming successful EV-mediated ABE8e delivery (Fig. S10C).

3. What was the rationale for EV dosage of “ 5.0×10^{10} EVs were added per 24-well plate well containing 1.0×10^5 reporter cells”? Was this the highest possible dose to observe an effect (in terms of recipient cell viability, EV-production capacity, potential clinical application)? It appears that in initial experiments EVs were added at this fixed dose before implementing the PhoCl domain (Fig 2) or other recruiting tags (Fig 3A-E) and only later when trying different tags a dose response was assessed (Figure S3B). An issue with this is that it is a bit hard to follow the effect of each step in improving delivery due to the changes in dosage. In Fig 2D we see that the system has 1.5% editing activity before addition of PhoCl which shows in Fig 3D-E editing improving $\sim 28\%$ but closer inspection of figure legend (but not main text) reveals this can't be compared because dosage increases from 5×10^{10} to 1×10^{12} . Then within the same Fig3, comparison of recruiting tags is shown in 3G-H and editing is closer to 5% with CD63 and 5×10^{10} dosage. It may be hard for readers to follow that, if I understand correctly PhoCl may represent a ~ 3 fold increase ($\sim 5\%/1.5\%$) but not a ~ 19 fold increase ($\sim 28\%/1\%$). To improve the ability to assess the effect of each engineering step please show the results at the same 5×10^{10} or 1×10^{12} dosage.

We agree with the reviewer that this description was somewhat confusing, as this did not apply to all experiments. Instead, we have now clearly listed the amount of EVs added in the respective figure legends of all EV addition experiments. Moreover, to avoid confusion the statement

“Unless stated otherwise, 5.0×10^{10} EVs were added per 24-well plate well containing 1.0×10^5 reporter cells” in the Methods section has been replaced with the following statement:

Unless stated otherwise, cells subjected to EV addition experiments were cultured in 24-well plate wells at a density of 1.0×10^5 cells per well in 1 ml culture medium at the time of EV addition.

Moreover, we agree that that the comparison between the experiments in panel 2D and 3D is highly valuable to determine the effect of PhoCl-mediated cargo release on RNP delivery, which is why it's particularly unfortunate that we had previously omitted to clearly mention that both conditions had received the same amount of EVs (1.0×10^{12}). We have addressed this by updating figure legends as stated above, but we have also included an additional dose response range of MCP-CD63 RNP EV delivery (shown in the newly included Fig. S3A, S3B), as this was previously only shown for MCP-PhoCl-CD63 RNP delivery (Fig. S6B). We thank the reviewer for pointing out this lack of clarity.

We have added the following text to refer to the newly added dose response range:

Addition of 5.0×10^{10} and 1.0×10^{11} EVs showed even lower levels of gene editing, at 0.2% and 0.3% respectively (Fig. S3A, S3B).

4. Figure 1 G and H show loading of sgRNA into EVs in the presence and absence of MCP-CD63, with an impressive $40\sim 50$ fold increase. Nevertheless, instead of showing increase compared to the control lacking MCP-CD63, please represent sgRNA loading in terms of copies per EV. This can easily be obtained using the qPCR results and the NTA particle counts as other studies have shown. For example, it has been reported that far less than one molecule of a given miRNA is present per exosome, actually closer to 1 miRNA per 100 exosomes, even for the most abundant miRNAs (PMID: 25267620). Using such an example, with a $40\sim 50$ fold increase in sgRNA does it mean we can assume 1 in 2 exosomes are loaded with sgRNA? Instead of having to assume, the readers would highly benefit from the actual number provided instead of the fold increase parameter. Additionally, this parameter could also help estimate copy number of Cas9 per EV it is actually recruited into EVs by the sgRNA. Lastly, comparison of Cell and EV fractions in 1G appears to indicate more sgRNA content in cells than EVs. Can Figure H also include a comparison relative sgRNA levels between cells and EVs?

We agree that absolute counts of sgRNA per EV would be highly informative, and give more context to the engineering strategies employed in this manuscript. Unfortunately, we were not able to directly convert qPCR data into absolute sgRNA counts as no absolute synthetic sgRNA range was included. Instead, we opted to perform digital droplet PCR (ddPCR). When combined with NTA particle count and normalization to a Spike-in RNA to correct for RNA isolation efficiency, this technique allows for calculation of sgRNAs per EV. These data are shown in the newly included panel 1H.

Moreover, as previously stated in the Methods section of the first submission, qPCR data was derived from a different cell line that was stably expressing sgRNA +/- MCP-CD63. As these conditions are less suitable for a direct comparison to the rest of the data in this manuscript, this panel has been replaced with qPCR on HEK293T-derived EVs, loaded with Cas9 RNP using similar protocols as used in the rest of the manuscript. This is now shown in Figure 1G.

The following text has been added to the manuscript:

To assess whether MS2-sgRNA was also actively enriched in EVs by MCP-CD63, MS2-sgRNA loading was analyzed by qPCR (Fig. 1G). As was also observed for Cas9 protein loading, co-expression of MCP-CD63 resulted in a significant increase in MS2-sgRNA abundance in isolated EVs, showing an approx. 270 fold increase (SD \pm 94). To more accurately and quantitatively assess MS2-sgRNA loading, digital droplet PCR (ddPCR) was performed (Fig. 1H). Using a Spike-in RNA to correct for RNA isolation efficiency, absolute RNA loading was extrapolated from NTA particle counts. In line with previous results, a substantial increase in MS2-sgRNA loading was observed of 350-fold (SD \pm 12.4), increasing the abundance from 1 sgRNA per $\sim 6.0 \times 10^6$ EVs to 1 sgRNA per $\sim 1.7 \times 10^4$ EVs after active loading using MCP-CD63. Whereas these numbers are low as compared to synthetic nanocarriers, such orders of magnitude are not unexpected as similar numbers have previously been reported for sgRNA loading into EVs in other cell lines^{36,46}.

36. de Jong, O. G. *et al.* A CRISPR-Cas9-based reporter system for single-cell detection of extracellular vesicle-mediated functional transfer of RNA. *Nat. Commun.* **11**, (2020).

46. Murphy, D. E. *et al.* Natural or synthetic RNA delivery: A stoichiometric comparison of extracellular vesicles and synthetic nanoparticles. *Nano Lett.* **21**, 1888–1895 (2021).

Lastly, we opted not to include direct relative expression comparison between cell and EV lysates, as direct comparison of RNA content between cell and EV RNA content in qPCR remains somewhat of a debated topic: is it fair to directly compare these? Normalization on RNA amount is not directly physiologically accurate, and housekeeping genes suitable for such comparisons are lacking / somewhat controversial. For endogenous genes, studies often employ comparisons of its expression levels to other endogenous genes, so enrichment could be extrapolated from RNA ratios between certain genes in EVs and cells. However, as sgRNAs are non-endogenous RNA molecules, a common housekeeping gene suitable for direct comparison between cells and EVs would be required. For context, a previous position paper from the International Society for Extracellular Vesicles states the following “The choice of a normalization control is important when comparing expression levels between replicates and treatment conditions. It should be noted that those household genes that are generally used to normalize expression levels between samples of cellular RNA are not per se appropriate normalization controls in samples of evRNA.” (ISEV position paper: extracellular vesicle RNA analysis and bioinformatics, A. Hill *et al.*, Journal of Extracellular Vesicles, 2013). As such, we opted to focus on the specific enrichment of cargo in EVs instead, as is currently shown in panels 1G and 1H.

5. Figure 2C and the accompanying text describe that “no significant decrease in Cas9-mediated activation of eGFP expression was observed in T MS2-sgRNA as compared to WT T sgRNA”. However

there appears to be a 10-20% reduction in GFP activation in the case of MS2-sgRNA. Were these two groups compare statistically to each other or just to the untreated condition? To avoid confusion, the authors should label the statistical significance to NT sgRNA as a better control than untreated where no editing is reasonably expected. Furthermore, if there is no statistically significant difference despite the 10-20% decrease in editing between MS2 and unmodified sgRNA then please also include a label between both bars indicating “n.s.”. However, if data show statistically significant difference in editing, please simply amend the accompanying text to reflect some x% decrease in editing activity is observed with MS2 modification of the sgRNA. Furthermore, Fig 4C also appears to show MS2 modification of the sgRNA results in lower dCas9-VPR transcriptional activation but also here no statistical label is included. Lastly, this is also noted in the discussion section “In line with previous reports, this did not compromise the functionality of Cas933” but the statistical comparison is missing indicating “n.s.” differences between the conditions noted above.

Statistical analysis on the data of figure 2C indeed confirmed that there was no statistically significant differences between the WT sgRNA and the MS2-sgRNA. For clarity, an additional bar between these samples indicating “n.s.” has been included.

In the first submission of the manuscript, no direct statistical comparison was made between WT sgRNA and MS2-sgRNA transfection, as a Dunnett’s multiple comparisons test was initially performed. To answer the question regarding statistical comparison of these sgRNAs in Fig. 4C, the statistical analysis was replaced with a post-hoc Sidaks multiple comparisons test. Indeed, unlike seen in WT Cas9 in Fig. 2C, there does appear to be a significant decrease in transcriptional activation between WT sgRNA and MS2-sgRNA. Figure 4C has been replaced with the new statistical analysis and the following statement has been added to the Results section of the manuscript:

Unlike previously observed in Cas9 gene editing (Fig. 2C), transfection of MS2-sgRNA does show a lower transcriptional activation of eGFP expression as compared to WT sgRNA. However, MS2-sgRNA does still show substantial and significant transcriptional activation (Fig. 4B, 4C).

Moreover, the following statement was added to the Discussion section:

We did observe a decrease in dCas9-VPR-mediated transcriptional activation in MS2-sgRNAs as compared to unmodified sgRNAs. However, as this sgRNA still showed substantial and significant activity, we were still able to demonstrate transcriptional activation levels comparable to plasmid DNA transfection using MCP-PhoCl-CD9 (Fig. 4D).

We would like to thank the reviewer for pointing this out, as this finding may give future leads to further increase EV-mediated transcriptional activation.

6. Cleavage of PhoCl domain needs UV-treatment but it has been well established that UV treatment can degrade DNA and RNA. The authors note this in the discussion and point to wavelength for damage being below 340nm while here 395nm is used. If proteins were the main cargo assessed to be functional there would be less or no concerns. However, due to the closeness of these wavelengths the potential degradation of the cargo sgRNA by the UV treatment needs to be further assessed by RNA sequencing. What percentage of cargo sgRNA remains undamaged, what percentage is damaged?

We agree that a more in-depth analysis of the effects of 395 nm UV-treatment on our EVs is relevant, and was insufficiently addressed in the initial submission of the manuscript. Thus, to address this question, we chose to evaluate the effects of UV treatment on EV sgRNA content, as well as size distribution, protein content, and EV morphology using ddPCR, NTA analysis, immunoblotting, total protein stain, and transmission electron microscopy. These analyses showed no notable difference in

EV count, size, sgRNA content, protein content, or EV morphology. Data are presented in the newly included Figure S4A-S4J. The following text has been added to the Results section:

To further study whether UV treatment had any deleterious effects on EV morphology or cargo, MCP-CD63 EVs were treated for 20 minutes with UV and characterized by NTA analysis (Fig. S4). No significant changes in size distribution (Fig. S4A, S4B), mean and mode particle size (Fig. S4C, S4D), or in particle count (Fig. S4E) were observed. Whereas UV wavelengths generally associated with nucleotide damage are below 340 nm⁴⁸, we measured the effect of UV treatment on MS2-sgRNA abundance in MCP-CD63 RNP-loaded EVs using qPCR (Fig. S4F) and ddPCR (Fig. S4G) analysis to rule out UV-mediated sgRNA degradation. Indeed, both qPCR and ddPCR showed no difference in EV MS2-sgRNA abundance after UV treatment. Lastly, we measured the effect of UV treatment on protein cargo content and studied the effect on EV integrity and morphology using western blot analysis and transmission electron microscopy, respectively. Western blot analysis showed no difference in levels of Cas9 protein after UV treatment, or in EV markers ALIX, CD63, and syntenin. Total protein stain also showed no notable differences (Fig. S4H). Transmission electron microscopy also did not reveal any changes in EV morphology, integrity, or particle count (Fig. S4I, S4J). Altogether, we observe no evidence of notable deleterious changes in EV content or morphology after 20 minutes of 50W 395 nm UV exposure.

7. One of the main concerns for CRISPR gene editing approaches is the potential of unintended off-targeted editing. For every sgRNA sequence potential off target sites can be predicted bioinformatically based on number of mismatches and other parameters. However, in this study's approach, given the potential of mutating the targeting sgRNA with UV treatment, off-target editing must be evaluated with more care as there may be more than the predicted off-target sites if the sgRNA is mutated. Overall, this aspect is a severe weakness of the study as no off-target analysis was performed. Thus, the authors should include experiments to assess off-targets. It would be very important to have a control plasmid sgRNA condition, not UV-treated, to compare with EV-delivered and UV-treated sgRNA to assess potential differences in off-target profiles.

Indeed, alongside intracellular delivery, one of the main concerns for CRISPR-Cas9 gene therapy are off-target editing events. We agree that, independent of the delivery platform, this is an important hurdle for the field. However, we would like once more underline that multiple studies have shown that both DNA and RNA damage does not happen at wavelengths above 340 nm, but rather in the UVC (short wave, 100 – 280 nm) and UVB (middle-wave, 280 -320 nm) range, rather than the UVA (320 – 400 nm) range that our experiments are performed in: <https://pubmed.ncbi.nlm.nih.gov/22816040> <https://pubmed.ncbi.nlm.nih.gov/23856615>, <https://pubmed.ncbi.nlm.nih.gov/21613571>. However, we agree with the reviewer that it's better to err on the side of caution. To address this question, the main off-target sequence from our Stoplight targeting sgRNA sequence (chr2:+120580702) was analyzed for off-target effects by both a T7E1 endonuclease assay (Fig. S10E) and Tracking of Indels by DEcomposition (TIDE) analysis (Fig. S10F, S10G). Whereas clear gene editing activity was observed on the Stoplight targeting locus using T7E1 analysis (Fig. S10D), no difference was observed between EV-treated samples, untreated samples, and samples transfected with a Cas9 with non-targeting sgRNA was observed. The following text has been added to the manuscript:

One of the main concerns for therapeutic applications of Cas9-mediated endogenous gene editing is the potential for off-target gene editing effects. To assess potential off-target effects of EV-mediated Cas9 delivery, the highest predicted off-target site for the Cas9 Stoplight targeting sgRNA, chr2:+120580702 was analyzed⁵². First, Cas9 delivery and gene-editing activity was verified by analysis of the targeted Cas9 Stoplight locus, where both MCP-PhoCl-CD63 EV and plasmid DNA transfection-mediated delivery of Cas9 showed high levels of editing (42% ± 2.2%, and 66% ± 6.8%, respectively;

Fig. S10D). Next, off-target activity on chr2:+120580702 position was analyzed by a T7E1 endonuclease (Fig. S10E), after MCP-PhoCl-CD63 EV or plasmid DNA transfection-mediated delivery of Cas9. Both T7E1 endonuclease analysis showed some low levels of T7E1 background activity, but these were similar to T7E1 activity observed in samples treated with pDNA transfection of a non-targeting sgRNA (Fig. S10E). For a more precise quantification of off-target activity, we performed a more sensitive Tracking of Indels by DEcomposition (TIDE) analysis⁴⁴ on these samples (Fig. S10F, S10G). Here, once more, no difference was observed with both untreated samples and samples transfected with plasmid DNA expression Cas9 and a non-targeting sgRNA.

44. Brinkman, E. K., Chen, T., Amendola, M. & Van Steensel, B. Easy quantitative assessment of genome editing by sequence trace decomposition. *Nucleic Acids Res.* **42**, e168–e168 (2014).

52. Hsu, P. D. *et al.* DNA targeting specificity of RNA-guided Cas9 nucleases. *Nat. Biotechnol.* **2013** 319 **31**, 827–832 (2013).

Minor comments

8. The authors made use of multiple (12?) addgene constructs to generate the constructs needed to develop their experimental system. However, there is no indication in the manuscript of an intent to deposit any of the generated construct for other researchers to potentially replicate or build upon their findings. I would suggest to please make these constructs available on addgene as this would be hugely beneficial for the EV community and also would lead to more citations to this work.

We appreciate the suggestion, as we have indeed greatly benefitted from the availability of the constructs of other research groups via Addgene for this work. We are currently in the process of writing a manuscript that contains a variety of fluorescent reporter for Cas9 activity that we intend to publish as a single “CRISPR-Cas reporter tool set” that we intend to submit to Addgene. This collection will also contain all the reporter sequences used in this work as well. In the meantime, as we have done in previous work, we have included all sequences for our constructs in the supplementary information of this manuscript, and are (and previously have) always been willing to ship plasmids to other groups on request.

Reviewer #2 (Remarks to the Author):

The authors in this work developed a versatile strategy based on HEK293T cell-derived exosomes with MCP-PhoCl-CD63 or CD9 expressed on the exosomal membrane to deliver Cas9 ribonucleoprotein (RNP) complex. The bio-engineered exosome could encapsulate RNP through aptamer interactions during the formation of exosome in HEK293T cells and achieve intracellular RNP delivery with high efficiency upon light irradiation. Besides, the fusion CD63 protein was replaced with a series of EV-enriched markers such as CD9 and CD81 and various RNP cargos including dCas9 and ABE8e adenine base editor were explored using this system. It can be considered for publication provided that the following issues are addressed.

We would like to thank the reviewer for their time, suggestions, and comments. We will address their comments per point below. Comments of the reviewers are shown in black, our replies are shown in blue text. Quoted text that has been added to the manuscript is highlighted in yellow.

1. In the Cell culture part (Line 136), the authors mentioned about two type of human cells, HEK293T and MDA-MB-231, to generate exosomes. What are the differences in the exosomes they derived? Why were MDA-MB-231 cell-derived exosomes used only in RNA isolation part (Line 284)?

We appreciate this comment, as the PCR data in Figure 1 is indeed the only context in which MDA-MB-231 cells were used. The reason why they were initially included was mostly based on practical considerations: we wanted to prevent any potential false positive signal that may arrive from co-isolation of DNA fragments from transient transfection. As such, we opted to make stable expression cell lines for this experiments instead. However, due to the genomic instability of HEK293T cells it was substantially less challenging to generate stable cell lines that show consistent high levels of expression of both the MS2-sgRNA and the MCP-CD63 constructs using MDA-MB-231 cells. That being said, we fully agree that this experiment yields data that is difficult to compare to the other figures as these conditions are less suitable for a direct comparison to the rest of the data in this manuscript. Therefore, this panel has been replaced with qPCR on HEK293T-derived EVs, loaded with Cas9 RNP using similar protocols as used in the rest of the manuscript. This is now shown in Figure 1G. To confirm these data and to assess absolute MS2-sgRNA loading as well, digital droplet PCR (ddPCR) analysis was also performed. When combined with NTA particle count and normalization to a Spike-in RNA to correct for RNA isolation efficiency, this technique allows for calculation of sgRNAs per EV. These data are shown in the newly included panel 1H. To address our initial concerns on pDNA contamination, we have included a DNase treatment directly after RNA isolation, and performed negative reverse transcriptase controls.

The following text has been added to the manuscript:

To assess whether MS2-sgRNA was also actively enriched in EVs by MCP-CD63, MS2-sgRNA loading was analyzed by qPCR (Fig. 1G). As was also observed for Cas9 protein loading, co-expression of MCP-CD63 resulted in a significant increase in MS2-sgRNA abundance in isolated EVs, showing an approx. 270 fold increase (SD \pm 94). To more accurately and quantitatively assess MS2-sgRNA loading, digital droplet PCR (ddPCR) was performed (Fig. 1H). Using a Spike-in RNA to correct for RNA isolation efficiency, absolute RNA loading was extrapolated from NTA particle counts. In line with previous results, a substantial increase in MS2-sgRNA loading was observed of 350-fold (SD \pm 12.4), increasing the abundance from 1 sgRNA per $\sim 6.0 \times 10^6$ EVs to 1 sgRNA per $\sim 1.7 \times 10^4$ EVs after active loading using MCP-CD63. Whereas these numbers are low as compared to synthetic nanocarriers, such orders of magnitude are not unexpected as similar numbers have previously been reported for sgRNA loading into EVs in other cell lines^{36,46}.

2. In Line 350, HEK293 cells were transfected with the plasmid to fabricate RNP-loaded EV. For a better understanding, the detailed mechanism of RNP interaction with MCP-fusion protein and its entrapment into exosomes in HEK293 cells should be thoroughly explained.

Whereas we explain the rationale between our construct design, we indeed do not specifically explain how we envision the exact mechanisms through which this results in EV Cas9 loading. To further expand on this, the yellow highlighted text was added to the main text of the manuscript (surrounding text included for context, unhighlighted):

“Instead, we intraluminally fused tandem MS2 coat proteins (MCPs), lacking the Fg loop to prevent capsid formation, to the N-terminus of the EV-enriched tetraspanin CD63 (Fig. 1A). These MCP proteins robustly bind to MS2 aptamers; specific RNA hairpin sequences. To bind the Cas9-sgRNA ribonucleoprotein (RNP) complex, MS2 aptamers that can be bound by the tandem MCPs were placed in the tetraloop and second stemloop of the sgRNA. Previous studies have shown that replacing the hairpins in these sections of the sgRNA does not interfere with the Cas9 RNP functionality, and that these hairpins protrude from the Cas9 RNP structure³⁵. Thus, we envisioned that cellular co-expression of MS2-sgRNAs and Cas9 results in the formation of intracellular Cas9 RNPs with protruding MS2 aptamers, that can then be bound by the intraluminal tandem MCPs fused to CD63. As CD63 is highly

enriched in EVs, we hypothesized that this would result in Cas9 RNP loading during EV biogenesis (Fig S2A). In this modular approach, sgRNAs and Cas9 are readily interchangeable, as no direct Cas9 fusion proteins are employed.”

3. In Fig 1E, the amount of GAPDH was not consistent in the 4th lane, and the Cas9 protein in the 5th lane was blurred. Please replace it with a clean background image.”

Indeed, there were some inconsistencies in the loading control of this figure. To address this, we have repeated the experiment and included a more commonly employed EV loading control, syntenin, alongside the GAPDH loading control. We removed the original western blot image, and the new image is now shown in Fig. 1E. As can be seen, the Flag-tag antibody does give some additional background signal (shown in Fig. 1E, 3I, S2B and S4H). This is unfortunately inherent to this antibody. However, most commercial antibodies targeted against Cas9 give similar, or often more, levels of background. That being said, we have updated our blocking protocols during immunoblotting and were able to include an image with substantially less background than the original image.

4. In Fig 1E-H, to better demonstrate the interaction between Cas9 protein and CD63 fusion protein, the co-IP experiment is suggested using Flag beads to collect Cas9 and to detect whether CD63 is present.

We agree that this would more accurately display the interaction between Cas9 and CD63. Unfortunately, such an IP experiment on Cas9 to capture CD63 would come with significant technical challenges as CD63 is a protein with 4 transmembrane domains and it is strongly incorporated into the cell membrane. It is possible that membrane lysis would potentially allow co-precipitation of CD63, however these conditions may also affect the binding interactions between MS2 aptamers and the MS2 coat proteins that facilitate indirect binding of the Cas9 RNP to CD63. Thus, as an alternative, we performed immunoprecipitation on intact EVs containing RNPs loaded with our MCP-CD63 construct using CD63 antibodies. EVs were captured using magnetic protein G beads, and both the captured EVs and the remaining supernatant were subjected to RIPA buffer-mediated lysis followed by western blot analysis. As controls, a CD9 antibody targeting a different subpopulations as well as an IgG isotype control antibody were included. Indeed, only co-ip with the CD63 antibody resulted in Cas9 enrichment in the pulldown sample as compared to the remaining supernatant. A schematic and these results are shown in the newly included supplementary figure S2. The following text has been added to the Results section:

This was further confirmed by immune precipitation of MCP-CD63 RNP-loaded EVs with CD63, CD9, or IgG isotype control antibodies, using magnetic protein G-coated beads (Fig. S2A). After bead capture, pulled down EVs and remaining supernatants were both analyzed through western blot analysis for Cas9 (Flag-tag), CD63, and EV-marker syntenin. As expected, pulldown with a CD63 antibody resulted in an enrichment of Cas9 in the pulldown fraction as compared to its respective supernatant sample. CD9 capture did result in the pulldown of some CD63 and Cas9 protein, but the majority of both proteins remained in the supernatant (Fig. S2B). Interestingly CD9 did show higher levels of syntenin capture, as compared to CD63. These results are likely explained by the reported increased abundance of CD63 on EVs of endosomal origin, whereas CD9 is more commonly associated with cell membrane-derived EVs⁴⁵. No Cas9 protein or EV markers were detected after a bead capture pulldown using isotype control IgG. These data confirm that Cas9 loaded through MCP-CD63 is, at least partially, enriched in CD63-positive EVs.

5. In Fig 3B, the GAPDH in the last lane should be consistent with previous lanes to demonstrate that the photocleaved proteins are increased upon light irradiation. Besides, in Fig 3C the GAPDH background is quite dirty. New results with clean background are suggested to present.

We agree, and have adjusted the figure. The timepoint for $t = 60$ minutes was removed, and the experiment and western blot for figure 3C were repeated. EV samples now include the EV marker syntenin as a loading control instead.

6. In Fig 3I, the quantitative results of Cas9/GAPDH are needed to demonstrate the different Cas9 loading capacity in different fusion EV markers' groups, especially for RNP only group.

We have included protein quantification corrected for the GAPDH loading control in Fig. 3I, relative to the RNP only group.

7. The flow cytometric results with MCP-PhoCl-CD63 in Fig 3D (around 30%) and 3G (only 5%) were not consistent. Please explain this phenomenon.

Indeed, there is a difference in gene editing efficiency levels between these experiments. The differences in these percentages are explained by the different dosages of EVs added in both these experiments. As can be seen in the figure legend of Fig. 3, cells in Fig. 3D were given 1.0×10^{12} EVs, whereas cells in Fig. 3G were given 5.0×10^{10} EVs. The main reason for the difference in experimental approach was the amount of samples included in Fig. 3G. Due to practical limitations (limitations in cell culture / incubator space), we opted to compare a lower dosage as to minimize strain on the rest of our cell culture facilities as we performed an $n = 3$ over the course of 4 weeks for this experiment. However, we would like to point out that observed gene editing efficiency levels from both panels 3D and 3G are in line with the dose response range for MCP-PhoCl-CD63 shown in Fig. S6B.

8. In Fig 4B, it seems that the eGFP intensity of T MS2-sgRNA group is stronger than T sgRNA group. However, the MFI results in Fig 4C showed that T sgRNA is better. The confocal imaging and flow cytometric data should be consistent to draw the conclusion.

We agree that the selected images indeed did not fully align with the flow cytometry data. New images from the same experiment were included, that more closely resemble the flow cytometry data presented in Fig. 4C. Moreover, a more thorough statistical analysis on the difference of the WT sgRNA and the MS2-sgRNA within this experiment was included based on a comment from reviewer 1 regarding statistically significant differences between these samples.

In the first submission of the manuscript, no direct statistical comparison was made between WT sgRNA and MS2-sgRNA transfection, as a Dunnett's multiple comparisons test was initially performed. To answer the question regarding statistical comparison of these sgRNAs in Fig. 4C, the statistical analysis was replaced with a post-hoc Sidaks multiple comparisons test. Indeed, there does appear to be a significant decrease in transcriptional activation between WT sgRNA and MS2-sgRNA. Figure 4C has been replaced with the new statistical analysis and the following statement has been added to the Results section of the manuscript:

Unlike previously observed in Cas9 gene editing (Fig. 2C), transfection of MS2-sgRNA does show a lower transcriptional activation of eGFP expression as compared to WT sgRNA. However, MS2-sgRNA does still show substantial and significant transcriptional activation (Fig. 4B, 4C).

Moreover, the following statement was added to the Discussion section:

We did observe a decrease in dCas9-VPR-mediated transcriptional activation in MS2-sgRNAs as compared to unmodified sgRNAs. However, as this sgRNA still showed substantial and significant

activity, we were still able to demonstrate transcriptional activation levels comparable to plasmid DNA transfection using MCP-PhoCI-CD9 (Fig. 4D).

9. In Fig 5C, it seems that some eGFP positive cells do not exhibit red fluorescence. Therefore, the colocalization figure of eGFP and mCherry is suggested to present and show the co-expression of two fluorophores using the reporter system.

The reviewer raises a valid concern. Indeed, there are some levels of variation of mCherry and eGFP expression in reporter cells for the ABE Stoplight reporter construct, as is commonly seen in polyclonal cell populations. As can be seen by flow cytometry analysis in Fig. S1C, approx. 4 – 5 % of reporter cells are negative for mCherry expression. To correct for this, we do indeed only show co-expression of these two fluorophores in our quantitative analysis (flow cytometry): analysis of %eGFP-positive cells is only performed on mCherry-positive cells, as we are certain that these cells express the Stoplight reporter construct. Our gating strategy for these analyses are shown in Supplementary Figure 1, demonstrating that we indeed quantify eGFP+ signal after mCherry+ gating.

10. In Line 677, the authors mentioned that LNPs might be limited due to endosomal entrapment. To demonstrate the advantages of EV-based delivery system, eGFP or mCherry fragment is suggested to be inserted into the plasmid and acquire fluorescent EV with RNP loaded to show the endosomal escape (LysoTracker) behaviors and the nucleus accumulation (DAPI / Hoechst) ability using confocal imaging.

Whereas fully elucidating the exact mechanism of VSV-G is somewhat beyond the scope of this manuscript, we agree that we did not include sufficient information in our initial submission to support our hypothesis on endosomal escape. To address this, we included a live-imaging confocal analysis of intracellular Cas9 localization upon EV addition. However, due to practical reasons we implemented a few adaptations to the suggested experiment:

As we are specifically interested in the cellular localization of the Cas9 RNP, we agree that fluorescently labeled Cas9 would be the most optimal approach, as fluorescent membrane dyes would not be representative of RNP localization. However, to increase sensitivity, we incorporated the highly stable and bright green fluorescent protein mStayGold instead. As DAPI / Hoechst staining show toxicity over time in live-imaging experiments that may affect cell behavior, we instead opted to use a cell line that stably expresses fluorescently tagged Histone 2B as previous studies have shown that this results in permanent bright fluorescent signal in the nucleus (<https://pubmed.ncbi.nlm.nih.gov/9545195/>). Lastly, we opted not to include LysoTracker, as it exclusively shows fluorescence in acidic organelles. As VSV-G is specifically activated in acidic conditions, and as many fluorophores are quenched in acidic conditions, this would complicate interpretation of these data.

Ultimately, Cas9 tagged with mStayGold was loaded into EVs using the MCP-PhoCI-CD9 construct, either with- or without VSV-G co-expression. EVs were isolated, treated with UV, and added to HEK293T cells stably expressing mTag-BFP2 Histone 2B (H2B-mTag-BFP2). Cells were imaged up to 12 hours after EV addition using a live-imaging confocal system. After 1 hour, the majority of green fluorescent signal was present outside of the nuclei. At later timepoints green fluorescent signal was also observed inside the nuclei, but mainly upon administration of EVs containing VSV-G. These data are presented in the newly included Figure S7. Moreover, the following text was added to the manuscript:

To further elucidate the role of VSV-G in EV-mediated Cas9 delivery, EV-mediated delivery of fluorescently labeled Cas9 was analyzed by confocal microscopy. To this end, Cas9 was labeled with the highly stable mStayGold green fluorescent protein⁴², and loaded into EVs using MCP-PhoCI-CD9

(Fig. S7A), either with- or without co-expression of VSV-G. After UV-mediated cargo release, the EVs were added to HEK293T cells stably expressing mTag-BFP2 labeled Histone 2B (H2B), allowing for confocal microscopy-mediated measuring of Cas9-mStayGold nuclear localization as fluorescently tagged H2B expression gives high levels of fluorescence in the nucleus⁴⁹. 1 hour after addition green fluorescent signal was observed in spotted patterns within the recipient cells as compared to untreated cells, including the perinuclear area, but no substantial levels of Cas9-mStayGold were observed in the nuclei for both -VSV-G and +VSV-G MCP-PhoCl-CD9 EV-treated cells (Fig. S7B, S7D). However, starting from 2 hours after EV addition, a significant increase in green fluorescent signal was observed in the nuclei of +VSV-G EV-treated cells, as compared to both untreated and -VSV-G EV treated cells (Fig. S7C, S7D). This pattern remained unchanged up until the final timepoint of the experiment at 12 hours after EV addition (Fig. S7D). These data show that, under these conditions and within these timeframes, VSV-G plays a pivotal role in facilitating EV-mediated Cas9 delivery to the nucleus of recipient cells.

42. Ivorra-Molla, E. *et al.* A monomeric StayGold fluorescent protein. *Nat. Biotechnol.* 2023 429 **42**, 1368–1371 (2023).

49. Kanda, T., Sullivan, K. F. & Wahl, G. M. Histone-GFP fusion protein enables sensitive analysis of chromosome dynamics in living mammalian cells. *Curr. Biol.* **8**, 377–385 (1998).

11. In Line 755-760, the authors provided a brief discussion regarding the issue of light wavelength. However, UV with potential skin toxicity and limited tissue penetration depth may not be applicable for human therapeutics. More discussion about wavelength and the possibility of using longer wavelength light-responsive protein linkers is suggested to be added in this part.

Indeed, whereas UV-activatable photocleavable domains show high efficiency and promise in ex vivo conditions, direct applicability of in situ UV treatment is limited due to the reported limited tissue penetration and skin toxicity of UV light. To circumvent this limitation, EVs could be treated prior to administration, or directly after isolation, to facilitate cargo release from the intraluminal EV membrane. As the cargo would still be encapsulated by the EV membrane, this would not result in the release of the cargo until after membrane fusion. However, we agree that UV-activatable photocleavable constructs are indeed limited in their applications for localized in situ cleavage and cargo release. The following statement has been added to the discussion of the manuscript to further discuss this:

However, despite the promising results achieved with these UV-activatable photocleavable domains, they do present certain limitations for in vivo applicability. In the methods employed in this manuscript, EVs are treated with UV for intraluminal release of the cargo, prior to their addition to cells. Such protocols could, theoretically, also be applied for in vivo cargo delivery: UV-treatment could either be applied prior to administration, or directly after isolation of EVs. However, UV-cleavable domains might not be suited for in vivo delivery where these photocleavable domains are activated at a specific site of interest for local protein cleavage, as UV wavelengths show limited tissue penetration and potential skin toxicity⁷². For such approaches, photoreleasable strategies that are activated using longer wavelengths would be preferable. Whereas direct photocleavable proteins in long wavelength ranges have not been described, light-switchable dimerization phytochromes that are associated under treatment of 600 nm light, and actively disassociated by 780 nm light such as MagRed have been described⁷³. However, as of yet these dimerizing phytochromes also show spontaneous disassociation over time in dark conditions. Thus, further engineering to shift the activation wavelength of photocleavable domains or to inhibit self-disassociation of long wavelength activatable phytochromes would be required for the future development of such in situ activatable release strategies.

72. Ash, C., Dubec, M., Donne, K. & Bashford, T. Effect of wavelength and beam width on

penetration in light-tissue interaction using computational methods. *Lasers Med. Sci.* **32**, 1909–1918 (2017).

73. Kuwasaki, Y. *et al.* A red light-responsive photoswitch for deep tissue optogenetics. *Nat. Biotechnol.* **2022 4011 40**, 1672–1679 (2022).

12. In Fig 3A, the authors mentioned about the photocleavage to release RNP complex to enhance the gene editing function. However, after cleavage, the RNP is still entrapped inside the exosome. The RNP drug release profiles with or without light irradiation under physiological conditions should be provided.

Indeed, whereas western blot analysis of a time range of UV treatment of the MCP-PhoCl-CD63 construct provides an indication of cleavage kinetics, this does not necessarily represent intracellular Cas9 delivery in physiological (cellular) conditions. Thus, instead of performing an end-point confocal microscopy experiment to study Cas9 localization within the cell to address Question 10, we opted to include a time range using a live-imaging confocal set-up instead. Cellular localization of fluorescently labeled Cas9 over the course of 12 hours after UV treatment has now been included in Fig. S7D to address this question.

Reviewer #3 (Remarks to the Author):

The manuscript "A modular strategy for extracellular vesicle-mediated CRISPR-Cas9 delivery through aptamer-based loading and UV-activated cargo release" presents a versatile, modular approach for loading and delivery of Cas9 via extracellular vesicles (EVs). The work includes detailed investigations into the impacts and improvements of loading and delivery of various gene editing tools. The presentation of the work is well-organized. However, in my opinion the manuscript presents important flaws, for which I cannot recommend its acceptance.

We would like to thank the reviewer for their time, suggestions, and comments. We will address their comments per point below. Comments of the reviewers are shown in black, or replies are shown in blue text. Quoted text that has been added to the manuscript is highlighted in yellow.

Firstly, several key methods appear inaccurate or are missing. For instance, the use of GAPDH as an internal reference protein for quantifying cargos in EVs is not recommended; Proteins on the EV membrane like CD81 (or CD63) would be more reliable.

Whereas it is not fully uncommon to use GAPDH as a western blot loading control for EV samples, we agree with the reviewer that EV markers are more suitable and preferable. As we generally overexpress EV membrane markers like CD9, CD63, or CD81 for our experimental conditions, we opted for a different EV marker instead: syntenin. We have repeated the experiments and replaced the blots for Fig. 1E and Fig. 2C, and have added syntenin as a loading control. Newly included supplementary figures S2B and S3H also include syntenin as a loading control for EV samples. The only western blot where opted not to change the loading control to an EV marker was Fig. 3I, as we are overexpressing a variety of EV-markers for comparison of cargo loading within the same experiment. As some of these EV markers play a role in EV biogenesis, overexpression of these proteins may affect co-expression of other EV markers. Therefore, we opted for a common housekeeping protein that is not involved in EV biogenesis for this specific experiment. Information on the new antibody staining procedure has also been added to the Methods section.

Additionally, no direct evidence for gene editing outcomes was shown in this work. Although engineered cells, including HEK293T stoplight reporter cells and adenine base editor stoplight cells,

were utilized to evaluate the cargo loading and delivery efficiency, precise gene editing and off-target effects should be assessed through deep sequencing, especially for endogenous genes.

We agree that targeting of endogenous genetic targets would be of significant value for the relevance and impact of this manuscript. Reviewer 1 raised similar concerns.

To address this point, we have included data on targeting of the endogenous CCR5 gene by EV-mediated delivery of both Cas9 and the adenine base editor ABE8e. First we verified gene editing of WT and MS2-sgRNAs on the targeted CCR5 locus via plasmid transfection, and then performed a comparison of MCP-PhoCI-CD9 and MCP-PhoCI-CD63 delivery of Cas9. Additionally, we performed a dose-response experiment for MCP-PhoCI-CD63 delivery of ABE8e (shown in the newly included Supplementary Figure S10, panels A – C). The following text was added to the manuscript:

Finally, we investigated the suitability of MCP-PhoCI-mediated EV CRISPR-Cas9 delivery for endogenous gene-editing. To this end, we targeted the CCR5 gene, which is present on chromosome 3 and consists of four exons and two introns⁵¹. First, we cloned a targeting sequence that we verified in a previous study¹⁷ in plasmids for the expression of WT sgRNA, MS2-sgRNA 1.1, and MS2-sgRNA 2.0. Transfection of these 3 plasmids alongside a plasmid expressing Cas9 into HEK293T cells resulted in similar levels of gene editing of the CCR5 locus, as was confirmed by T7E1 endonuclease analysis (Fig. S10A). Next, CCR5 was targeted by EV-mediated Cas9 delivery using MCP-PhoCI-CD63 and MCP-PhoCI-CD9, with co-expression of VSV-G. Once more, successful CCR5 gene editing was confirmed by T7E1 endonuclease analysis (Fig. S10B). As previously observed, MCP-PhoCI-CD9-mediated Cas9 delivery showed higher levels of gene-editing than was observed for MCP-PhoCI-CD63. Lastly, adenine base editor ABE8e was delivered using MCP-PhoCI-CD63 in combination with a CCR5 targeting MS2-sgRNA 1.1, with co-expression of VSV-G. Using a T7E1 endonuclease assay, a dose-dependent effect on gene editing was observed, confirming successful EV-mediated ABE8e delivery (Fig. S10C).

Indeed, alongside intracellular delivery, one of the main concerns for CRISPR-Cas9 gene therapy are off-target editing events. We agree that, independent of the delivery platform, this is an important hurdle for the field. To address this question, the main off-target sequence from our Stoplight targeting sgRNA sequence (chr2:+120580702) was analyzed for off-target effects by both a T7E1 endonuclease assay (Fig. S10E) and Tracking of Indels by DEcomposition (TIDE) analysis (Fig. S10F, S10G). Whereas clear gene editing activity was observed on the Stoplight targeting locus using T7E1 analysis (Fig. S10D), no difference was observed between EV-treated samples, untreated samples, and samples transfected with a Cas9 with non-targeting sgRNA was observed. The following text has been added to the manuscript:

One of the main concerns for therapeutic applications of Cas9-mediated endogenous gene editing is the potential for off-target gene editing effects. To assess potential off-target effects of EV-mediated Cas9 delivery, the highest predicted off-target site for the Cas9 Stoplight targeting sgRNA, chr2:+120580702 was analyzed⁵². First, Cas9 delivery and gene-editing activity was verified by analysis of the targeted Cas9 Stoplight locus, where both MCP-PhoCI-CD63 EV and plasmid DNA transfection-mediated delivery of Cas9 showed high levels of editing ($42\% \pm 2.2\%$, and $66\% \pm 6.8\%$, respectively; Fig. S10D). Next, off-target activity on chr2:+120580702 position was analyzed by a T7E1 endonuclease (Fig. S10E), after MCP-PhoCI-CD63 EV or plasmid DNA transfection-mediated delivery of Cas9. Both T7E1 endonuclease analysis showed some low levels of T7E1 background activity, but these were similar to T7E1 activity observed in samples treated with pDNA transfection of a non-targeting sgRNA (Fig. S10E). For a more precise quantification of off-target activity, we performed a more sensitive Tracking of Indels by DEcomposition (TIDE) analysis⁴⁴ on these samples (Fig. S10F, S10G). Here, once

more, no difference was observed with both untreated samples and samples transfected with plasmid DNA expression Cas9 and a non-targeting sgRNA.

42. Brinkman, E. K., Chen, T., Amendola, M. & Van Steensel, B. Easy quantitative assessment of genome editing by sequence trace decomposition. *Nucleic Acids Res.* **42**, e168–e168 (2014).

50. Hsu, P. D. *et al.* DNA targeting specificity of RNA-guided Cas9 nucleases. *Nat. Biotechnol.* **2013** 319 **31**, 827–832 (2013).

The manuscript mentions that MS2 aptamer-MCP binding on the second stemloop of the sgRNA has a strong negative effect on ABE8e functionality, potentially impacting editing efficiency; However, the loading capacity of the EVs in this work was not investigated.

This was indeed an overlooked parameter, and we would like to thank the reviewer for pointing this out as further investigating this point of concern has led to some relevant new insights in the effects of this steric hindrance on EV-mediated ABE delivery. To address this comment, ABE8e tagged with the highly stable and bright green fluorescent protein mStayGold was loaded into EVs by co-expression with sgRNAs with MS2 hairpins integrated in the tetraloop, second stemloop, both, or neither (WT sgRNA), and loaded into EVs using the MCP-CD63 construct followed by flow cytometry analysis. After isolation, EVs were treated with CD63 antibodies, and captured using Protein G-coupled beads. To confirm and assess EV capture, bead-bound EVs were stained with the self-quenching lipid dye MemGlow 640. As this lipid dye self quenches, it is not fluorescent in case of aggregation or micelle formation, but fluoresces when integrated and diluted into lipid membranes. After washing, EV-loaded beads were analyzed by flow cytometry to assess ABE8e-mStayGold loading with the various MS2-sgRNAs. Fluorescent data was compared to non-labeled ABE8e, as well as sgRNAs without MS2 aptamers. Strikingly, we only observed a significant increase in loading using sgRNA 1.1 (tetraloop MS2 aptamer only), but not with the sgRNA containing the MS2 aptamer in the second stemloop, or in both locations. There was a small increase in fluorescent signal, but this was similar to the signal in sgRNAs without MS2 aptamers – implying that this due to stochastic MS2-independent loading. Altogether, it appears that MCP binding to the second stemloop not only interferes with deoxyadenosine deaminase activity of ABEs, but that the presence of the deoxyadenosine deaminase also interferes with the MS2 hairpin-MCP interaction which in our case results in decreased ABE8e loading. These data are shown in the newly included Fig. S9A – S9D. The following text has been added to the manuscript as well:

These data demonstrate that MCPs bound to the second stemloop of the sgRNA inhibit EV-mediated ABE activity. If ABE activity is indeed hindered due to steric hindrance of the deoxyadenosine deaminase by the MCPs, the presence of the deoxyadenosine deaminase domain may also negatively affect EV loading by blocking the binding of MCPs to the MS2 hairpins of sgRNAs. To assess the effects of the location of MS2 hairpins in sgRNA on MCP-mediated ABE EV loading, fluorescently labeled ABE8e with mStayGold was co-expressed with MCP-CD63 and sgRNAs with MS2 hairpins integrated in the tetraloop (sgRNA 1.1), second stemloop (sgRNA 1.2), both (sgRNA 2.0), or neither (sgRNA 1.0). EVs were then isolated, treated with CD63 antibodies, and captured using Protein G-coupled beads, stained with the lipid membrane dye MemGlow 640, washed, and analyzed by flow cytometry to determine the effects of the varying MS2-sgRNAs on ABE8e loading (Fig. S9A, S9B). Indeed, only sgRNA 1.1 showed a significant increase in green fluorescent signal as compared to EVs loaded with unlabeled ABE8e. Both MS2 sgRNAs with MS2 hairpins in the second stemloop only (sgRNA 1.2), and MS2-sgRNAs with MS2 hairpins in both the tetraloop and the second stemloop (sgRNA 2.0) showed no difference in ABE8e-mStayGold loading as compared to WT sgRNA with no MS2 hairpins, and no significant increase in fluorescence as compared to EVs loaded with unlabeled ABE8e (Fig. S9C). These data show that MCPs bound to the second stemloop do not only inhibit ABE activity, but presence of

the deoxyadenosine deaminase domain on Cas9 also has an inhibitory effect on MCP binding to MS2 hairpins on the second stemloop, resulting in decreased ABE RNP EV loading.

Moreover, the following lines have been added to the Discussion section:

We did observe a decrease in dCas9-VPR-mediated transcriptional activation in MS2-sgRNAs as compared to unmodified sgRNAs. However, as this sgRNA still showed substantial and significant activity, we were still able to demonstrate transcriptional activation levels comparable to plasmid DNA transfection using MCP-PhoCl-CD9 (Fig. 4D). Furthermore, we did observe interference with the functionality of adenine base editor ABE8e using sgRNAs with MS2 hairpins in the second stemloop. As this was only observed when incorporating MS2 aptamers into the second stemloop of the sgRNA alongside co-expression of cytosolic MCPs, we hypothesize that the reduction in functionality was caused by steric hindrance of the MCPs with the interaction of the TadA domain with the targeted DNA sequence, as the second stemloop is relatively close to the TadA domain⁸⁰. Moreover, incorporation of the MS2 hairpin in the second stemloop of the sgRNA also showed decreased MCP-mediated ABE8e EV loading, which further impacts EV-mediated ABE8e delivery.

Notably, suboptimal activation efficiency was observed when EVs were loaded with dCas9-VPR transcriptional activator. This decrease in efficiency may be attributed to the larger size of the ABE8e editor compared to Cas9.

Whereas we do see far less brighter signal in the dCas9-VPR EV-treated cells, we do not necessarily agree that this is solely caused by suboptimal activation efficiency. Instead, we would like to point out that dCas9-VPR-mediated activation of the fluorescent reporter system does not facilitate a absolute “on-or-off” activation of the reporter as seen in the Cas9 and ABE8e fluorescent reporter, but rather a dose-dependent transcriptional response. This is why for these experiments fluorescent data is not expressed in %positive cells, but rather in Δ MF_I. That being said, we agree that transcriptional activation is substantially lower than the Doxycycline-mediated activation (positive control) of this inducible construct. This may, in part, be explained by the fact that we already see substantially lower activation of eGFP by dCas9-VPR pDNA transfection as compared to doxycycline treatment. However, we also see that MS2-sgRNA shows significantly lower dCas9-VPR-mediated transcriptional activation than the WT sgRNA.

That being said, we fully agree with the reviewer that the importance of cargo size on EV loading efficiency should not be overlooked and indeed warrants mentioning. Therefore, the following statements have been added to the text:

Results section:

Unlike previously observed in Cas9 gene editing (Fig. 2C), transfection of MS2-sgRNA does show a lower transcriptional activation of eGFP expression as compared to WT sgRNA. However, MS2-sgRNA does still show substantial and significant transcriptional activation (Fig. 4B, 4C).

Discussion section:

We did observe a decrease in dCas9-VPR-mediated transcriptional activation in MS2-sgRNAs as compared to unmodified sgRNAs. However, as this sgRNA still showed substantial and significant activity, we were still able to demonstrate transcriptional activation levels comparable to plasmid DNA transfection using MCP-PhoCl-CD9 (Fig. 4D).

Regarding the comment on the effects of increased size on loading: we have added the following sentence to the Discussion section:

Furthermore, it should be taken into consideration that the increase in size of Cas9 moieties containing additional enzymatic groups such as base editors, transcriptional activators and prime editors may also affect EV loading efficiency, as it has been reported that increased size of Cas9 due to protein conjugation may result in decreased EV loading⁸³.

Furthermore, the manuscript lacks investigation into the impacts of UV exposure on the integrity of EVs, including morphology changes, size alterations and protein profiles changes.

Indeed, we agree that a thorough characterization of the effects on UV integrity, cargo, size and morphology would be an appropriate addition to this manuscript. This point was also raised by reviewer 1 (Question 6), where we answered the following:

We agree that a more in-depth analysis of the effects of 395 nm UV-treatment on our EVs is relevant, and was insufficiently addressed in the initial submission of the manuscript. Thus, to address this question, we chose to evaluate the effects of UV treatment on EV sgRNA content, as well as size distribution, protein content, and EV morphology using ddPCR, NTA analysis, immunoblotting, total protein stain, and transmission electron microscopy. These analyses showed no notable difference in EV count, size, sgRNA content, protein content, or EV morphology. Data are presented in the newly included Figure S4A-S4J. The following text has been added to the Results section:

To further study whether UV treatment had any deleterious effects on EV morphology or cargo, MCP-CD63 EVs were treated for 20 minutes with UV and characterized by NTA analysis (Fig. S4). No significant changes in size distribution (Fig. S4A, S4B), mean and mode particle size (Fig. S4C, S4D), or in particle count (Fig. S4E) were observed. Whereas UV wavelengths generally associated with nucleotide damage are below 340 nm⁴⁸, we measured the effect of UV treatment on MS2-sgRNA abundance in MCP-CD63 RNP-loaded EVs using qPCR (Fig. S4F) and ddPCR (Fig. S4G) analysis to rule out UV-mediated sgRNA degradation. Indeed, both qPCR and ddPCR showed no difference in EV MS2-sgRNA abundance after UV treatment. Lastly, we measured the effect of UV treatment on protein cargo content and studied the effect on EV integrity and morphology using western blot analysis and transmission electron microscopy, respectively. Western blot analysis showed no difference in levels of Cas9 protein after UV treatment, or in EV markers ALIX, CD63, and syntenin. Total protein stain also showed no notable differences (Fig. S4H). Transmission electron microscopy also did not reveal any changes in EV morphology, integrity, or particle count (Fig. S4I, S4J). Altogether, we observe no evidence of notable deleterious changes in EV content or morphology after 20 minutes of 50W 395 nm UV exposure.

The safety evaluation of the EVs treatment was also not investigated.

Indeed, this is an important parameter to include. In terms of safety, we opted to investigate 2 parameters: off-target effects (as previously discussed), and a cytotoxicity assay. We have included analysis of EV Cas9-mediated off-target gene editing by T7E1 and TIDE analysis (Fig. S10E, S10F, S10G), as well as a MTS cytotoxicity assay of cells treated with 5.0×10^{10} and 5.0×10^{11} EVs, alongside Cas9 pDNA transfection (Fig. S10H). A positive control consisting of 1% Triton X-100 and a negative control consisting of untreated cells were also included in the MTS assay.

Alongside the aforementioned text included for off-target analysis, the following text was added to the manuscript regarding the newly performed MTS cytotoxicity assay:

Lastly, we performed an MTS assay on HEK293T cells after MCP-PhoCI-CD63 EV or plasmid DNA transfection-mediated delivery of Cas9 to test for potential toxicity (Fig. S10H). The MTS assay showed no signs of toxicity over the course of 72 hours after 5.0×10^{10} or 5.0×10^{11} MCP-PhoCI-CD63 RNP-

loaded EVs with co-expression of VSV-G were added per 1.0×10^5 HEK293T cells. Plasmid DNA transfection of the same constructs showed a trend towards a low level of toxicity ($p = 0.052$), and treatment with Triton X-100 resulted in complete loss of viability.

As discussed before in a previous comment, the following text was added regarding off-target analysis:

Finally, we investigated the suitability of MCP-PhoCl-mediated EV CRISPR-Cas9 delivery for endogenous gene-editing. To this end, we targeted the CCR5 gene, which is present on chromosome 3 and consists of four exons and two introns⁵¹. First, we cloned a targeting sequence that we verified in a previous study¹⁷ in plasmids for the expression of WT sgRNA, MS2-sgRNA 1.1, and MS2-sgRNA 2.0. Transfection of these 3 plasmids alongside a plasmid expressing Cas9 into HEK293T cells resulted in similar levels of gene editing of the CCR5 locus, as was confirmed by T7E1 endonuclease analysis (Fig. S10A). Next, CCR5 was targeted by EV-mediated Cas9 delivery using MCP-PhoCl-CD63 and MCP-PhoCl-CD9, with co-expression of VSV-G. Once more, successful CCR5 gene editing was confirmed by T7E1 endonuclease analysis (Fig. S10B). As previously observed, MCP-PhoCl-CD9-mediated Cas9 delivery showed higher levels of gene-editing than was observed for MCP-PhoCl-CD63. Lastly, adenine base editor ABE8e was delivered using MCP-PhoCl-CD63 in combination with a CCR5 targeting MS2-sgRNA 1.1, with co-expression of VSV-G. Using a T7E1 endonuclease assay, a dose-dependent effect on gene editing was observed, confirming successful EV-mediated ABE8e delivery (Fig. S10C). One of the main concerns for therapeutic applications of Cas9-mediated endogenous gene editing is the potential for off-target gene editing effects. To assess potential off-target effects of EV-mediated Cas9 delivery, the highest predicted off-target site for the Cas9 Stoplight targeting sgRNA, chr2:+120580702 was analyzed⁵². First, Cas9 delivery and gene-editing activity was verified by analysis of the targeted Cas9 Stoplight locus, where both MCP-PhoCl-CD63 EV and plasmid DNA transfection-mediated delivery of Cas9 showed high levels of editing ($42\% \pm 2.2\%$, and $66\% \pm 6.8\%$, respectively; Fig. S10D). Next, off-target activity on chr2:+120580702 position was analyzed by a T7E1 endonuclease (Fig. S10E), after MCP-PhoCl-CD63 EV or plasmid DNA transfection-mediated delivery of Cas9. Both T7E1 endonuclease analysis showed some low levels of T7E1 background activity, but these were similar to T7E1 activity observed in samples treated with pDNA transfection of a non-targeting sgRNA (Fig. S10E). For a more precise quantification of off-target activity, we performed a more sensitive Tracking of Indels by DEcomposition (TIDE) analysis⁴⁴ on these samples (Fig. S10F, S10G). Here, once more, no difference was observed with both untreated samples and samples transfected with plasmid DNA expression Cas9 and a non-targeting sgRNA.

Secondly, the introduction and discussion sections require careful revision. Some information is outdated. For example, recent studies have reported that a series of LNPs exhibit tropism not only to the liver but also to various organs including the lung, spleen, and bone marrow.

We agree that the discussion on LNP biodistribution was indeed somewhat limited, especially in regards to publications on the effects of lipid formulations on biodistribution. We have included the following text to the introduction and discussion (surrounding text included for context, unhighlighted):

Introduction:

“Both LNPs and polymer-based systems have shown potential for delivery of Cas9 RNP and mRNA, but show cellular toxicity and limited delivery efficiency due to endosomal entrapment²⁰. That being said, recent studies on the effects of formulation on biodistribution have lead to an increased understanding and control of LNP tissue targeting. For example, recent studies have shown that use of specific lipids increases targeting to lungs, spleen, liver, or bone-marrow^{21,22}”

Discussion:

“Whereas these LNP-based strategies have shown clinical promise for in vivo delivery of Cas9, their restricted biodistribution profile initially limited their application to gene editing strategies specifically targeting the liver. Recent studies have made great strides in improving tunability and control over LNP biodistribution. For example, using select organ targeting lipid (SORT) LNP formulations, LNP tropism can be shifted towards lungs or spleen²². Similarly, specific LNP formulations allow for the preparation of bone-marrow-homing lipid nanoparticles²¹. Moreover, focus has increasingly shifted to adapt LNP cell- and tissue targeting by incorporation of targeting antibodies⁵⁷. “

21. Lian, X. *et al.* Bone-marrow-homing lipid nanoparticles for genome editing in diseased and malignant haematopoietic stem cells. *Nat. Nanotechnol.* **19**, (2024).

22. Wang, X. *et al.* Preparation of selective organ-targeting (SORT) lipid nanoparticles (LNPs) using multiple technical methods for tissue-specific mRNA delivery. *Nat. Protoc.* **2022** *181* **18**, 265–291 (2022).

57. Marques, A. C., Costa, P. C., Velho, S. & Amaral, M. H. Lipid Nanoparticles Functionalized with Antibodies for Anticancer Drug Therapy. *Pharmaceutics* **15**, 216 (2023).

The lower immunogenicity of EVs in comparison with LNPs is controversial. It has been reported that factors that are likely to contribute to EV immunogenicity include origin, size, surface composition, internal content, production/storage methods, dosage, infusion rate, and the biomolecular corona (REF: Adv. Mater. 2024, 2403199).

We agree that this is still a somewhat controversial topic within the EV field, especially with the incorporation of the fusogenic VSV-G protein. We have added a section to the discussion discussing both EV and VSV-G immunogenicity. We would like to thank the reviewer for this comment, and have added the following text to the Discussion section:

Despite thorough documentation of EVs inately mediating intracellular transfer of biological cargos, including protein and RNA, we did not observe efficient EV-mediated Cas9 delivery without the incorporation of the Vesicular stomatitis virus G (VSV-G) fusogenic glycoprotein under the conditions described in this manuscript. VSV-G is commonly incorporated into biological particles such as lentiviral particles to aid in cargo delivery by facilitating endosomal escape⁷⁴. Indeed, confocal analysis of recipient cells treated with Cas9-loaded EVs showed a significant increase in fluorescently labeled Cas9 localization to the nucleus in recipient cells, as opposed to EVs without VSV-G that showed no increase of Cas9 localization to the nucleus over time (Fig. S7). Whereas incorporation of VSV-G facilitated high levels of Cas9 delivery, there are concerns regarding the immunogenicity of VSV-G. For example, a study by S. Kuate *et al.* demonstrated a 100-fold higher antibody titer response against structural proteins of viral like particles (VLPs) when VSV-G was co-expressed⁷⁵. Thus, future incorporation of less immunogenic or endogenous (human) fusogenic proteins such as members of the syncytin family might be preferable in the design of potential therapeutic EV-based delivery strategies⁷⁶. Moreover, despite multiple reports in preclinical and early-stage clinical trials showing a lack of immunotoxicity, various parameters should be taken into consideration during EV production such as cell source, isolation and storage methods, as these may have a significant impact on to further mitigate potential EV immunogenicity^{77,78}. However, it should be noted that toxicity and immunogenicity studies revealed minimal effects in mice after sustained dosing of HEK293T-derived EVs⁷⁹.

74 C, A. Pseudotyping human immunodeficiency virus type 1 (HIV-1) by the glycoprotein of vesicular stomatitis virus targets HIV-1 entry to an endocytic pathway and suppresses both the requirement for Nef and the sensitivity to cyclosporin A. *J. Virol.* **71**, 5871–5877 (1997).

75. Kuate, S. *et al.* Immunogenicity and efficacy of immunodeficiency virus-like particles pseudotyped with the G protein of vesicular stomatitis virus. *Virology* **351**, 133–144 (2006).
76. Uygur, B., Melikov, K., Arakelyan, A., Margolis, L. B. & Chernomordik, L. V. Syncytin 1 dependent horizontal transfer of marker genes from retrovirally transduced cells. *Sci. Reports* **2019 91 9**, 1–11 (2019).
77. Xia, Y., Zhang, J., Liu, G. & Wolfram, J. Immunogenicity of Extracellular Vesicles. *Adv. Mater.* **36**, 2403199 (2024).
78. Kou, M. *et al.* Mesenchymal stem cell-derived extracellular vesicles for immunomodulation and regeneration: a next generation therapeutic tool? *Cell Death Dis.* **13**, (2022).
79. Zhu, X. *et al.* Comprehensive toxicity and immunogenicity studies reveal minimal effects in mice following sustained dosing of extracellular vesicles derived from HEK293T cells. *J. Extracell. Vesicles* **6**, 1324730 (2017).

REVIEWER COMMENTS

For convenience of the reviewers, all their questions are shown in Green, and our answers are shown in blue. Text that has been added to the manuscript, is highlighted in yellow.

Reviewer #1:

We would like to thank the reviewer for their thorough comments. As the reviewer had added their comments underneath our previous comments, we have removed our original replies as this document would become too confusing and lengthy. However, for context we did retain the original comments (indicated with "Original reviewer comment"), followed by their current comments indicated with "Follow-up reviewer comment"). Our answers / replies are listed thereafter, indicated in blue.

Major comments

1. Original reviewer comment: Please display individual data points in all bar graphs.

Follow-up reviewer comment: Thank you for addressing this.

2. Original reviewer comment: While the systems developed are very elegant and could help new generation of EV therapeutics, gene editing by EVs has already been demonstrated by multiple groups including with disease relevant gene corrections and/or in vivo systems which are lacking in this work. Can the authors please include in vivo experiments or at a minimum targeting of disease relevant genes instead of fluorescent reporters?

Follow-up reviewer comment: It is appreciated that some effort to edit an endogenous gene was included. However, as it is included as a supplementary figure it won't add much value to the study and the doses of EVs are also inconsistent within the figure and across of the rest of the paper. Furthermore, there is a much-needed discussion to be had about the difference in efficiencies observed between the endogenous gene and the reporter gene editing efficiencies. However, this is difficult to interpret because endogenous gene was edited with 7.5×10^{11} EVs per well for Cas9 editing (reaching ~11-22%) whereas 1×10^{11} and 5×10^{11} EVs per well for base editing (reaching ~7% and ~21% respectively) and the fluorescent reporter was edited at a higher 1×10^{12} EVs per well for EVs without or with photocleavable domain attached to CD63 (reaching ~2% and ~28% respectively) but 5×10^{10} EVs per well for EVs with photocleavable domain attached to various candidate domains (reaching ~1-11%). This is a major point separately highlighted below about how difficult it is to assess the different approaches used in the study.

We agree that the results were difficult to compare, as different dosages and cell densities were used in our original revisions. To make it easier to interpret and compare different experiments throughout the manuscript, several experiments have been repeated with dose-ranges or normalized dosages (see next reviewer comment as well).

We have performed several additional experiments to further elucidate on this comment:
- Firstly, to verify the read-out used on endogenous gene editing (T7E1 assay), we performed a dose-range of EV-mediated Cas9 delivery targeting the stoplight construct followed by analysis by flow cytometry and a T7E1 assay. T7E1 analysis shows consistent gene-editing, in a dose-dependent manner. As we have previously observed¹, T7E1 analysis shows lower editing percentages than flow cytometry analysis. Adding 1×10^{12} MCP-PhoCl-CD63 EVs (results in $28.0 \pm 6.2\%$ eGFP activation, and $18.2 \pm 0.2\%$ T7E1 efficiency (Fig. S13A-B). As previously reported, this is due to the Stoplight construct being incorporated multiple times, resulting in a discrepancy between T7E1 and flow cytometry outcomes.

We added the following statements to the manuscript:

Results section:

Finally, we investigated the suitability of MCP-PhoCI-mediated EV CRISPR-Cas9 delivery for endogenous gene-editing, using T7E1 endonuclease assays. To verify suitability of this assay, we first analyzed the stoplight reporter locus after MCP-PhoCI-CD63 VSV-G⁺ EV-mediated delivery of Cas9 at various dosages (Fig. S13A). Indeed, this assay demonstrated consistent and dose-dependent gene-editing, (Fig. S13B). In line with previous observations, measuring gene-editing efficiencies with a T7E1 assay shows lower editing than flow cytometry. This results from multiple genomic integrations of the reporter construct, resulting in lower relative indel numbers as compared to the percentage of cells that have undergone Cas9-mediated reporter recombination¹.

- Next, we repeated EV-mediated Cas9 and ABE8e delivery targeting the *CCR5* locus. Using a similar EV dose and cell density has resulted in a substantial increase in gene-editing efficiency (Fig. S13C-F). Moreover, we included an additional read-out based on the BsaHI restriction enzyme to demonstrate specific enzymatic confirmation of ABE-mediated base editing (Fig. S13G), as ABE activity generates a specific A → G mutation that creates a BsaHI restriction site that is unique in the PCR product of the surrounding locus. These results align more closely with the flow cytometry data in the manuscript. However, as Cas9-based gene-editing efficiency varies highly between different sgRNA sequences, we chose not to make these direct comparisons in the manuscript – as this may differ for other (endogenous) targets.

We added the following statements to the manuscript:

Results section:

To assess gene-editing in an endogenous locus, we targeted the *CCR5* gene on chromosome 3⁴⁶ using a previously verified targeting sequence¹. Using a MS2-sgRNA 1.1 *CCR5* sgRNA (Table S2), Cas9 (Fig. S13C, S13D) and ABE8e (Fig. S13E, S13F) were delivered to HEK293T cells through VSV-G⁺ MCP-PhoCI-CD63 and MCP-PhoCI-CD9 EVs, alongside pDNA transfection. Once more, successful gene editing was confirmed by T7E1 endonuclease analysis, in the *CCR5* locus. As previously observed, MCP-PhoCI-CD9-mediated Cas9 delivery showed higher levels of gene-editing than was observed for MCP-PhoCI-CD63 for both Cas9 and ABE8e-mediated gene editing. As T7E1 assays have decreased efficiency on single-nucleotide substitutions as is often the case with adenine base editors, we performed BsaHI restriction enzyme-mediated analysis of the *CCR5* locus. Alongside increased sensitivity, this read-out is suitable to confirm specific adenine base editor activity of the target *CCR5* locus (Fig. S13G). Concordantly, both EV-mediated ABE8e delivery and pDNA transfection resulted in high levels of BsaHI-mediated cleavage of PCR products of the targeted *CCR5* locus, confirming specific ABE8e-mediated base editing.

Regarding the statement “However, as it is included as a supplementary figure it won’t add much value to the study”, we respectfully disagree. We thank the reviewer for the suggestion to add these experiments to the manuscript, as we feel that they substantially improve relevance and thereby impact of the manuscript. We opted to include these data in the supplementary data as the key focus of the main manuscript is on the establishment and technical aspects of this Cas9 delivery platform, rather than specific clinical applications. For future follow-up studies that focus on specific pathogenic mutations, these data will certainly be included in the main publication. However, in the current manuscript we feel that they represent repeats of experiments that were already illustrated

in the main manuscript using other read-outs, but with different target sgRNA sequences – which is why we opted to place them in the supplementary files for now. That being said, if the editor feels that these should be included in the main manuscript we would be willing to do so.

1. Öktem, M., Mastrobattista, E. & de Jong, O. G. Amphipathic Cell-Penetrating Peptide-Aided Delivery of Cas9 RNP for In Vitro Gene Editing and Correction. *Pharmaceutics* 15, (2023).

3. Original reviewer comment: What was the rationale for EV dosage of “5.0 x 10¹⁰ EVs were added per 24-well plate well containing 1.0 x 10⁵ reporter cells”? Was this the highest possible dose to observe an effect (in terms of recipient cell viability, EV-production capacity, potential clinical application)? It appears that in initial experiments EVs were added at this fixed dose before implementing the PhoCl domain (Fig 2) or other recruiting tags (Fig 3A-E) and only later when trying different tags a dose response was assessed (Figure S3B). An issue with this is that it is a bit hard to follow the effect of each step in improving delivery due to the changes in dosage. In Fig 2D we see that the system has 1.5% editing activity before addition of PhoCl which shows in Fig 3D-E editing improving ~28% but closer inspection of figure legend (but not main text) reveals this can't be compared because dosage increases from 5x10¹⁰ to 1x10¹². Then within the same Fig3, comparison of recruiting tags is shown in 3G-H and editing is closer to 5% with CD63 and 5x10¹⁰ dosage. It may be hard for readers to follow that, if I understand correctly PhoCl may represent a ~3 fold increase (~5%/1.5%) but not a ~19 fold increase (~28%/1.%). To improve the ability to assess the effect of each engineering step please show the results at the same 5x10¹⁰ or 1x10¹² dosage.

Follow-up reviewer comment: These clarifications are a much needed step in the right direction but unfortunately, they also show what appears to be an arbitrary choice of doses which make it so hard for the reader to evaluate the efficiency of different EV loading approaches. Below I am listing the doses used throughout the paper and it doesn't appear to make sense why there are logarithmic fold differences used across the study:

- Fig2: 1E12 EVs per well for EVs without photocleavable domain
- Fig3: 1E12 EVs per well for EVs with photocleavable domain attached to CD63 but 5E10 EVs per well for EVs with photocleavable domain attached to various candidate domains
- Fig4: 4E11 EVs per well for transcriptional activator EVs
- Fig5: 1E10, 5E10, and 1E11 EVs per well for base editing EVs (dose dependency is reasonable but still difficult to compare with other results)
- FigS3: 5E10, 1E11, and 1E12 EVs per well (dose dependency is reasonable but still difficult to compare with other results)
- FigS5: 5E11 EVs per well
- FigS6: Various dose dependency curves, this is ok.
- FigS7: 1.5E11 EVs per well (Normalized to 1.5E11 VSVG EVs?).
- FigS10: 7.5E11 EVs per well for Cas9 editing, 1E11 and 5E11 EVs per well for Base editing, unknown for the off-target assay (S10G), and 5E10 and 5E11 for the cell viability assay.

We thank the reviewer for presenting such a thorough overview of the different dosages used throughout the manuscript, as it clearly illustrates the issue. In part, different dosages are used for different Cas9 moieties (Cas9, dCas9-VPR, ABE8e), as all these Cas9 moieties have different efficiencies. We had initially optimized dosages for all experiments, and performed follow-up experiments with these optimized conditions. However, as we had not included these dose-ranges to the manuscript, we appreciate that this looks somewhat inconsistent. We discussed this situation with the editor, who agreed that including dose-ranges for all Cas9 moieties would provide more clarity. Moreover, we have repeated some experiments to include more consistent experimental conditions. The following changes were made:

- Alongside the original dose response for EV-mediated Cas9 delivery (Fig. S8B), a 1.0×10^{10} – 1.0×10^{12} dose response range for T7E1 analysis on the stoplight locus has now been included (Fig. S13A,B).
- A dose response range for multiple additional reporter cell lines ranging from 1.0×10^{10} – 1.0×10^{12} EVs has now been included
- The MTS viability assay has now been repeated with a 1.0×10^{10} – 1.0×10^{12} dose response.
- T7E1 and BsaHI enzymatic assays for EV-mediated gene-editing efficiencies have been repeated with a similar EV dosage (1.0×10^{12}) and cell count (1.0×10^5) as used throughout the rest of the manuscript (Fig. S13C-I).
- Dose response ranges for MCP-PhoCI-CD63 EV-mediated delivery of dCas9-VPR (Fig. S10A, B) and ABE8e (Fig. S10C, D) have now been included, ranging from 5.0×10^8 – 1.0×10^{12} .
- Next generation sequencing analysis for on- and off-target gene-editing using a 1.0×10^{12} EV dosage.
- The only addition experiment that retains a different dosage is the comparison between different loading fusion proteins (Fig. 3G). As stated previously: “Cells in Fig. 3G were given 5.0×10^{10} EVs. The main reason for the difference in experimental approach was the amount of samples included in Fig. 3G (6 conditions, n=3). Due to practical limitations of our cell culture facilities (limitations in cell culture / incubator space), we opted to compare a lower dosage as to minimize strain on the rest of our cell culture facilities as we performed an n = 3 over the course of 4 weeks for this experiment. However, we would like to point out that observed gene editing efficiency levels from both panels 3D and 3G are in line with the dose response range for MCP-PhoCI-CD63 shown in Fig. S6B.

The following statements have been added to the manuscript:

Dose-range of different reporter cell types:

Results:

To confirm that these observations are not specific to HEK293T reporter cells, a cell line that is easy to transfect⁴², 3 additional stable stoplight reporter cell lines were tested for EV-mediated Cas9 delivery (Fig. S4). Alongside HEK293T cells (Fig. S4A, S4B), human primary tumor cell lines MDA-MB-231 (Fig. S4C, S4D), MCF-7 (Fig. S4E, S4F), and T47D (Fig. S4G, S4H) were treated with various dosages of MCP-PhoCI-CD9 and MCP-PhoCI-CD63 VSV-G⁺ EVs. All cell types showed a dose-dependent EV-mediated delivery of Cas9, albeit at a 5 – 10 fold lower efficiency as compared to HEK293T cells.

Discussion:

Lastly, we also observed substantial differences in Cas9 RNP delivery based on the recipient cell type, and the choice of EV-enriched protein used for targeted loading. Whereas primary tumor cell lines showed lower gene-editing rates than the easy-to-transfect HEK293T cells, differences were within an order of magnitude (Fig. S4). Moreover, as all tested cell types showed significant gene-editing activity, these results suggest potential applicability on a broad range of cells. However, these data also indicate that recipient cell-specific optimization for gene-editing may be necessary

New MTS assay:

Lastly, we performed an MTS assay on HEK293T cells after MCP-PhoCI-CD63 EV or plasmid DNA transfection-mediated delivery of Cas9 to test for potential toxicity (Fig. S6H). The MTS assay showed no signs of toxicity over the course of 72 hours after MCP-PhoCI-CD63 RNP-loaded VSV-G⁺ EVs were added. Plasmid DNA transfection of the same constructs showed a trend towards a low level of toxicity ($p = 0.086$), and treatment with Triton X-100 resulted in complete loss of viability.

New MCP-PhoCI-CD63 EV dCas9-VPR dose range:

A dose-range of MCP-PhoCI-CD63 VSV-G⁺ EVs show a dose-dependent effect on transcriptional activation (Fig. S10A, S10B). However, as compared to delivery of WT Cas9 a substantially higher dose of EVs is required, as dosages of 1.0×10^{11} EVs per 1.0×10^5 cells or lower showed no increase in eGFP fluorescence intensity.

New MCP-PhoCI-CD63 EV ABE8e dose range:

However, EV-mediated delivery of ABE8e using MS2-sgRNA 1.1 (tetraloop MS2 aptamer only), resulted in high levels of dose-dependent EV-mediated ABE8e delivery (Fig. 5F, 5G, S10C, S10D)

New MCP-PhoCI-CD63 T7E1 stoplight dose range & endogenous gene-editing (as discussed above):

Finally, we investigated the suitability of MCP-PhoCI-mediated EV CRISPR-Cas9 delivery for endogenous gene-editing, using T7E1 endonuclease assays. To verify suitability of this assay, we first analyzed the stoplight reporter locus after MCP-PhoCI-CD63 VSV-G⁺ EV-mediated delivery of Cas9 at various dosages (Fig. S13A). Indeed, this assay demonstrated consistent and dose-dependent gene-editing, (Fig. S13B). In line with previous observations, measuring gene-editing efficiencies with a T7E1 assay shows lower editing than fluorescence flow cytometry. This results from multiple genomic integrations of the reporter construct, resulting in lower relative indel numbers as compared to the percentage of cells that have undergone Cas9-mediated reporter recombination¹³. To assess gene-editing in an endogenous locus, we targeted the *CCR5* gene on chromosome 3⁴⁶ using a previously verified targeting sequence¹³. Using a MS2-sgRNA 1.1 *CCR5* sgRNA (Table S2), Cas9 (Fig. S13C, S13D) and ABE8e (Fig. S13E, S13F) were delivered to HEK293T cells through VSV-G⁺ MCP-PhoCI-CD63 and MCP-PhoCI-CD9 EVs, alongside pDNA transfection. Once more, successful gene editing was confirmed by T7E1 endonuclease analysis, in the *CCR5* locus. As previously observed, MCP-PhoCI-CD9-mediated Cas9 delivery showed higher levels of gene-editing than was observed for MCP-PhoCI-CD63 for both Cas9 and ABE8e-mediated gene editing. As T7E1 assays have decreased efficiency on single-nucleotide substitutions as is often the case with adenine base editors, we performed BsaHI restriction enzyme-mediated analysis of the *CCR5* locus. Alongside increased sensitivity, this read-out is suitable to confirm specific adenine base editor activity of the target *CCR5* locus (Fig. S13G). Concordantly, both EV-mediated ABE8e delivery and pDNA transfection resulted in high levels of BsaHI-mediated cleavage of PCR products of the targeted *CCR5* locus, confirming specific ABE8e-mediated base editing.

4. **Original reviewer comment:** Figure 1 G and H show loading of sgRNA into EVs in the presence and absence of MCP-CD63, with an impressive 40~50 fold increase. Nevertheless, instead of showing increase compared to the control lacking MCP-CD63, please represent sgRNA loading in terms of copies per EV. This can easily be obtained using the qPCR results and the NTA particle counts as other studies have shown. For example, it has been reported that far less than one molecule of a given miRNA is present per exosome, actually closer to 1 miRNA per 100 exosomes, even for the most abundant miRNAs (PMID: 25267620). Using such an example, with a 40~50 fold increase in sgRNA does it mean we can assume 1 in 2 exosomes are loaded with sgRNA? Instead of having to assume, the readers would highly benefit from the actual number provided instead of the fold increase parameter. Additionally, this parameter could also help estimate copy number of Cas9 per EV it is actually recruited into EVs by the sgRNA. Lastly, comparison of Cell and EV fractions in 1G appears to indicate more sgRNA content in cells than EVs. Can Figure H also include a comparison relative sgRNA levels between cells and EVs?

Follow-up reviewer comment: This is great and now it is much more informative to the readership, thank you!

Thank you for the suggestion, we feel that it greatly contributed to the quality of the manuscript.

5. **Original reviewer comment:** Figure 2C and the accompanying text describe that “no significant decrease in Cas9-mediated activation of eGFP expression was observed in T MS2-sgRNA as compared to WT T sgRNA”. However there appears to be a 10-20% reduction in GFP activation in the case of MS2-sgRNA. Were these two groups compare statistically to each other or just to the untreated condition? To avoid confusion, the authors should label the statistical significance to NT sgRNA as a better control than untreated where no editing is reasonably expected. Furthermore, if there is no statistically significant difference despite the 10-20% decrease in editing between MS2 and unmodified sgRNA then please also include a label between both bars indicating “n.s.”. However, if data show statistically significant difference in editing, please simply amend the accompanying text to reflect some x% decrease in editing activity is observed with MS2 modification of the sgRNA. Furthermore, Fig 4C also appears to show MS2 modification of the sgRNA results in lower dCas9-VPR transcriptional activation but also here no statistical label is included. Lastly, this is also noted in the discussion section “In line with previous reports, this did not compromise the functionality of Cas9³³” but the statistical comparison is missing indicating “n.s.” differences between the conditions noted above.

Follow-up reviewer comment: This is great, thank you.

6. **Original reviewer comment:** Cleavage of PhoCI domain needs UV-treatment but it has been well established that UV treatment can degrade DNA and RNA. The authors note this in the discussion and point to wavelength for damage being below 340nm while here 395nm is used. If proteins were the main cargo assessed to be functional there would be less or no concerns. However, due to the closeness of these wavelengths the potential degradation of the cargo sgRNA by the UV treatment needs to be further assessed by RNA sequencing. What percentage of cargo sgRNA remains undamaged, what percentage is damaged?

Follow-up reviewer comment: This is great, thank you!

7. **Original reviewer comment:** One of the main concerns for CRISPR gene editing approaches is the potential of unintended offtargeted editing. For every sgRNA sequence potential off target sites can be predicted bioinformatically based on number of mismatches and other parameters. However, in this study’s approach, given the potential of mutating the targeting sgRNA with UV treatment, off-target editing must be evaluated with more care as there may be more than the predicted off-target sites if the sgRNA is mutated. Overall, this aspect is a severe weakness of the study as no off-target analysis was performed. Thus, the authors should include experiments to assess off-targets. It would

be very important to have a control plasmid sgRNA condition, not UV-treated, to compare with EV-delivered and UV-treated sgRNA to assess potential differences in off-target profiles.

Follow-up reviewer comment: The efforts to address this question are also much appreciated but I am afraid there are more minor points can be easily addressed to improve the manuscript. First the authors selected the off-target site as chr2:+120580702 based on prediction with a tool found in an excellent Feng Zhang publication cited as 52. Unfortunately, the tool in that publication, <http://www.genome-engineering.org/>, is no longer available. Nowadays, even the Zhang lab website (<https://www.zlab.bio/resources>) lists another tool called CasOFFinder which is routinely used and cited by many other groups and is available in this website <http://www.rgenome.net/cas-offinder/> to evaluate off-targets. Off-target prediction based on the stoplight reporter guide, GGACAGTACTCCGCTCGAGT, reveals 12 other potential off-targets than the chr2:+120580702 site with the same amount of mismatched bases of 4 without DNA or RNA bulges, and if we include DNA or RNA bulges we find 5 and 10 respective off-target sites with as few as 2 mismatched bases. Most importantly, among other off-target sites with similar or fewer mismatched bases (when including bulges), there are gene coding regions which are much more relevant to study as the chr2:+120580702 targets no gene. These include chrX:+141177037 (4 mismatched bases without bulges) targeting the LDOC1 gene, chr14:-104942108 (4 mismatched bases without bulges) targeting the AHNK2 gene, or chr5:-180630776 (2 mismatched bases, 1 base RNA bulge) targeting the FLT4 gene. The authors should at a minimum revise the off-targeting section with the above considerations and update the off-target search resource, but it would be much preferable to study one from the list of off-target sites within gene coding regions.

Once more, we would like to thank the reviewer for their thorough explanation and advice. We have chosen 5 off-target sequences with 2 – 4 mismatches, targeting the following genes: lncRNA ENSG0000037014, AHNK2, FLK4, and KHDRBS3, as well as the original off-target sequence that did not target a specific gene – as it was the only available off-target sequence with only 2 mismatches. Based on the recommendations of the reviewer (and the appropriate critique of reviewer 3 on the use of TIDE analysis), we have analyzed these 5 off-target sequences, alongside the stoplight targeting locus, for next generation sequencing (NGS). Alongside 1.0×10^{12} MCP-PhoCI-CD63 EVs, untreated cells and cells transfected with Cas9 with a targeting or non-targeting sgRNA were also submitted (n = 3). The results are shown in Supplementary Figure S6). The following statement has been added to the manuscript:

Results:

To determine whether Cas9 delivery through UV-treated EVs resulted in increased off-target gene-editing events due to potential UV-mediated sgRNA damage, genomic DNA of HEK293T stoplight cells treated with MCP-PhoCI-CD63 VSV-G⁺ EVs or plasmid DNA transfection for Cas9 delivery was isolated and analyzed by amplicon next generation sequencing (NGS). Using Cas-OFFinder⁴⁴, 5 off-target sites with 2 – 4 nucleotide mismatches were uncovered and analyzed (Fig. S6A-S6G). EV-mediated delivery did not result in any observable increase in off-target gene-editing events.

Discussion:

Further analyses also revealed no noticeable effects on EV integrity, protein cargo, sgRNA levels, or down-stream of target gene-editing events.

Minor comments

8. **Original reviewer comment:** The authors made use of multiple (12?) addgene constructs to generate the constructs needed to develop their experimental system. However, there is no indication in the manuscript of an intent to deposit any of the generated construct for other researchers to potentially replicate or build upon their findings. I would suggest to please make

these constructs available on addgene as this would be hugely beneficial for the EV community and also would lead to more citations to this work.

Follow-up reviewer comment: That is ok and entirely the right of the authors. However, I would be remiss if I did not point out that future high-quality studies such as this one will not be facilitated if authors or teams have to separately reach out and coordinate with multiple labs for different constructs to arrive or have to order gene synthesis at a much higher cost instead of benefitting from obtaining them all from addgene as the authors of this study did. Overall, I see how other fields such as AAV (5283 addgene plasmids) and CRISPR (21198 plasmids) benefit so much from a more open approach, but the EV field is probably hampered by an absolute lack of constructs and tools available in addgene (eg 18 CD63 EV marker plasmids). I have never made such a minor comment except here it is glaring that the authors benefit from many addgene constructs but provide none back to the scientific community.

Thank you for the clear explanation. Whereas we always made an active effort to include all the sequences of our constructs in the supplementary information of our manuscripts, and we have always been very forthcoming in terms of sharing constructs and cell lines upon request, we did not consider the fact that the EV field is indeed somewhat hampered by a lack of publicly available constructs. As I've always been able to interact directly with many labs in the field through the International Society for Extracellular Vesicles, I must admit that this has been somewhat of a blind spot on my behalf. Whereas our research department has no experience with sharing / depositing plasmids to Addgene, this does align with the Open Science policy our department would like to implement. As of now, our constructs are freely available upon request via direct contact, but I appreciate that this may present an additional hurdle for some. This has given us something to think about..

Reviewer #2 (Remarks to the Author):

The authors have provided additional data and adequately responded the comments. The acceptance of this manuscript for publication is suggested.

Thank you, that is great to hear!

Reviewer #3 (Remarks to the Author):

Elsharkasy et al. reported an EV strategy for CRISPR-Cas9 editor delivery, they encapsulate the Cas9 or other editors by aptamer interaction and UV exposure. I find the experimental design of this work to be reasonable and logically coherent. However, the current version of the manuscript is too long to read. The authors should focus more on the main study and reduce unrelated content and discussion. Please bring the word count down to below 5,000 words, which is in line with the length of most Nature Communications articles. Besides, I have further concerns need to be issued.

Indeed, as it stands our manuscript is quite lengthy. This is in part because of the large amount of additional experiments and points of discussion that were added throughout the review process. That being said, we have been able to streamline some sections to address this issue. However a large number of additional experiments and points of discussion were specifically added per request of the reviewers throughout the review process (hence the somewhat lengthy discussion). The

current word count is slightly above 7000, but in consultation with the editor we would happily streamline the manuscript further if required.

Major concerns:

1. Most of the gene editing experiments described in the manuscript were performed in HEK293T cells, which are relatively easy to transfect. High editing efficiency in this cell line does not indicate therapeutic potential. To strengthen the evidence for an EV-based strategy, it would be more convincing to evaluate its effectiveness in human or mouse primary cells, or in animal models.

We thank the reviewer for their suggestion, and agree with the reviewer that HEK293T cells are indeed relatively easy to transfect – and may therefore not serve as the ideal model for therapeutic applications. Whereas animal models are outside of the scope of this work, and we rather plan to include such models in a follow-up study focusing on specific pathologies, we fully agree that solely including HEK293T cells does not indicate therapeutic potential. To address this issue, we generated 3 reporter cell lines from human primary tumor cell lines with varying reported levels of ease-to-transfect: MCF-7, MDA-MB-231, and T47D. Dose-ranges of both MCP-PhoCI-CD63 and MCP-PhoCI-CD9 EVs between 1.0×10^{10} and 1.0×10^{12} EVs were simultaneously added to all 4 reporter cell lines for direct comparison ($n = 3$). We were able to observe EV-mediated delivery of Cas9 in all cell lines, in a dose-dependent manner. Indeed, as expected the levels of gene-editing were ~5-fold (MCF-7 and MDA-MB-231) to ~9-fold (T47D) lower than to HEK293T cells. Whereas these data confirm that primary cell lines show lower gene-editing levels than HEK293T cells, we did observe successful, consistent, and dose-dependent gene-editing in all cell lines. We would like to thank the reviewer for this suggestion, as we feel that these data provide important context and new insights to the manuscript. The following statements were added:

Results section:

To confirm that these observations are not specific to HEK293T reporter cells, a cell line that is easy to transfect⁴², 3 additional stable stoplight reporter cell lines were tested for EV-mediated Cas9 delivery (Fig. S4). Alongside HEK293T cells (Fig. S4A, S4B), human primary tumor cell lines MDA-MB-231 (Fig. S4C, S4D), MCF-7 (Fig. S4E, S4F), and T47D (Fig. S4G, S4H) were treated with various dosages of MCP-PhoCI-CD9 and MCP-PhoCI-CD63 VSV-G⁺ EVs. All cell types showed a dose-dependent EV-mediated delivery of Cas9, albeit at a 5 – 10 fold lower efficiency as compared to HEK293T cells.

Discussion:

Lastly, we also observed substantial differences in Cas9 RNP delivery based on the recipient cell type, and the choice of EV-enriched protein used for targeted loading. Whereas primary tumor cell lines showed lower gene-editing rates than the easy-to-transfect HEK293T cells, differences were within an order of magnitude (Fig. S4). Moreover, as all tested cell types showed significant gene-editing activity, these results suggest potential applicability on a broad range of cells. However, these data also indicate that recipient cell-specific optimization for gene-editing may be necessary

2. The T7E1 assay and TIDE cannot detect mutations below 1%, making them unsuitable for effectively identifying off-target editing. The authors should consider using deep sequencing for thorough off-target analysis.

Indeed, we agree that the read-out that we had previously chosen to study potential off-target effects was not the optimal choice. To address this issue (as also stated above, in response to comments from reviewer 1), we have chosen 5 off-target sequences with 2 – 4 mismatches, targeting the following genes: lncRNA ENSG0000037014, AHNAK2, FLK4, and KHDRBS3, as well as the original off-target sequence that did not target a specific gene – as it was the only available off-target sequence with only 2 mismatches. We have analyzed these 5 off-target sequences, alongside the stoplight targeting locus, for next generation sequencing (NGS). Alongside 1.0×10^{12} MCP-PhoCl-CD63 EVs, untreated cells and cells transfected with Cas9 with a targeting or non-targeting sgRNA were also submitted (n = 3). The results are shown in Supplementary Figure S6). The following statement has been added to the manuscript:

Results:

To determine whether Cas9 delivery through UV-treated EVs resulted in increased off-target gene-editing events due to potential UV-mediated sgRNA damage, genomic DNA of HEK293T stoplight cells treated with MCP-PhoCl-CD63 VSV-G⁺ EVs or plasmid DNA transfection for Cas9 delivery was isolated and analyzed by amplicon next generation sequencing (NGS). Using Cas-OFFinder⁴⁴, 5 off-target sites with 2 – 4 nucleotide mismatches were uncovered and analyzed (Fig. S6A-S6G). EV-mediated delivery did not result in any observable increase in off-target gene-editing events.

Discussion:

Further analyses also revealed no noticeable effects on EV integrity, protein cargo, sgRNA levels, or down-stream of target gene-editing events.

Minor concerns:

1. VSVG has been used in these extracellular vesicles to enhance their cellular entry. However, in the authors' current schematic, the presence of VSVG is not depicted. It is recommended that the authors revise the schematic to include VSVG.

We apologize for the confusion. Indeed, the initial schematic does not depict the presence of VSV-G, as all experiments prior to Figure 2 did not include VSV-G. VSV-G is only mentioned in the results section throughout the explanation of the results of Figure 2 (where VSV-G was included). As stated in the description of the results from Figure 2 "*To further enhance EV-mediated Cas9 delivery, EV-producing cells were also transfected to express the Vesicular stomatitis virus (VSV) envelope glycoprotein VSV-G.*". Thus, all schematics from figure 2 onwards where VSV-G was included (Fig. 3A, Fig. S9A) contain a depiction of VSV-G, whereas schematics from experiments that do not include VSV-G (Fig. 1A, Fig. S2A, Fig. S12a) show no depiction of VSV-G.

2. Western blot image of Cas9 in Fig3i is over exposure, which is not allowed in Nature Communications.

Our apologies, we have replaced the Western Blot image with an image with a shorter exposure time.

I would like to thank the authors for addressing my earlier comments. I am commenting below in green coloured font to their point by point responses.

Major comments

1. Please display individual data points in all bar graphs.

Individual data points have now been included to all bar graphs in the main manuscript and supplementary figures.

Thank you for addressing this.

2. While the systems developed are very elegant and could help new generation of EV therapeutics, gene editing by EVs has already been demonstrated by multiple groups including with disease relevant gene corrections and/or in vivo systems which are lacking in this work. Can the authors please include in vivo experiments or at a minimum targeting of disease relevant genes instead of fluorescent reporters?

Whereas we feel that in vivo gene editing experiments are outside of the scope of the current manuscript, we agree that targeting of endogenous genetic targets would be of significant value for the relevance and impact of this manuscript. To address this point, we have included data on targeting of the endogenous CCR5 gene by EV-mediated delivery of both Cas9 and the adenine base editor ABE8e. First we verified gene editing of WT and MS2-sgRNAs on the targeted CCR5 locus via plasmid transfection, and then performed a comparison of MCP-PhoCI-CD9 and MCP-PhoCI-CD63 delivery of Cas9. Lastly, we performed a dose-response experiment for MCP-PhoCI-CD63 delivery of ABE8e (shown in the newly included Supplementary Figure S10, panels A – C). The following text was added added to the manuscript:

Finally, we investigated the suitability of MCP-PhoCI-mediated EV CRISPR-Cas9 delivery for endogenous gene-editing. To this end, we targeted the CCR5 gene, which is present on chromosome 3 and consists of four exons and two introns⁵¹. First, we cloned a targeting sequence that we verified in a previous study¹⁷ in plasmids for the expression of WT sgRNA, MS2-sgRNA 1.1, and MS2-sgRNA 2.0. Transfection of these 3 plasmids alongside a plasmid expressing Cas9 into HEK293T cells resulted in similar levels of gene editing of the CCR5 locus, as was confirmed by T7E1 endonuclease analysis (Fig. S10A). Next, CCR5 was targeted by EV-mediated Cas9 delivery using MCP-PhoCI-CD63 and MCP-PhoCI-CD9, with co-expression of VSV-G. Once more, successful CCR5 gene editing was confirmed by T7E1 endonuclease analysis (Fig. S10B). As previously observed, MCP-PhoCI-CD9-mediated Cas9 delivery showed higher levels of gene-editing than was observed for MCP-PhoCI-CD63. Lastly, adenine base editor ABE8e was delivered using MCP-PhoCI-CD63 in combination with a CCR5 targeting MS2-sgRNA 1.1, with co-expression of VSV-G. Using a T7E1 endonuclease assay, a dose-dependent effect on gene editing was observed, confirming successful EV-mediated ABE8e delivery (Fig. S10C).

It is appreciated that some effort to edit an endogenous gene was included. However, as it is included as a supplementary figure it won't add much value to the study and the doses of EVs are also inconsistent within the figure and across of the rest of the paper. Furthermore, there is a much-needed discussion to be had about the difference in efficiencies observed between the endogenous gene and the reporter gene editing efficiencies. However, this is difficult to interpret because endogenous gene was edited with 7.5e11 EVs per well for Cas9 editing (reaching ~11-22%) whereas 1e11 and 5e11 EVs per well for base editing (reaching ~7% and ~21% respectively) and the fluorescent reporter was edited at a higher 1E12 EVs per well for EVs without or with photocleavable domain attached to CD63

(reaching ~2% and ~28% respectively) but 5E10 EVs per well for EVs with photocleavable domain attached to various candidate domains (reaching ~1-11%). This is a major point separately highlighted below about how difficult it is to assess the different approaches used in the study.

3. What was the rationale for EV dosage of “5.0 x 10¹⁰ EVs were added per 24-well plate well containing 1.0 x 10⁵ reporter cells”? Was this the highest possible dose to observe an effect (in terms of recipient cell viability, EV-production capacity, potential clinical application)? It appears that in initial experiments EVs were added at this fixed dose before implementing the PhoCl domain (Fig 2) or other recruiting tags (Fig 3A-E) and only later when trying different tags a dose response was assessed (Figure S3B). An issue with this is that it is a bit hard to follow the effect of each step in improving delivery due to the changes in dosage. In Fig 2D we see that the system has 1.5% editing activity before addition of PhoCl which shows in Fig 3D-E editing improving ~28% but closer inspection of figure legend (but not main text) reveals this can't be compared because dosage increases from 5x10¹⁰ to 1x10¹². Then within the same Fig3, comparison of recruiting tags is shown in 3G-H and editing is closer to 5% with CD63 and 5x10¹⁰ dosage. It may be hard for readers to follow that, if I understand correctly PhoCl may represent a ~3 fold increase (~5%/1.5%) but not a ~19 fold increase (~28%/1%). To improve the ability to assess the effect of each engineering step please show the results at the same 5x10¹⁰ or 1x10¹² dosage.

We agree with the reviewer that this description was somewhat confusing, as this did not apply to all experiments. Instead, we have now clearly listed the amount of EVs added in the respective figure legends of all EV addition experiments. Moreover, to avoid confusion the statement

“Unless stated otherwise, 5.0 x 10¹⁰ EVs were added per 24-well plate well containing 1.0 x 10⁵ reporter cells” in the Methods section has been replaced with the following statement:

Unless stated otherwise, cells subjected to EV addition experiments were cultured in 24-well plate wells at a density of 1.0 x 10⁵ cells per well in 1 ml culture medium at the time of EV addition.

Moreover, we agree that that the comparison between the experiments in panel 2D and 3D is highly valuable to determine the effect of PhoCl-mediated cargo release on RNP delivery, which is why it's particularly unfortunate that we had previously omitted to clearly mention that both conditions had received the same amount of EVs (1.0 x 10¹²). We have addressed this by updating figure legends as stated above, but we have also included an additional dose response range of MCP-CD63 RNP EV delivery (shown in the newly included Fig. S3A, S3B), as this was previously only shown for MCP-PhoCl-CD63 RNP delivery (Fig. S6B). We thank the reviewer for pointing out this lack of clarity.

We have added the following text to refer to the newly added dose response range:

Addition of 5.0 x 10¹⁰ and 1.0 x 10¹¹ EVs showed even lower levels of gene editing, at 0.2% and 0.3% respectively (Fig. S3A, S3B).

These clarifications are a much needed step in the right direction but unfortunately, they also show what appears to be an arbitrary choice of doses which make it so hard for the reader to evaluate the efficiency of different EV loading approaches. Below I am listing the doses used throughout the paper and it doesn't appear to make sense why there are logarithmic fold differences used across the study:

- Fig2: 1E12 EVs per well for EVs without photocleavable domain
- Fig3: 1E12 EVs per well for EVs with photocleavable domain attached to CD63 but 5E10 EVs per well for EVs with photocleavable domain attached to various candidate domains
- Fig4: 4E11 EVs per well for transcriptional activator EVs

- Fig5: 1E10, 5E10, and 1E11 EVs per well for base editing EVs (dose dependency is reasonable but still difficult to compare with other results)
- FigS3: 5E10, 1E11, and 1E12 EVs per well (dose dependency is reasonable but still difficult to compare with other results)
- FigS5: 5E11 EVs per well
- FigS6: Various dose dependency curves, this is ok.
- FigS7: 1.5E11 EVs per well (Normalized to 1.5E11 VSVG EVs?).
- FigS10: 7.5E11 EVs per well for Cas9 editing, 1E11 and 5E11 EVs per well for Base editing, unknown for the off-target assay (S10G), and 5E10 and 5E11 for the cell viability assay.

4. Figure 1 G and H show loading of sgRNA into EVs in the presence and absence of MCP-CD63, with an impressive 40~50 fold increase. Nevertheless, instead of showing increase compared to the control lacking MCP-CD63, please represent sgRNA loading in terms of copies per EV. This can easily be obtained using the qPCR results and the NTA particle counts as other studies have shown. For example, it has been reported that far less than one molecule of a given miRNA is present per exosome, actually closer to 1 miRNA per 100 exosomes, even for the most abundant miRNAs (PMID: 25267620). Using such an example, with a 40~50 fold increase in sgRNA does it mean we can assume 1 in 2 exosomes are loaded with sgRNA? Instead of having to assume, the readers would highly benefit from the actual number provided instead of the fold increase parameter. Additionally, this parameter could also help estimate copy number of Cas9 per EV it is actually recruited into EVs by the sgRNA. Lastly, comparison of Cell and EV fractions in 1G appears to indicate more sgRNA content in cells than EVs. Can Figure H also include a comparison relative sgRNA levels between cells and EVs?

We agree that absolute counts of sgRNA per EV would be highly informative, and give more context to the engineering strategies employed in this manuscript. Unfortunately, we were not able to directly convert qPCR data into absolute sgRNA counts as no absolute synthetic sgRNA range was included. Instead, we opted to perform digital droplet PCR (ddPCR). When combined with NTA particle count and normalization to a Spike-in RNA to correct for RNA isolation efficiency, this technique allows for calculation of sgRNAs per EV. These data are shown in the newly included panel 1H.

Moreover, as previously stated in the Methods section of the first submission, qPCR data was derived from a different cell line that was stably expressing sgRNA +/- MCP-CD63. As these conditions are less suitable for a direct comparison to the rest of the data in this manuscript, this panel has been replaced with qPCR on HEK293T-derived EVs, loaded with Cas9 RNP using similar protocols as used in the rest of the manuscript. This is now shown in Figure 1G.

The following text has been added to the manuscript:

To assess whether MS2-sgRNA was also actively enriched in EVs by MCP-CD63, MS2-sgRNA loading was analyzed by qPCR (Fig. 1G). As was also observed for Cas9 protein loading, co-expression of MCP-CD63 resulted in a significant increase in MS2-sgRNA abundance in isolated EVs, showing an approx. 270 fold increase (SD \pm 94). To more accurately and quantitatively assess MS2-sgRNA loading, digital droplet PCR (ddPCR) was performed (Fig. 1H). Using a Spike-in RNA to correct for RNA isolation efficiency, absolute RNA loading was extrapolated from NTA particle counts. In line with previous results, a substantial increase in MS2-sgRNA loading was observed of 350-fold (SD \pm 12.4), increasing the abundance from 1 sgRNA per $\sim 6.0 \times 10^6$ EVs to 1 sgRNA per $\sim 1.7 \times 10^4$ EVs after active loading using MCP-CD63. Whereas these numbers are low as compared to synthetic nanocarriers, such orders of

magnitude are not unexpected as similar numbers have previously been reported for sgRNA loading into EVs in other cell lines^{36,46}.

36. de Jong, O. G. *et al.* A CRISPR-Cas9-based reporter system for single-cell detection of extracellular vesicle-mediated functional transfer of RNA. *Nat. Commun.* **11**, (2020).

46. Murphy, D. E. *et al.* Natural or synthetic RNA delivery: A stoichiometric comparison of extracellular vesicles and synthetic nanoparticles. *Nano Lett.* **21**, 1888–1895 (2021).

Lastly, we opted not to include direct relative expression comparison between cell and EV lysates, as direct comparison of RNA content between cell and EV RNA content in qPCR remains somewhat of a debated topic: is it fair to directly compare these? Normalization on RNA amount is not directly physiologically accurate, and housekeeping genes suitable for such comparisons are lacking / somewhat controversial. For endogenous genes, studies often employ comparisons of its expression levels to other endogenous genes, so enrichment could be extrapolated from RNA ratios between certain genes in EVs and cells. However, as sgRNAs are non-endogenous RNA molecules, a common housekeeping gene suitable for direct comparison between cells and EVs would be required. For context, a previous position paper from the International Society for Extracellular Vesicles states the following “The choice of a normalization control is important when comparing expression levels between replicates and treatment conditions. It should be noted that those household genes that are generally used to normalize expression levels between samples of cellular RNA are not per se appropriate normalization controls in samples of evRNA.” (ISEV position paper: extracellular vesicle RNA analysis and bioinformatics, A. Hill et al., *Journal of Extracellular Vesicles*, 2013). As such, we opted to focus on the specific enrichment of cargo in EVs instead, as is currently shown in panels 1G and 1H.

This is great and now it is much more informative to the readership, thank you!

5. Figure 2C and the accompanying text describe that “no significant decrease in Cas9-mediated activation of eGFP expression was observed in T MS2-sgRNA as compared to WT T sgRNA”. However there appears to be a 10-20% reduction in GFP activation in the case of MS2-sgRNA. Were these two groups compare statistically to each other or just to the untreated condition? To avoid confusion, the authors should label the statistical significance to NT sgRNA as a better control than untreated where no editing is reasonably expected. Furthermore, if there is no statistically significant difference despite the 10-20% decrease in editing between MS2 and unmodified sgRNA then please also include a label between both bars indicating “n.s.”. However, if data show statistically significant difference in editing, please simply amend the accompanying text to reflect some x% decrease in editing activity is observed with MS2 modification of the sgRNA. Furthermore, Fig 4C also appears to show MS2 modification of the sgRNA results in lower dCas9-VPR transcriptional activation but also here no statistical label is included. Lastly, this is also noted in the discussion section “In line with previous reports, this did not compromise the functionality of Cas933” but the statistical comparison is missing indicating “n.s.” differences between the conditions noted above.

Statistical analysis on the data of figure 2C indeed confirmed that there was no statistically significant differences between the WT sgRNA and the MS2-sgRNA. For clarity, an additional bar between these samples indicating “n.s.” has been included.

In the first submission of the manuscript, no direct statistical comparison was made between WT sgRNA and MS2-sgRNA transfection, as a Dunnett’s multiple comparisons test was initially performed. To answer the question regarding statistical comparison of these sgRNAs in Fig. 4C, the statistical analysis was replaced with a post-hoc Sidaks multiple comparisons test. Indeed, unlike seen in WT Cas9 in Fig. 2C, there does appear to be a significant decrease in transcriptional activation between WT

sgRNA and MS2-sgRNA. Figure 4C has been replaced with the new statistical analysis and the following statement has been added to the Results section of the manuscript:

Unlike previously observed in Cas9 gene editing (Fig. 2C), transfection of MS2-sgRNA does show a lower transcriptional activation of eGFP expression as compared to WT sgRNA. However, MS2-sgRNA does still show substantial and significant transcriptional activation (Fig. 4B, 4C).

Moreover, the following statement was added to the Discussion section:

We did observe a decrease in dCas9-VPR-mediated transcriptional activation in MS2-sgRNAs as compared to unmodified sgRNAs. However, as this sgRNA still showed substantial and significant activity, we were still able to demonstrate transcriptional activation levels comparable to plasmid DNA transfection using MCP-PhoCl-CD9 (Fig. 4D).

We would like to thank the reviewer for pointing this out, as this finding may give future leads to further increase EV-mediated transcriptional activation.

This is great, thank you.

6. Cleavage of PhoCl domain needs UV-treatment but it has been well established that UV treatment can degrade DNA and RNA. The authors note this in the discussion and point to wavelength for damage being below 340nm while here 395nm is used. If proteins were the main cargo assessed to be functional there would be less or no concerns. However, due to the closeness of these wavelengths the potential degradation of the cargo sgRNA by the UV treatment needs to be further assessed by RNA sequencing. What percentage of cargo sgRNA remains undamaged, what percentage is damaged?

We agree that a more in-depth analysis of the effects of 395 nm UV-treatment on our EVs is relevant, and was insufficiently addressed in the initial submission of the manuscript. Thus, to address this question, we chose to evaluate the effects of UV treatment on EV sgRNA content, as well as size distribution, protein content, and EV morphology using ddPCR, NTA analysis, immunoblotting, total protein stain, and transmission electron microscopy. These analyses showed no notable difference in EV count, size, sgRNA content, protein content, or EV morphology. Data are presented in the newly included Figure S4A-S4J. The following text has been added to the Results section:

To further study whether UV treatment had any deleterious effects on EV morphology or cargo, MCP-CD63 EVs were treated for 20 minutes with UV and characterized by NTA analysis (Fig. S4). No significant changes in size distribution (Fig. S4A, S4B), mean and mode particle size (Fig. S4C, S4D), or in particle count (Fig. S4E) were observed. Whereas UV wavelengths generally associated with nucleotide damage are below 340 nm⁴⁸, we measured the effect of UV treatment on MS2-sgRNA abundance in MCP-CD63 RNP-loaded EVs using qPCR (Fig. S4F) and ddPCR (Fig. S4G) analysis to rule out UV-mediated sgRNA degradation. Indeed, both qPCR and ddPCR showed no difference in EV MS2-sgRNA abundance after UV treatment. Lastly, we measured the effect of UV treatment on protein cargo content and studied the effect on EV integrity and morphology using western blot analysis and transmission electron microscopy, respectively. Western blot analysis showed no difference in levels of Cas9 protein after UV treatment, or in EV markers ALIX, CD63, and syntenin. Total protein stain also showed no notable differences (Fig. S4H). Transmission electron microscopy also did not reveal any changes in EV morphology, integrity, or particle count (Fig. S4I, S4J). Altogether, we observe no evidence of notable deleterious changes in EV content or morphology after 20 minutes of 50W 395 nm UV exposure.

This is great, thank you!

7. One of the main concerns for CRISPR gene editing approaches is the potential of unintended off-target editing. For every sgRNA sequence potential off target sites can be predicted bioinformatically based on number of mismatches and other parameters. However, in this study's approach, given the potential of mutating the targeting sgRNA with UV treatment, off-target editing must be evaluated with more care as there may be more than the predicted off-target sites if the sgRNA is mutated. Overall, this aspect is a severe weakness of the study as no off-target analysis was performed. Thus, the authors should include experiments to assess off-targets. It would be very important to have a control plasmid sgRNA condition, not UV-treated, to compare with EV-delivered and UV-treated sgRNA to assess potential differences in off-target profiles.

Indeed, alongside intracellular delivery, one of the main concerns for CRISPR-Cas9 gene therapy are off-target editing events. We agree that, independent of the delivery platform, this is an important hurdle for the field. However, we would like once more underline that multiple studies have shown that both DNA and RNA damage does not happen at wavelengths above 340 nm, but rather in the UVC (short wave, 100 – 280 nm) and UVB (middle-wave, 280 -320 nm) range, rather than the UVA (320 – 400 nm) range that our experiments are performed in: <https://pubmed.ncbi.nlm.nih.gov/22816040> <https://pubmed.ncbi.nlm.nih.gov/23856615>, <https://pubmed.ncbi.nlm.nih.gov/21613571>. However, we agree with the reviewer that it's better to err on the side of caution. To address this question, the main off-target sequence from our Stoplight targeting sgRNA sequence (chr2:+120580702) was analyzed for off-target effects by both a T7E1 endonuclease assay (Fig. S10E) and Tracking of Indels by DEcomposition (TIDE) analysis (Fig. S10F, S10G). Whereas clear gene editing activity was observed on the Stoplight targeting locus using T7E1 analysis (Fig. S10D), no difference was observed between EV-treated samples, untreated samples, and samples transfected with a Cas9 with non-targeting sgRNA was observed. The following text has been added to the manuscript:

One of the main concerns for therapeutic applications of Cas9-mediated endogenous gene editing is the potential for off-target gene editing effects. To assess potential off-target effects of EV-mediated Cas9 delivery, the highest predicted off-target site for the Cas9 Stoplight targeting sgRNA, chr2:+120580702 was analyzed⁵². First, Cas9 delivery and gene-editing activity was verified by analysis of the targeted Cas9 Stoplight locus, where both MCP-PhoCI-CD63 EV and plasmid DNA transfection-mediated delivery of Cas9 showed high levels of editing (42% ± 2.2%, and 66% ± 6.8%, respectively; Fig. S10D). Next, off-target activity on chr2:+120580702 position was analyzed by a T7E1 endonuclease (Fig. S10E), after MCP-PhoCI-CD63 EV or plasmid DNA transfection-mediated delivery of Cas9. Both T7E1 endonuclease analysis showed some low levels of T7E1 background activity, but these were similar to T7E1 activity observed in samples treated with pDNA transfection of a non-targeting sgRNA (Fig. S10E). For a more precise quantification of off-target activity, we performed a more sensitive Tracking of Indels by DEcomposition (TIDE) analysis⁴⁴ on these samples (Fig. S10F, S10G). Here, once more, no difference was observed with both untreated samples and samples transfected with plasmid DNA expression Cas9 and a non-targeting sgRNA.

44. Brinkman, E. K., Chen, T., Amendola, M. & Van Steensel, B. Easy quantitative assessment of genome editing by sequence trace decomposition. *Nucleic Acids Res.* **42**, e168–e168 (2014).

52. Hsu, P. D. *et al.* DNA targeting specificity of RNA-guided Cas9 nucleases. *Nat. Biotechnol.* **2013** 319 **31**, 827–832 (2013).

The efforts to address this question are also much appreciated but I am afraid there are more minor points can be easily addressed to improve the manuscript. First the authors selected the off-target site as chr2:+120580702 based on prediction with a tool found in an excellent Feng Zhang publication cited as 52. Unfortunately, the tool in that publication, <http://www.genome-engineering.org/>, is no longer

available. Nowadays, even the Zhang lab website (<https://www.zlab.bio/resources>) lists another tool called CasOFFinder which is routinely used and cited by many other groups and is available in this website <http://www.rgenome.net/cas-offinder/> to evaluate off-targets. Off-target prediction based on the stoplight reporter guide, GGACAGTACTCCGCTCGAGT, reveals 12 other potential off-targets than the chr2:+120580702 site with the same amount of mismatched bases of 4 without DNA or RNA bulges, and if we include DNA or RNA bulges we find 5 and 10 respective off-target sites with as few as 2 mismatched bases. Most importantly, among other off-target sites with similar or fewer mismatched bases (when including bulges), there are gene coding regions which are much more relevant to study as the chr2:+120580702 targets no gene. These include chrX:+141177037 (4 mismatched bases without bulges) targeting the LDOC1 gene, chr14:-104942108 (4 mismatched bases without bulges) targeting the AHNAK2 gene, or chr5:-180630776 (2 mismatched bases, 1 base RNA bulge) targeting the FLT4 gene.

The authors should at a minimum revise the off-targeting section with the above considerations and update the off-target search resource, but it would be much preferable to study one from the list of off-target sites within gene coding regions.

Minor comments

8. The authors made use of multiple (12?) addgene constructs to generate the constructs needed to develop their experimental system. However, there is no indication in the manuscript of an intent to deposit any of the generated construct for other researchers to potentially replicate or build upon their findings. I would suggest to please make these constructs available on addgene as this would be hugely beneficial for the EV community and also would lead to more citations to this work.

We appreciate the suggestion, as we have indeed greatly benefitted from the availability of the constructs of other research groups via Addgene for this work. We are currently in the process of writing a manuscript that contains a variety of fluorescent reporter for Cas9 activity that we intend to publish as a single "CRISPR-Cas reporter tool set" that we intend to submit to Addgene. This collection will also contain all the reporter sequences used in this work as well. In the meantime, as we have done in previous work, we have included all sequences for our constructs in the supplementary information of this manuscript, and are (and previously have) always been willing to ship plasmids to other groups on request.

That is ok and entirely the right of the authors. However, I would be remiss if I did not point out that future high-quality studies such as this one will not be facilitated if authors or teams have to separately reach out and coordinate with multiple labs for different constructs to arrive or have to order gene synthesis at a much higher cost instead of benefitting from obtaining them all from addgene as the authors of this study did. Overall, I see how other fields such as AAV (5283 addgene plasmids) and CRISPR (21198 plasmids) benefit so much from a more open approach, but the EV field is probably hampered by an absolute lack of constructs and tools available in addgene (eg 18 CD63 EV marker plasmids). I have never made such a minor comment except here it is glaring that the authors benefit from many addgene constructs but provide none back to the scientific community.